# MRE11 liberates cGAS from nucleosome sequestration during tumorigenesis

Min-Guk Cho[1,11], Rashmi J. Kumar[1,2,11], Chien-Chu Lin[3], Joshua A. Boyer[4], Jamshaid A. Shahir[5], Katerina Fagan-Solis[1], Dennis A. Simpson[1], Cheng Fan[1], Christine E. Foster[1], Anna M. Goddard[1,6], Lynn M. Lerner[1,7], Simon W. Ellington[1], Qinhong Wang[1], Ying Wang[1,7], Alice Y. Ho[8], Pengda Liu[1,4], Charles M. Perou[1,7,9], Qi Zhang[1,4], Robert K. McGinty[1,3,4], Jeremy E. Purvis[1,7,9] & Gaorav P. Gupta[1,2,4,7,10 ✉]

Oncogene-induced replication stress generates endogenous DNA damage that activates cGAS–STING-mediated signalling and tumour suppression[1–3]. However, the precise mechanism of cGAS activation by endogenous DNA damage remains enigmatic, particularly given that high-affinity histone acidic patch (AP) binding constitutively inhibits cGAS by sterically hindering its activation by double-stranded DNA (dsDNA)[4–10]. Here we report that the DNA double-strand break sensor MRE11 suppresses mammary tumorigenesis through a pivotal role in regulating cGAS activation. We demonstrate that binding of the MRE11–RAD50–NBN complex to nucleosome fragments is necessary to displace cGAS from acidic-patch-mediated sequestration, which enables its mobilization and activation by dsDNA. MRE11 is therefore essential for cGAS activation in response to oncogenic stress, cytosolic dsDNA and ionizing radiation. Furthermore, MRE11-dependent cGAS activation promotes ZBP1–RIPK3–MLKL-mediated necroptosis, which is essential to suppress oncogenic proliferation and breast tumorigenesis. Notably, downregulation of *ZBP1* in human triple-negative breast cancer is associated with increased genome instability, immune suppression and poor patient prognosis. These findings establish MRE11 as a crucial mediator that links DNA damage and cGAS activation, resulting in tumour suppression through ZBP1-dependent necroptosis.

Chromosomal instability (CIN) is associated with poor patient prognosis across many different cancer types[11]. Oncogene-induced replication stress is an established driver of CIN through the generation of replication-associated DNA damage[12]. Despite this knowledge, how cancers adapt to tolerate chronically increased levels of DNA damage and CIN remains poorly understood. Although p53 deficiency is an important aspect of this adaptive process, p53-deficient cancers exhibit heterogenous levels of CIN that point to the existence of additional regulatory mechanisms.

A by-product of oncogene-induced replication stress is the accumulation of cytoplasmic chromatin fragments, which are transmitted to the cytoplasm through aberrant mitoses and micronucleus formation[3]. Cytoplasmic chromatin fragments comprise histone-bound DNA fragments and are associated with the activation of cGAS–STING-dependent innate immune signalling that promotes cellular senescence and tumour suppression[1,2,13]. Although cGAS can directly bind dsDNA to stimulate its 2′3′-cyclic GMP-AMP (2′3′-cGAMP) synthase enzymatic activity[14,15], recent studies have demonstrated that cGAS is

constitutively bound to and inhibited by nuclear chromatin. Specifically, cGAS is inhibited by high-affinity binding to the histone H2A-H2B acidic patch (AP) region on the nucleosome disk face that prevents its oligomerization and activation in response to dsDNA[4–10,16,17]. How cGAS is activated by chromatin-associated self-DNA, despite these constitutively inhibitory histone interactions, is currently unknown.

In this study, we investigate how tumours adapt to tolerate DNA damage caused by replication stress through alterations in DNA damage response (DDR) genes. Through our analyses, we identify crucial mechanisms involved in transducing DNA damage signals to activate cGAS and initiate downstream innate immune responses. Our findings argue that innate immune sensing of DNA damage is an important barrier to oncogenic transformation of p53-deficient preneoplasias.

Breast cancers in The Cancer Genome Atlas (TCGA)[18] with both *MYC* amplification and *TP53* alterations exhibit increased levels of CIN, DNA double-strand break signalling (for example, phosphorylated CHK2 (pCHK2)) and replication stress (for example, pCHK1) (Fig. 1a). We recapitulated this molecular profile of breast

[1]Lineberger Comprehensive Cancer Center, University of North Carolina at Chapel Hill, Chapel Hill, NC, USA. [2]UNC MD–PhD Program, University of North Carolina at Chapel Hill School of Medicine, Chapel Hill, NC, USA. [3]Division of Chemical Biology and Medicinal Chemistry, Eshelman School of Pharmacy, University of North Carolina at Chapel Hill, Chapel Hill, NC, USA. [4]Department of Biochemistry and Biophysics, University of North Carolina at Chapel Hill, Chapel Hill, NC, USA. [5]Curriculum in Bioinformatics and Computational Biology, University of North Carolina at Chapel Hill, Chapel Hill, NC, USA. [6]Department of Pathology and Laboratory Medicine, University of North Carolina at Chapel Hill, Chapel Hill, NC, USA. [7]Curriculum in Genetics and Molecular Biology, University of North Carolina at Chapel Hill, Chapel Hill, NC, USA. [8]Department of Radiation Oncology, Massachusetts General Hospital, Boston, MA, USA. [9]Computational Medicine Program, University of North Carolina at Chapel Hill, Chapel Hill, NC, USA. [10]Department of Radiation Oncology, University of North Carolina at Chapel Hill, Chapel Hill, NC, USA. [11]These authors contributed equally: Min-Guk Cho, Rashmi J. Kumar. ✉e-mail: gaorav_gupta@med.unc.edu

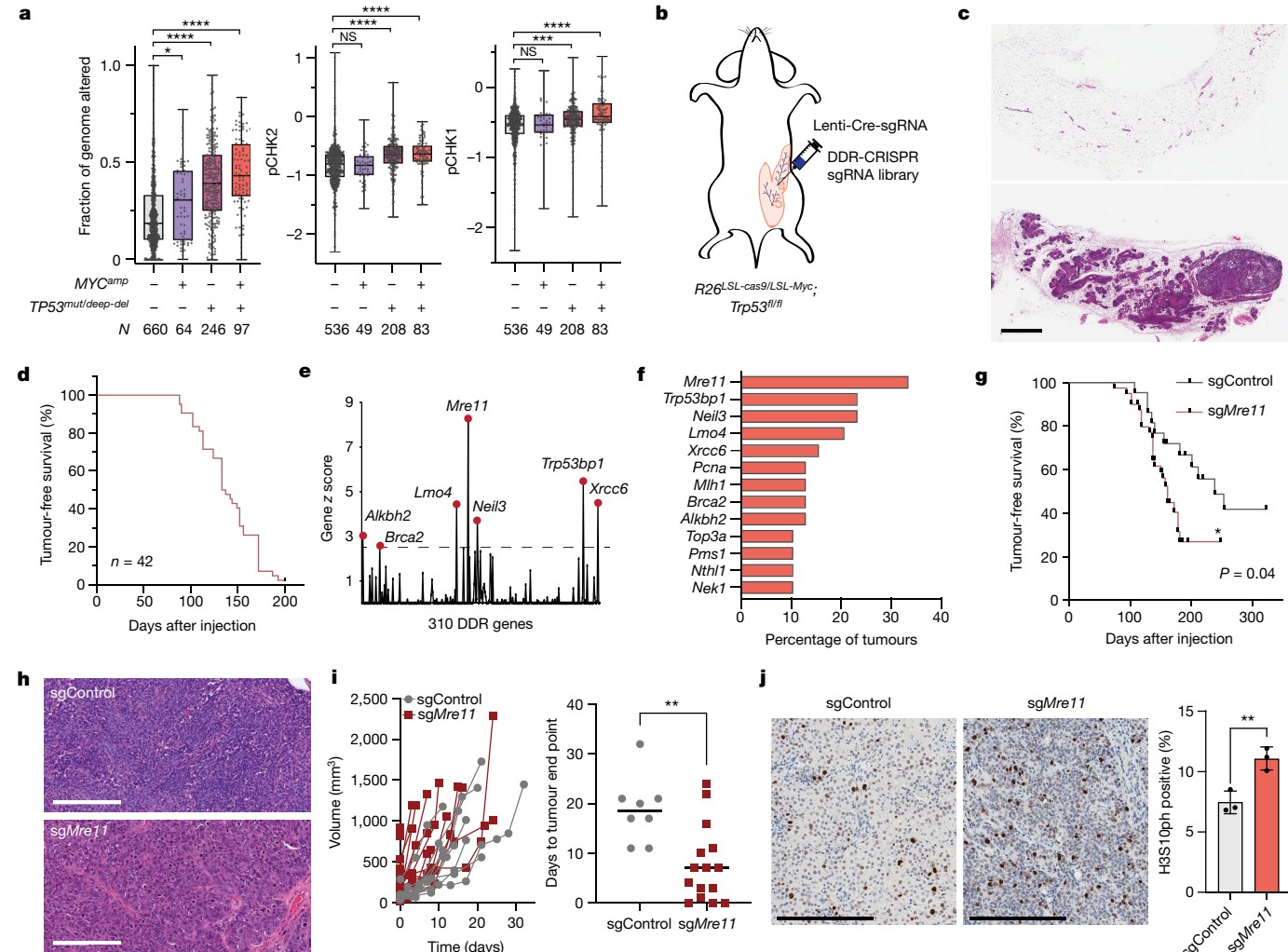

**Fig. 1 | MRE11 suppresses breast tumorigenesis driven by MYC overexpression and p53 deficiency. a,** Fraction of genome altered (left), and RPPA signals for pCHK2(T68) (middle) and pCHK1(S345) (right) in TCGA breast cancers stratified by *MYC* amplification (*MYC^amp^*) and *TP53* alterations (*TP53^mut/deep-del^*), analysed by one-way analysis of variance (ANOVA). Box plots show the median, the 25th to 75 percentiles and range. *N* indicates the number of independent tumours from the TCGA-BRCA Pan-Cancer Atlas dataset. *P* values (left to right): fraction of genome altered, $P = 0.0251$, $P < 0.0001$, $P < 0.0001$; pCHK2, $P = 0.9999$, $P < 0.0001$, $P < 0.0001$; pCHK1, $P = 0.9999$, $P = 0.0003$, $P < 0.0001$. **b,** Schematic of the transgenic breast cancer model used for the DDR-CRISPR sgRNA library screen. **c,** Sections from a non-injected mammary gland (top) and a mammary gland 4 months after DDR-CRISPR lentiviral injection (bottom) and stained with haematoxylin and eosin. **d,** Kaplan–Meier tumour-free survival curve after injection with the DDR-CRISPR library. **e,** Normalized gene *z* scores relative to control. **f,** Bar plot depicting the most commonly targeted DDR genes. **g,** Kaplan–Meier tumour-free survival curves of *R26^LSL-cas9/LSL-Myc^;Trp53^fl/fl^* female mice after mammary intraductal injection with Cre-sgControl (*n* = 24 biologically independent animals) or Cre-sg*Mre11* (*n* = 42 biologically independent animals), analysed by two-tailed log-rank test. **h,** Sections of mammary tumours induced by Cre-sgControl or Cre-sg*Mre11* (bottom) stained with haematoxylin and eosin. **i,** Spider plot (left) of tumour growth after initial palpation of mammary tumours and bar plot (right) of time from palpation to tumour end point. sgControl versus sg*Mre11*, $P = 0.0033$. **j,** Representative immunohistochemistry images (left and middle) and quantification (right) of histone H3S10ph. sgControl (*n* = 3 biologically independent tumours) versus sg*Mre11* (*n* = 3 biologically independent tumours), $P = 0.01$. Graph depicts mean ± s.e.m. Grouped comparisons analysed by unpaired, two-tailed *t*-test. NS, not significant, *$P < 0.05$, **$P < 0.01$, ***$P < 0.001$, ****$P < 0.0001$. Scale bar, 200 μm (**h,j**), 1 mm (**c**).

cancer using a mouse transgenic model with conditional alleles for *Myc* overexpression, *cas9* expression and *Trp53* deficiency (*Rosa26^LSL-Myc/LSL-cas9^;Trp53^fl/fl^*) (Fig. 1b). Breast tumorigenesis was initiated through intraductal injection into mammary glands of lentivirus expressing Cre recombinase and a single guide RNA (sgRNA) of interest that directs Cas9 cleavage to specific genomic targets, as previously described[19]. Mammary tumours from this genetic model induced by lentivirus-expressing Cre and control sgRNA (sgControl, which targets an intronic sequence on chromosome 2) exhibited abundant cytoplasmic DNA damage (gH2A.X) foci and cytoplasmic cGAS localization (Extended Data Fig. 1a). Based on these observations, we postulated that the cellular response to MYC-induced DNA damage may promote

cGAS activation and tumour suppression in this transgenic breast cancer model.

To evaluate this hypothesis, we conducted an in vivo CRISPR screen for DDR genes that suppress mammary tumorigenesis induced by MYC overexpression and p53 deficiency using a lentiviral sgRNA library targeting 310 DDR genes (DDR-CRISPR)[20]. A total of 42 female *Rosa26^LSL-Myc/LSL-cas9^;Trp53^fl/fl^* mice 6–10 weeks old were injected with $5 \times 10^5$ transduction units (TU) of the DDR-CRISPR lentiviral library into the fourth mammary glands. Examination of mammary glands 4 months after intraductal injection with the DDR-CRISPR library revealed polyclonal mammary epithelial outgrowths of heterogenous sizes, consistent with clonal expansion and competition (Fig. 1c). Mice

developed mammary tumours with a median latency of 137 days (Fig. 1d). Genomic DNA from 39 mammary tumours were successfully analysed for sgRNA abundance relative to DDR-CRISPR plasmid controls. Positive selection for DDR gene-targeting sgRNAs was evident across the cohort of mammary tumours. Overall, 35 DDR genes were recurrently targeted in the tumour cohort, including well-established tumour suppressors such as *Brca2* and *Palb2*, and mismatch repair pathway genes such as *Mlh1*, *Pms1* and *Msh6*. The most frequently targeted genes were *Mre11*, *Neil3*, *S3bp1*, *Lmo4*, *Xrcc6*, *Alkbh2*, *Brca2*, *Mlh1* and *Pcna* (Fig. 1e,f). MRE11 is a multifunctional sensor of DNA double-strand breaks that stimulates ATM activation and facilitates DNA repair through nuclease and scaffolding functions. Notably, the predominant sgRNA against *Mre11* (sg*Mre11*) enriched in the screen targeted the carboxy-terminal region of the coding sequence, where frameshift-truncating mutations resemble the naturally occurring *Mre11*[ATLD1] hypomorphic allele[21].

To validate the tumour-suppressive role of MRE11 in this breast cancer model, we injected an independent group of *Rosa26*[LSL-Myc/LSL-cas9];*Trp53*[fl/fl] mice intraductally with lentivirus expressing Cre-sgControl (n = 22 glands) or Cre-sg*Mre11* (n = 40 glands). Targeting *Mre11* significantly shortened tumour latency compared with the control sgRNA (Fig. 1g). Histopathological analyses of mammary tumours from both sgControl and sg*Mre11* groups revealed high grade invasive ductal carcinomas (Fig. 1h). After tumour onset, *Mre11* mutant tumours exhibited rapid growth, which resulted in a shorter time interval from initial palpation to tumour collection (Fig. 1i). Immunohistochemistry for histone H3S10ph also revealed a high mitotic index in *Mre11* mutant tumours (Fig. 1j). Additionally, *Mre11* mutant tumours displayed an increased frequency of mitotic aberrations, particularly chromatin bridges (Extended Data Fig. 1b). These observations indicate that *Mre11* mutant tumours exhibit increased proliferation despite heightened genome instability, which indicates a deficiency in DNA damage-induced checkpoints. Previous studies have also demonstrated the suppressive role of MRE11 in *Her2/Neu*[+] and *Rb1*[−/−] *Trp53*[−/−] breast cancer, although the precise mechanism for tumour suppression was not elucidated[19,22].

To gain further insights into the mechanism of MRE11-dependent cell cycle regulation during oncogenic transformation, we performed time-lapse microscopy experiments. We co-transduced *Rosa26*[LSL-Myc/LSL-cas9];*Trp53*[fl/fl] primary mouse mammary epithelial cells (pMMECs) with Cre-sgRNA (sgControl compared with sg*Mre11*, GFP-tagged) and PCNA–mCherry to track cell cycle state transitions in individual pMMECs after inducing MYC overexpression and p53 deficiency[23] (that is, *Myc*[OE]*Trp53*[−/−] pMMECs; Fig. 2a). Cell fate analyses revealed a significant reduction in the rate of G2 arrest and post-mitotic arrest in sg*Mre11* compared with sgControl *Myc*[OE]*Trp53*[−/−] pMMECs (Fig. 2b). Although the role of MRE11 in the ATM-dependent G2/M checkpoint was anticipated[21], the substantial effect of MRE11 on the post-mitotic cell fate of *Myc*[OE]*Trp53*[−/−] pMMECs was unexpected. We validated these findings with a 24-h EdU pulse−chase assay, which demonstrated higher rates of quiescence (EdU negativity) in sgControl *Myc*[OE]*Trp53*[−/−] pMMECs than in sg*Mre11* cells (Fig. 2c). Notably, cell cycle exit in *Myc*[OE]*Trp53*[−/−] pMMECs was associated with an accumulation of micronuclei and cGAS foci, which were less prevalent in *Mre11* mutant cells (Fig. 2d,e). Moreover, *Mre11* mutant *Myc*[OE]*Trp53*[−/−] pMMECs exhibited reduced expression of interferon-stimulated genes (ISGs), such as *Ifit1* and *Ifnb1*, compared with sgControl *Myc*[OE]*Trp53*[−/−] pMMECs (Fig. 2f). Notably, although MRE11 is required for ATM activation[21], sg*Atm* did not phenocopy sg*Mre11* in terms of quiescence and ISG induction in *Myc*[OE]*Trp53*[−/−] pMMECs, which indicated that cGAS activation by MRE11 is independent of ATM (Extended Data Fig. 2a−d).

Although cGAS activation has a known role in DNA damage-induced senescence in p53 wild-type (WT) cells, its role in p53-independent cell cycle exit is not well-established[1,2,13]. In *Myc*[OE]*Trp53*[−/−] pMMECs, cGAS-positive foci were more common in quiescent (EdU-negative) cells, which suggested that cGAS may be directly regulating p53-independent quiescence programs (Fig. 2g). To investigate whether cGAS−STING activation counteracts proliferation, we treated *Myc*[OE]*Trp53*[−/−] pMMECs with 2′3′-cGAMP (a STING agonist) or C-176 (a STING inhibitor). Notably, C-176 reduced the rate of cell cycle exit in control *Myc*[OE]*Trp53*[−/−] pMMECs, whereas 2′3′-cGAMP increased it in *Mre11* mutant *Myc*[OE]*Trp53*[−/−] pMMECs (Fig. 2h). Furthermore, 2′3′-cGAMP stimulated *Ifit1* expression in sg*Mre11* and sg*Gas* treated *Myc*[OE]*Trp53*[−/−] pMMECs, which indicated that MRE11 and cGAS function upstream of 2′3′-cGAMP-dependent STING activation (Fig. 2i). CRISPR targeting of *cGas* and *Sting* increased the proliferation of *Myc*[OE]*Trp53*[−/−] pMMECs to a comparable level as *Mre11* mutation (Fig. 2j,k). These findings demonstrate that MRE11 promotes cell cycle exit through cGAS−STING activation in *Myc*[OE]*Trp53*[−/−] preneoplastic mouse mammary epithelial cells.

We next investigated the relationship between MRE11 and cGAS activation in response to other sources of cytosolic DNA and in different cell line models. MRE11 was required for cGAS foci formation, 2′3′-cGAMP production, STING pathway activation and ISG expression after transfection with either interferon stimulating dsDNA 90 bp (ISD90) or nucleosomal core particles (NCPs) in mouse embryonic fibroblasts (MEFs), in human MDA-MB-231 cells and in BJ-5ta human immortalized fibroblasts (Fig. 3a−d and Extended Data Fig. 3a−l). Time-lapse microscopy demonstrated a failure to generate phase-separated cGAS aggregates onto cytosolic DNA foci in *MRE11* mutant MDA-MB-231 cells (Extended Data Fig. 4). A similar requirement for MRE11 to stimulate cGAS foci formation and ISG expression was observed after ionizing radiation (Extended Data Fig. 5a−d). Because MRE11 can directly suppress replication-associated DNA damage, which might indirectly affect cGAS activation, we irradiated cells in mitosis and again observed an essential role for MRE11 in the subsequent accumulation of cGAS-positive micronuclei (Fig. 3e and Extended Data Fig. 5e). Notably, previous work has indicated that abrogation of the G2/M checkpoint through ATM inhibition enhances cGAS activation[24]. Therefore, reduced cGAS activation after *MRE11* hypomorphism was unexpected and probably independent of its effects on ATM inhibition.

MRE11 is a component of the MRE11–RAD50–NBN (MRN) complex, which is a pleiotropic DSB sensor that mediates both DDR signalling and DNA repair through direct binding to DSB ends, protein interaction domains and nuclease activity[21]. Knockdown of *Rad50* or *Nbn* impeded the recruitment of cGAS to cytoplasmic DNA, phenocopying MRE11 inhibition, implicating the broader MRN complex in cGAS activation (Extended Data Fig. 6a,b). However, the nuclease activity of MRE11 was dispensable for the localization of cGAS to cytosolic DNA and downstream STING pathway activation, as it remained unaffected by treatment with the MRE11 nuclease inhibitors mirin, PFM01 or PFM039 (Fig. 3f and Extended Data Fig. 6c−e). Furthermore, the impaired recruitment of cGAS to cytosolic DNA observed in *MRE11* mutant MDA-MB-231 cells could be rescued through the stable expression of full-length MRE11 or a nuclease-deficient MRE11 variant, but not by a *MRE11* mutant lacking DNA-binding domains (Fig. 3g and Extended Data Fig. 6f−g). Owing to the observed requirement for MRN DNA binding, we investigated its potential colocalization with cGAS at sites of cytosolic dsDNA. We observed frequent colocalization of MRE11 and cGAS after transfection with cytosolic NCPs, which raised the possibility that MRN may have a direct role in facilitating cGAS activation (Extended Data Fig. 6h).

Previous studies have shown that nucleosomes chronically inhibit cGAS activation through high-affinity interactions between cGAS and the histone H2A-H2B AP surface[4–10,17]. We postulated that MRN binding to NCP fragments may modulate these interactions with cGAS. Electrophoretic mobility shift assays demonstrated that MRN binds to NCPs. Moreover, this binding is only partially dependent on the histone H2A-H2B AP region, a result consistent with the known binding of MRN to dsDNA ends (Extended Data Fig. 7a−c). In the presence of cGAS, MRN appeared to form super-shifted complexes that contain

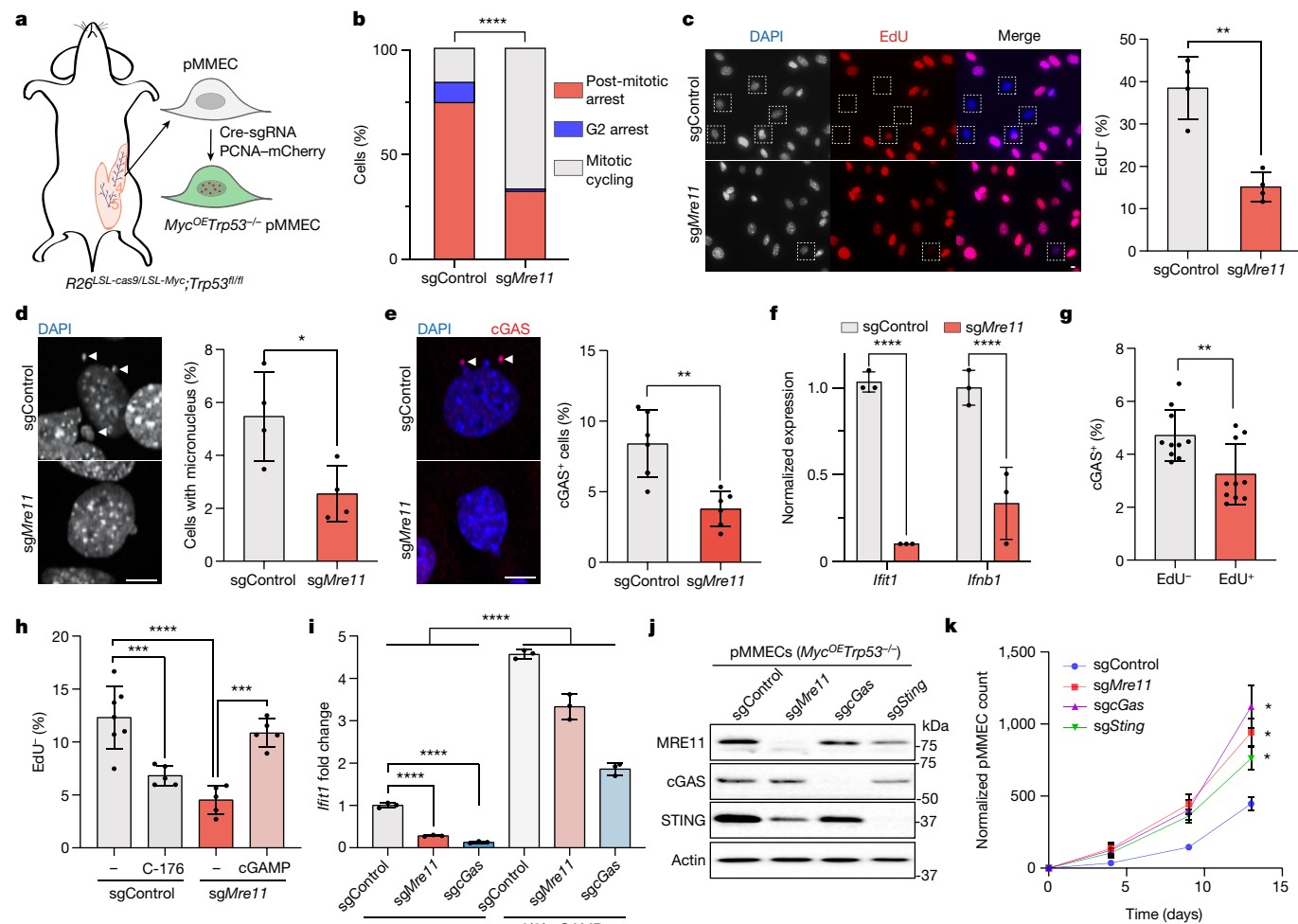

**Fig. 2 | MRE11 promotes post-mitotic arrest through cGAS–STING activation.**
**a**, Schematic of the generation of PCNA–mCherry-labelled $Myc^{OE}Trp53^{-/-}$ pMMECs for time-lapse microscopy. **b**, Cell fate analyses of sgControl ($n = 135$ independent cells) versus sg$Mre11$ ($n = 117$ independent cells) pMMECs by time-lapse microscopy. sgControl versus sg$Mre11$, $P < 0.0001$, two-tailed chi-squared test. **c**, Left, representative images of 24-h EdU pulse quiescence assays, with dashed squares indicating EdU-negative (that is, quiescent) nuclei. Right, percentage of EdU-negative cells in sgControl and sg$Mre11$ pMMECs. Each data point represents $n = 300$ independent cells, $P = 0.0012$. Representative of three independent biological experiments. **d,e**, Images (left) and percentage (right) of cells with micronuclei (**d**) or cGAS foci (**e**), annotated with arrowheads. $n = 440$ independent sgControl and sg$Mre11$ cells, $P = 0.0256$ (**d**) and $n = 1,800$ independent sgControl and sg$Mre11$ cells, $P = 0.0018$ (**e**). Representative of three independent biological experiments. **f**, Relative mRNA levels of ISGs $Ifit1$ and $Ifnb1$, normalized to $Actb$, measured by quantitative PCR with reverse transcription. Graph shows three biological replicates, $P < 0.0001$. Representative of three independent biological experiments. **g**, Percentage of EdU$^-$ and EdU$^+$ sgControl pMMECs with cGAS foci. $n = 338$ (EdU$^-$) and $n = 1,971$ (EdU$^+$) independent cells, $P = 0.0061$. Representative of three independent biological experiments. **h**, Percentage of quiescent (EdU$^-$) sgControl pMMECs ($n = 797$ independent cells) with or without 24 h of C-176 (STING antagonist, $n = 633$ independent cells) treatment, and sg$Mre11$ pMMECs ($n = 646$ independent cells) with or without 24 h of 2′3′-cGAMP ($n = 580$ independent cells) treatment. sgControl versus sgControl + C-176, $P = 0.0009$; sgControl versus sg$Mre11$, $P < 0.0001$; sg$Mre11$ versus sg$Mre11$ + cGAMP, $P = 0.0004$. Representative of three independent biological experiments. **i**, $Ifit1$ expression is induced by treatment of sgControl, sg$Mre11$ and sg$cGas$ $Myc^{OE}Trp53^{-/-}$ pMMECs with 2′3′-cGAMP. Data show mean ± s.e.m. for $n = 3$ biological replicates; ****$P < 0.0001$. Representative of three independent biological experiments. Grouped analyses performed with a two-tailed $t$-test (**c–i**). *$P < 0.05$, **$P < 0.01$, ***$P < 0.001$, ****$P < 0.0001$. **j**, Western blot confirmation of MRE11, cGAS and STING CRISPR targeting in $Myc^{OE}Trp53^{-/-}$ pMMECs. **k**, Normalized pMMEC counts over time. Data show mean ± s.e.m. for $n = 4$ independent biological replicates for each genotype and time point. Representative of three independent biological experiments. Grouped analyses performed with a two-tailed $t$-test. *$P < 0.0001$. Scale bars, 10 µm (**c,d,e**).

both DNA and cGAS (Extended Data Fig. 7d). Notably, at reduced cGAS concentrations, at which both 1:1 and 2:1 molar ratio complexes of cGAS and NCP are observed, MRN seemed to preferentially interact with the 1:1 complex (Extended Data Fig. 7e,f). Given the fact that NCPs possess two cGAS binding sites[4], these observations raised the possibility that MRN binding to NCPs may be incompatible with binding to both cGAS molecules, which potentially contributes to cGAS displacement and its subsequent activation by dsDNA.

To directly test this hypothesis, we developed a time-resolved fluorescence resonance energy transfer (TR-FRET) competition assay to measure the interaction between cGAS and NCPs[25]. Consistent with the established AP-dependent cGAS–nucleosome interaction, TR-FRET signals could be attenuated by as little as 10 nM of unlabelled WT NCPs, whereas >1 µM AP-mutant NCPs was required to interfere with the TR-FRET signal (Fig. 3h). Titration of MRN also reduced the TR-FRET signal, but unlike WT NCPs, plateaued at approximately 50% reduction, which indicated the displacement of one out of two bound cGAS molecules per nucleosome. Human cGAS also promotes nucleosome stacking, which we assessed using a modified TR-FRET assay (Extended Data Fig. 7g). MRN titration significantly disrupted nucleosome stacking, even at concentrations as low as 50 nM (Fig. 3i). Collectively, these observations demonstrate that the DNA-binding activities of MRN

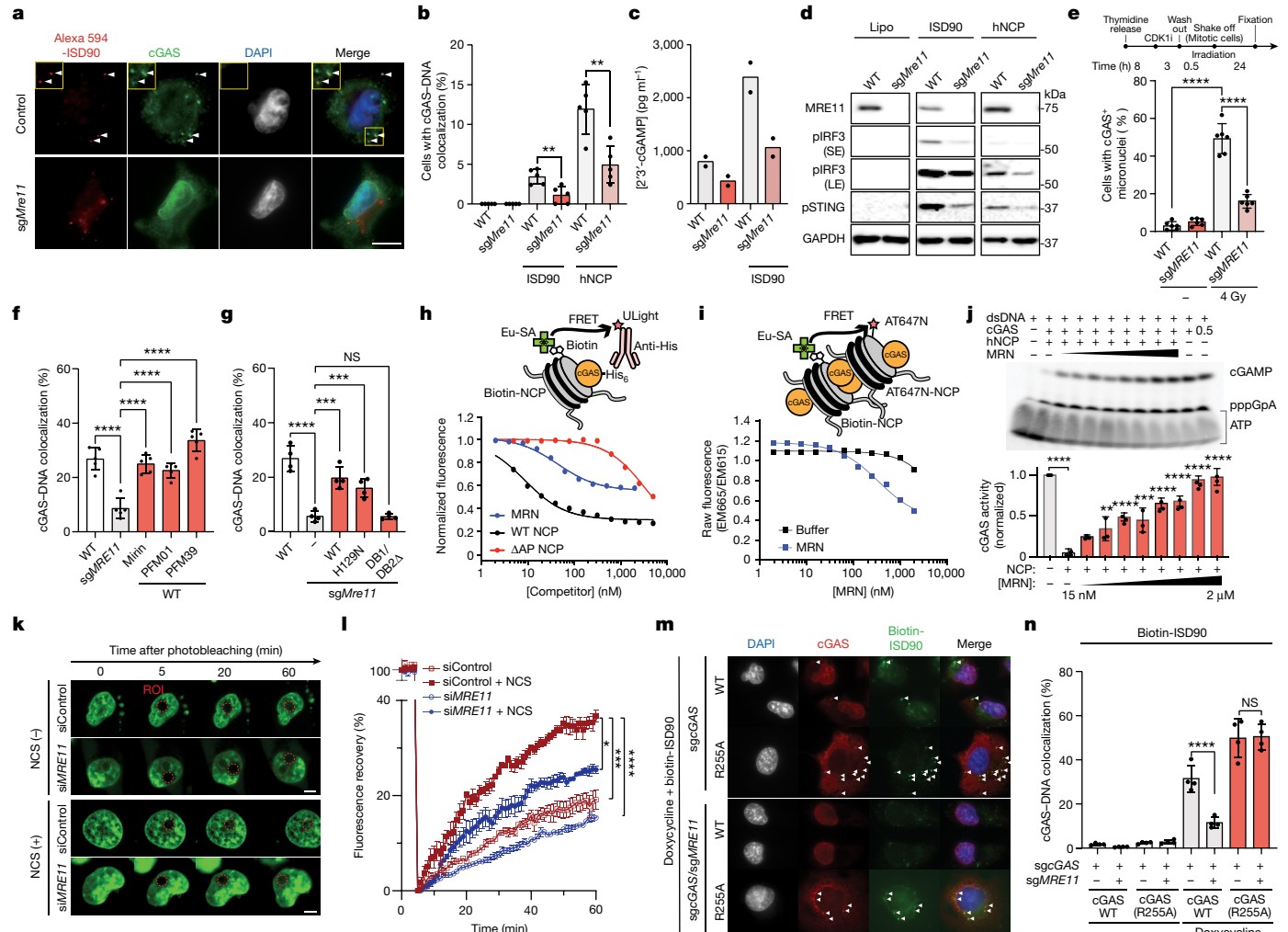

**Fig. 3 | MRN stimulates cGAS activation by antagonizing nucleosome sequestration. a,b,** Images (**a**) and quantification (**b**) of cGAS and ISD90, or cGAS and human nucleosomal core particles (hNCP) colocalization in control cells and sg*MRE11* MDA-MB-231 cells. Left to right, *n* = 344, 312, 491, 405, 366 and 243 independent cells. WT versus sg*MRE11*, ISD90, *P* = 0.007; hNCP, *P* = 0.0038. **c,** 2′3′-cGAMP ELISA after ISD90 transfection in sgControl versus sg*MRE11* cells. Two independent biological replicates for WT, sg*MRE11*, WT + ISD90 and sg*MRE11* + ISD90. CDK1i refers to the CDK1 inhibitor (Ro-3306). Representative of two independent biological experiments. **d,** Top, schematic of the experiment. Bottom, western blot for the STING pathway in WT versus sg*MRE11* cells after Lipofectamine (Lipo) only, ISD90 or hNCP transfection. **e,** cGAS⁺ micronuclei after 24 h of irradiation in WT or sg*MRE11* mitotic cells. *n* = 404 (WT), 441 (sg*MRE11*), 419 (WT + 4 Gy) and 438 (sg*MRE11* + 4 Gy) independent cells. Representative of two independent biological experiments. **f,g,** cGAS–ISD90 colocalization. **f,** In the presence of MRE11 nuclease inhibitors, *n* = 500 independent cells per cell line per condition, ****P* < 0.0001, representative of two independent biological experiments. **g,** sg*MRE11* cells expressing different MRE11 constructs. *n* = 1,200 independent cells per cell line; WT versus sg*MRE11*, *P* = 0.0001; sg*MRE11* versus sg*MRE11* + WT, *P* = 0.0002; sg*MRE11* versus sg*MRE11* + H129N, *P* = 0.0004; sg*MRE11* versus sg*MRE11* + DB1/DB2Δ, *P* = NS. Representative of two independent biological experiments. **h,i,** Schematic (top) and quantification (bottom) of TR-FRET fluorescence assays using europium–streptavidin (Eu–SA)-labelled hNCP and an anti-His antibody conjugated to the

FRET acceptor ULight to measure cGAS–NCP interaction (**h**) and FRET acceptor AT647N conjugated hNCP to measure nucleosome stacking (**i**). Results representative of three independent experiments. **j,** Top, cGAMP accumulation in the presence of 1 μM cGAS, 5 μM dsDNA, 0.5 μM NCP and a 2-fold gradient of MRN concentrations (0.0156–2 μM). Bottom, quantified cGAS activity from triplicate experiments. Left to right, including comparisons to +NCP/ −MRN: *P* < 0.0001, *P* = 0.0693 (NS), *P* = 0.0042, *P* < 0.0001, *P* = 0.0002, *P* < 0.0001, *P* < 0.0001, *P* < 0.0001 and *P* < 0.0001, by two-tailed one-way ANOVA. Representative of three independent experiments. **k,l,** Images (**k**) and quantification (**l**) FRAP assays in sgControl versus si*MRE11* cells stably expressing GFP–cGAS. Images taken at 1-min intervals before and after photobleaching an arbitrary nuclear region of interest (ROI) for 60 min. Neocarzinostatin (NCS; 0.1 mg ml⁻¹) was added immediately after photobleaching as indicated. *n* = 3 independent cells for each condition. Two-tailed two-way ANOVA: siControl + NCS versus si*MRE11* + NCS, *P* = 0.041; siControl versus siControl + NCS, *P* = 0.0003; siControl + NCS versus si*MRE11*, *P* < 0.0001. **m,n,** Images (**m**) and quantification (**n**) of colocalization of cGAS-ISD90 in cGAS-deficient control or sg*MRE11* cells reconstituted with cGAS WT or cGAS(R255A). White arrowheads indicate colocalization of cGAS and ISD90 foci. Data are mean ± s.e.m., *n* = 1,200 independent cells per condition. sg*cGAS* + cGAS WT + doxycycline versus sg*cGAS*/sg*MRE11* + cGAS WT + doxycycline, *P* < 0.0001; sg*cGAS* + cGAS(R255A) + doxycycline versus sg*cGAS*/sg*MRE11* + cGAS(R255A) + doxycycline, *P* > 0.99 (NS). Unless otherwise specified, grouped analyses performed with a two-tailed *t*-test. Scale bars, 10 μm (**k,m**), 20 μm (**a**).

lead to disruption of cGAS-dependent nucleosome stacking and a 50% displacement of cGAS from the nucleosome AP surface.

Next, we examined whether the partial release of cGAS from the nucleosome AP surface enables cGAS activation by dsDNA. To

that end, we used a previously established biochemical assay for cGAS 2′3′-cGAMP synthase activity, which is potently inhibited by NCPs[4]. At a 2:1 molar ratio of cGAS and NCPs, the addition of MRN resulted in a concentration-dependent restoration of 2′3′-cGAMP

synthesis, with stimulatory effects observed at MRN concentrations as low as 15 nM (Fig. 3j). Similar effects were observed when using a nuclease-deficient MRN complex (Extended Data Fig. 7h). Notably, the stimulatory effect of MRN on cGAS activity in the presence of NCPs was not observed at a 1:1 molar ratio (cGAS to NCPs) (Extended Data Fig. 7i,j). These findings align with a molecular displacement model wherein MRN interaction with a 2:1 ratio of cGAS to NCP complexes liberates approximately 50% of the cGAS molecules for activation by dsDNA.

Based on these biochemical findings, we postulated that MRN may be required for the displacement of cGAS from inhibitory nucleosome AP binding after DNA damage. To test this, we performed fluorescence recovery after photobleaching (FRAP) experiments in MDA-MB-231 cells stably expressing GFP–cGAS after transfection with control siRNA (siControl) or siRNA against *MRE11* (si*MRE11*). In siControl cells, we observed about 20% fluorescence recovery of GFP–cGAS after 60 min, which indicated a relatively low rate of cGAS mobilization from non-photobleached chromatin-binding sites (Fig. 3k–l). Notably, DNA damage induced by neocarzinostatin significantly increased GFP–cGAS fluorescence recovery in siControl cells. By contrast, si*MRE11* cells had a significantly slower rate of fluorescence recovery, particularly after DNA damage induction, which suggested that MRE11 has a crucial role in DNA damaged-induced release of cGAS from nucleosome sequestration. Additionally, MRE11 deficiency impaired cGAS relocalization from the nucleus to the cytoplasm after cytosolic DNA transfection and DNA damage induction (Extended Data Fig. 8a–d).

To further investigate whether the release of cGAS from nucleosome sequestration is the primary role of MRN in promoting cGAS activation by dsDNA, we generated cGAS-deficient MDA-MB-231 cells that expressed either WT or mutant *MRE11*. Direct cytoplasmic transfection of recombinant cGAS protein restored its recruitment to NCPs and its ability to activate STING signalling, even in *MRE11* mutant cells (Extended Data Fig. 9a,b). We also genetically reconstituted cGAS-deficient cells with either cGAS WT or cGAS(R255A), the latter of which disrupts its binding to the histone AP surface, thereby abolishing nucleosome sequestration (Extended Data Fig. 9c). Whereas cGAS WT cells required MRE11 for colocalization of cGAS to cytosolic DNA, cGAS(R255A) cells demonstrated cGAS recruitment to cytosolic DNA irrespective of MRE11 status (Fig. 3m,n and Extended Data Fig. 9d,e). These findings establish the binding of MRN to DNA damage as a crucial mechanism that liberates cGAS from nucleosome sequestration, which enables its activation by dsDNA.

To gain insights into the functional consequences of MRE11-mediated cGAS activation during mammary tumorigenesis, we performed single-cell RNA sequencing (scRNA-seq) of *Myc^OE^Trp53^−/−^* pMMECs with control or *Mre11* mutant (sg*Mre11*) genotypes. Uniform manifold approximation and projection (UMAP) analyses revealed a region where sgControl cells were more abundant than sg*Mre11* cells (Fig. 4a, left). Overlaying the UMAP plots with Seurat classification of cell cycle stages demonstrated that sgControl cells were enriched in G2/M and G1 phases of the cell cycle, whereas sg*Mre11* cells were enriched in S phase (Fig. 4a, right). Differential gene expression analyses of control versus *Mre11* mutant *Myc^OE^Trp53^−/−^* pMMECs in G1 phase revealed numerous inflammatory pathway genes that were upregulated in control pMMECs, including *Isg15*, *Il1rl1*, *Ifit1* and *Zbp1* (Fig. 4b).

*Zbp1* is an ISG that binds viral and cellular Z-RNA or Z-DNA, which triggers RIPK3-dependent and MLKL-dependent necroptosis[26,27]. We observed accumulation of Z-RNA and Z-DNA during neoplastic transformation driven by p53 deficiency and MYC overexpression in pMMECs, along with evidence of necroptosis activation through induction of ZBP1 and phosphorylated MLKL (pMLKL, S345) (Extended Data Fig. 10a–c). In comparison to sgControl *Myc^OE^Trp53^−/−^* pMMECs, *Mre11* mutant (sg*Mre11*) *Myc^OE^Trp53^−/−^* pMMECs exhibited reduced levels of ZBP1 and pMLKL (Fig. 4c–e). Disruption of cGAS, ZBP1 and RIPK3 also

decreased levels of the necroptosis marker pMLKL in *Myc^OE^Trp53^−/−^* pMMECs (Fig. 4e). Treatment with 2′3′-cGAMP restored ZBP1 and pMLKL levels in sg*Mre11* and sg*cGas Myc^OE^Trp53^−/−^* pMMECs, a result consistent with necroptosis activation as a downstream effector of MRE11–cGAS–STING activation during mammary cell transformation (Extended Data Fig. 4d,e). CRISPR-mediated targeting of *Zbp1* in *Myc^OE^Trp53^−/−^* pMMECs reduced rates of cell cycle exit and increased tumorigenic potential, phenocopying genetic inactivation of *Mre11* or *cGas* (Fig. 4f,g and Extended Data Fig. 4f). Collectively, these findings reveal that engagement of MRE11–cGAS during MYC-induced and p53-deficient mammary neoplasia stimulates ZBP1–RIPK3–MLKL-dependent necroptosis and tumour suppression.

Necroptosis activation is associated with the secretion of MLKL-positive extracellular vesicles that trigger macrophage phagocytosis, thereby promoting inflammation[28]. Consistently, we found that supernatant from preneoplastic sgControl *Myc^OE^Trp53^−/−^* pMMECs induced an interferon reporter gene in macrophages, whereas supernatant from sg*Mre11* and sg*Mlkl Myc^OE^Trp53^−/−^* pMMECs did not (Fig. 4h). Accordingly, we postulated that the immune suppression and DNA damage tolerance observed in triple-negative breast cancer (TNBC) might be linked to the suppression of DNA damage-induced necroptosis. Indeed, TNBC with reduced expression of *ZBP1* in TCGA dataset[18] exhibited high levels of copy number aberrations and low expression of T cell and inflammatory gene signatures (Fig. 4i). Furthermore, low *ZBP1* expression in TNBC correlated with worse overall survival in the SCAN-B and METABRIC cohorts[29,30] (Fig. 4j). These findings provide support for a key role for ZBP1-dependent necroptosis in maintaining genome integrity and immune signalling in TNBC, which is significantly correlated with patient survival.

## Discussion

Clinically aggressive cancers often exhibit tolerance to chronically increased DNA damage, which suggests that there are perturbations in physiological DDR pathways. Through our in vivo CRISPR screen targeting DDR genes in a MYC-induced and p53-deficient breast cancer model, we discovered an unexpected role for MRE11 as a direct stimulator of cGAS-dependent innate immune activation by oncogene-induced DNA damage. Disruption of MRE11 function resulted in DNA damage tolerance, immune suppression and accelerated tumorigenesis. This new function for MRE11 in facilitating cGAS activation by dsDNA is independent of both its nuclease activity and ATM signalling. Binding of the MRN complex to nucleosomal fragments releases cGAS from nucleosomal AP surfaces, which enables cGAS mobilization, cytoplasmic relocalization and activation by dsDNA (Fig. 4k). These MRE11-dependent effects may be particularly relevant within micronuclei, where chromosomal fragments extruded from the primary nucleus undergo histone modifications and fragmentation that often results in cGAS activation despite the presence of histone-bound DNA fragments[31–34]. Previous studies have also reported a role for the MRE11 nuclease in replication fork degradation when other DNA repair factors are deficient, which leads to accumulation of cytosolic DNA substrates that trigger cGAS activation[35,36]. Collectively, these findings establish MRE11 as an important mediator of cGAS activation in response to diverse forms of DNA damage.

Our study also highlighted the significance of ZBP1-dependent necroptosis as an effector of MRE11–cGAS-mediated tumour suppression (Fig. 4k). These findings align with recent studies demonstrating the crucial role of ZBP1 as a downstream effector of cGAS–STING signalling in mediating doxorubicin-associated cardiotoxicity and telomeric stress-associated tumour suppression[37,38]. We propose that MRE11–cGAS is required to activate a DNA damage-induced necroptosis programme that exerts p53-independent tumour

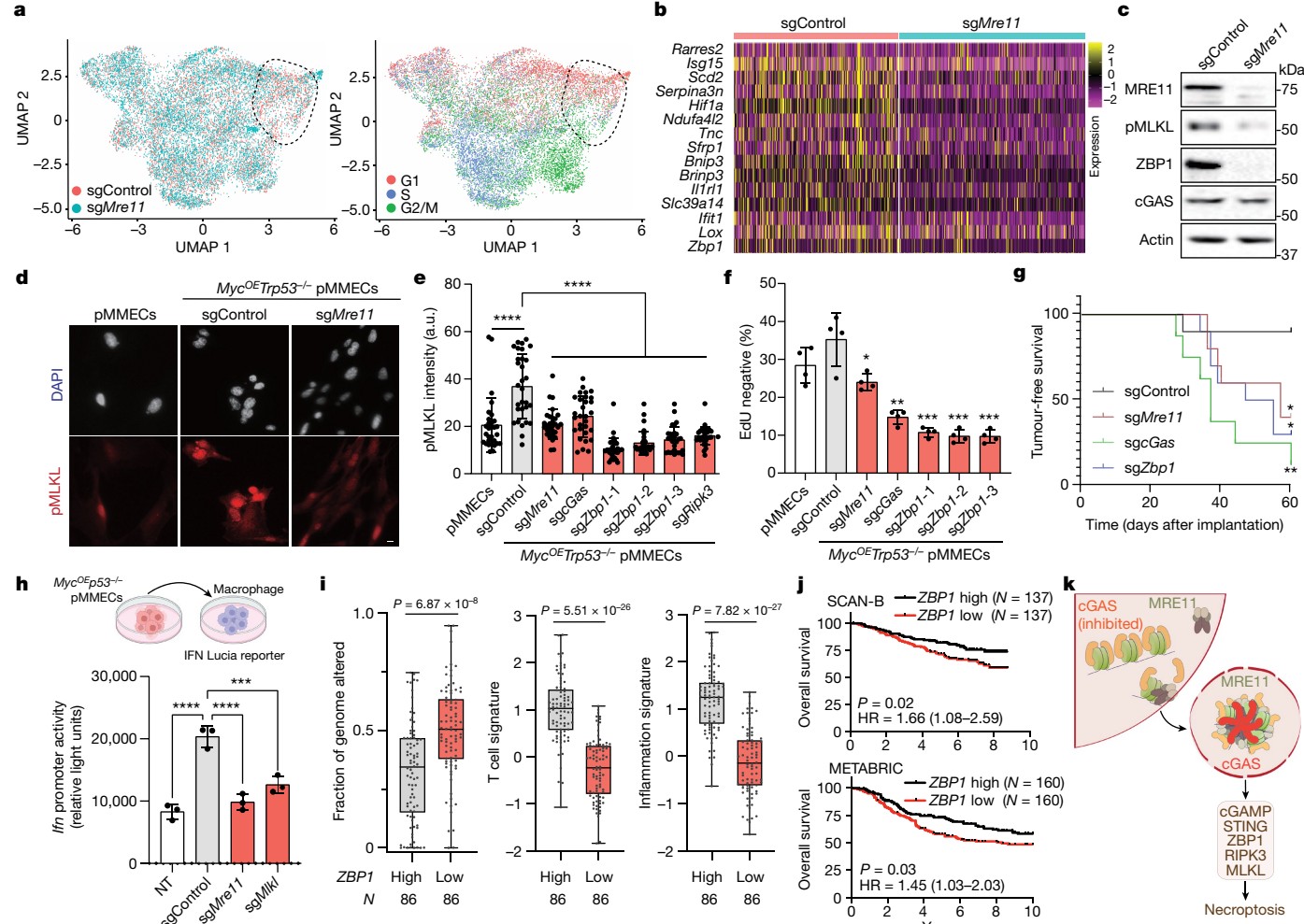

**Fig. 4 | MRE11–cGAS–STING stimulates ZBP1–RIPK3–MLKL-dependent necroptosis. a**, scRNA-seq UMAP analyses of sgControl versus sg*Mre11* *Myc^OE^Trp53^−/−* pMMECs. The UMAP on the right is pseudocoloured according to the predicted cell cycle state. A ROI with an increased abundance of sgControl compared with sg*Mre11* pMMECs is outlined. **b**, Heatmap showing gene expression in sgControl G1 compared with sg*Mre11* G1 single cells. **c**, Western blot of ZBP1 and pMLKL in sgControl and sg*Mre11* pMMECs, representative across three independent experiments. **d**, Immunocytochemistry analysis of pMLKL in pMMECs: untransduced, sgControl and sg*Mre11*. **e**, Quantification of pMLKL fluorescence intensity across genotypes in arbitrary units (a.u.). Data are mean ± s.e.m., *n* = 30 cells for each cell line, representative of 3 independent biological experiments. Statistical comparisons to sgControl by two-tailed t-test: ****P < 0.0001. Scale bar, 10 μm. **f**, Twenty-four hour EdU⁻ percentage (mean ± s.e.m.) in the indicated pMMEC genotypes. Number of independent cells analysed per condition: 401 (pMMECs), 400 (sgControl, sg*Mre11* and sg*cGas*, sg*Zbp1*-1 and sg*Zbp1*-2), 390 (sg*Zbp1*-3) and 398 (sg*Ripk3*). Representative of three independent biological experiments. Statistical comparisons by two-tailed t-test to the sgControl genotype: sg*Mre11*, *P* = 0.0224; sg*cGas*, *P* = 0.0014; sg*Zbp1*-1, *P* = 0.0005; sg*Zbp1*-2, *P* = 0.0004; and sg*Zbp1*-3, *P* = 0.0004. **g**, Tumour-free survival after orthotopic transplantation of pMMECs expressing the

indicated sgRNAs into the fourth mammary fat pad of female *NOD/RAG1* mice, *n* = 10 independent animals per group. Two-tailed log-rank (Mantel–Cox) tests compared with sgControl are shown. *P* = 0.0336 (sg*Mre11*); **P* = 0.0014 (sg*cGas*); *P* = 0.0126 (sg*Zbp1*). **h**, Schematic (top) and quantification (bottom) of IFN Lucia reporter activity in RAW 264.7 macrophages after treatment with supernatant from control pMMECs (NT) or pMMECs expressing the indicated sgRNAs. Graph depicts mean ± s.e.m., *n* = 3 (NT, sgControl, sg*Mre11*, sg*Mlkl*) samples in 3 independent biological experiments. Comparisons by two-tailed t-tests, NT versus sgControl, *P* < 0.0001; sgControl versus sg*Mre11*, *P* < 0.0001; sgControl versus sg*Mlkl*, *P* = 0.0007. **i**, Fraction of genome altered (left), T cell signature (middle) and inflammation signature (right) levels in TCGA TNBC cohort stratified by median-thresholded expression of *ZBP1*. Box plots display medians, interquartile range, and minimum and maximum values. Statistical comparisons by two-tailed t-tests. **j**, Kaplan–Meier overall survival analysis of TNBC with *ZBP1* high versus low expression in the SCAN-B and METABRIC cohorts, using a two-tailed log-rank test. HR, hazard ratio. **k**, MRE11-mediated activation of cGAS and ZBP1-dependent necroptosis during tumorigenesis. Schematic in **h** was created using BioRender (www.biorender.com).

suppression. Low *ZBP1* expression may serve as a promising biomarker for identifying functional deficits in this pathway. TNBCs with low *ZBP1* expression exhibit increased genome instability, immune suppression and worse clinical outcomes. Based on these observations, it is plausible that cancer-specific disruptions in MRN, cGAS, STING and/or ZBP1 may influence responses to DNA damaging therapy and immunotherapy, warranting further exploration of their clinical implications.

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

# Methods

## Cell culture

**Culturing of cell lines.** MDA-MB-231 (ATCC, CRM-HTB-26), HEK 293T/17 (ATCC, CRL-11268) and BJ-5ta (ATCC, CRL-4001) cells were obtained from the American Type Culture Collection (ATCC) and were cultured according to the manufacturer's specifications HEK 293T/17 cells were maintained in Dulbecco's modified Eagle medium (DMEM, Corning, 10-013-CV) supplemented with 10% FBS (VWR, 76294-120). Triple-negative MDA-MB-231 cells were cultured in minimum essential medium (MEM) (Gibco, 11095-080) supplemented with 1% sodium pyruvate (VWR, 97061-448) and 10% FBS. The hTERT-immortalized human dermal fibroblast BJ-5ta cells were cultured in DMEM supplemented with 10% FBS and 0.01 mg ml$^{-1}$ hygromycin B (Sigma-Aldrich, H7772). RAW-Lucia ISG cells (Invivogen, rawl-isg) were grown in DMEM with 10% FBS and 100 μg ml$^{-1}$ Normocin (Invivogen, ant-nr-05). At every other passage, 200 μg ml$^{-1}$ of Zeocin (VWR, AAJ67140-XF) was added for selection. All cells were cultured at 37 °C in a humidified incubator with 5% $CO_2$ in the air and were tested monthly for mycoplasma using a PlasmoTest kit (Invivogen, rep-pt1).

**Cell line identification.** In accordance with the guidelines set by the International Cell Line Authentication Committee, we verified the identity of all cell lines used in this study. The cell lines were tested by Genetica DNA Laboratories. For human cell lines, the PowerPlex16HS method was used, which includes a mouse marker for the detection of mouse DNA. For mouse cell lines, the mouse STR method was adopted, incorporating two human markers for detecting human DNA. The results from these tests confirmed the identity of each cell line, ensuring their authenticity. The last authentication test was conducted on 10 August 2023.

**Collection of pMMECs.** pMMECs were derived by collecting the fourth and fifth mammary glands from 8–12-week-old virgin female transgenic mice with the desired genotype. pMMECs were isolated without performing randomization. Glands were incubated in Liberase digestion medium ((EpiCult-B Mouse Medium kit, StemCell Technologies, 05610), 150 U ml$^{-1}$ collagenase type 3 (Worthington, LS004182), 20 mM HEPES (Thermo Fisher Scientific, 15630106) and 20 μg ml$^{-1}$ Liberase Blendzyme 2 (Roche, 11988425001)) and shaken vertically at 37 °C overnight. The resulting digestion was spun down and resuspended in trypsin (Gibco, 25200056) with DNase I (Worthington, LS002060) and incubated at 37 °C for 5 min. DMEM with 5% FBS was added to neutralize the trypsin. Cells were spun down and resuspended in dispase (Stem Cell Technologies, 07913) and DNase I and incubated at 37 °C for 5 min. Cells were washed twice with DMEM, and the resulting cells were resuspended in Epi-Cult medium and seeded onto plates coated with Cultrex 3D culture matrix rat collagen I (Fisher Scientific, 3447-020-01).

**Collecting tumour-derived pMMECs.** For mammary tumour induction, 6–12-week-old virgin female $R26^{cas9/cas9}$, $Myc^{OE}$ and $Trp53^{fl/fl}$ mice received bilateral intraductal injections into the fourth mammary gland containing $5 \times 10^5$ TU of the indicated LentiCRISPR-Cre-V2-sgRNA lentivirus. Mouse cohorts were palpated for the development of mammary tumours. The mice were regularly examined twice a week for the development of mammary tumours. After the tumours had formed, they were examined three times a week. The mice were euthanized in a humane manner in accordance with the guidelines set by the UNC Institutional Animal Care and Use Committee (IACUC) when they reached a predetermined experimental end point. The mammary tumours were then collected during necropsy and divided into four pieces for further analysis. Tumour pMMECs were isolated without performing randomization. One of the tumour samples was incubated in digestion medium (DMEM with 10% FBS, 1 mg ml$^{-1}$ collagenase type 3 and 1 mg ml$^{-1}$ hyaluronidase (Worthington, LS005477)) and shaken horizontally at

37 °C for 4 h. The digestion mix was spun down and resuspended with trypsin containing DNase I at 37 °C for 5 min. DMEM was added to neutralize the trypsin. Cells were spun down and incubated in dispase and DNase I at 37 °C for 5 min. Cells were washed with DMEM and passed through a 70 μm filter. Tumour cells were resuspended in HuMEC Ready Medium (1X) (Thermo Fisher Scientific, 12752010) medium.

**MRE11 nuclease inhibition assay.** The MRE11 inhibitors mirin (Sigma-Aldrich, M9948), PFM39 (Sigma-Aldrich, SML1839) and PFM01 (Sigma-Aldrich, SML1735) were administered at a concentration of 100 μM unless otherwise specified. This was followed by the transfection of biotin-conjugated 90-mer dsDNA (biotin-ISD90) at a concentration of 200 pmol, 30 min after application of the MRE11 inhibitor. Cells were then collected for western blotting 3 h after dsDNA transfection. The localization of cGAS and biotin-ISD90 was detected using immunocytochemistry (ICC) 2 h after dsDNA transfection, with biotin-ISD90 detected using streptavidin–Alexa 488 (Thermo Fisher Scientific, S32354). Last, for quantitative PCR (qPCR) analysis, cells were collected 6 h after dsDNA transfection.

## Transgenic mouse models

All animal experimentation was conducted with approval by the UNC IACUC. $R26^{LSL-cas9}$ (JAX, 024857) and $R26^{LSL-Myc^{OE}}$ (JAX, 020458) transgenic mouse strains were obtained from the Jackson Laboratory. $Trp53^{fl/fl}$ mouse strains were provided by the Perou Laboratory and originally obtained from the Frederick National Laboratory for Cancer Research (strain 01XC2). $R26^{cas9/cas9}$, $Myc^{OE}$ and $Trp53^{fl/fl}$ lines were used in the in vivo DDR-CRISPR screen. A subset of interbred mouse strains used in this study were analysed using a Mouse Universal Genotyping Array from Neogen Genomics and determined to be >90% FVB with a minor contribution from C57BL/6J. Blinding of animal experiments was not performed.

**Housing conditions for mice.** At the Division of Comparative Medicine, the standard light cycle for mice is set from 07:00 to 19:00. The ambient temperature for rodents is maintained between 20 and 23 °C, with a humidity level ranging from 30% to 70%. When setting up breeder cages, nesting material was added, and the types of nesting materials that can be used are outlined in the IACUC's Environmental Enrichment Program for Animals. We documented the date of birth of the mice and recorded it on the Division of Comparative Medicine's Weaning Notice within 3 days of their birth. For non-breeding cages, a maximum of five adult mice of the same sex were housed per cage. Pups are typically weaned by the time they are 23 days old, especially when their weight reaches around 10 g. However, if a pup weighs less than 10 g by the 23rd day, exceptions to this guideline are considered.

## In vivo DDR-CRISPR screen

A total of 40 virgin female transgenic mice of $R26^{cas9/cas9}$, $Myc^{OE}$ and $Trp53^{fl/fl}$ lines aged 6–10 weeks were intraductally injected with 10 μl lentivirus generated from the Lenti-CRISPR-Cre-V2-sgRNA DDR Library plasmid at $5 \times 10^5$ TU into the fourth mammary glands bilaterally. Mice were selected for the experiment without randomization. Mice were palpated twice a week for tumour detection. Two mice each were collected at 1, 3, 6, 9 and 12 weeks to assess for mammary hyperplasia and to stain for GFP to assess viral infectivity. Mice with a single tumour were euthanized for collection when the tumour reached a maximum diameter of 2.0 cm. If bilateral tumours were present, the mouse was euthanized when the largest tumour reached a diameter of 1.5 cm. Mice were euthanized in compliance with IACUC protocols. At the time of necropsy, abdominal exploration was performed for gross liver metastases and thoracotomy was performed for gross lung metastases. Splenic tissue was collected and banked for gDNA collection. Each mammary tumour was sectioned into four pieces, with two pieces flash-frozen for RNA and DNA extraction. One piece was fixed

in 4% paraformaldehyde (PFA) for tissue processing and staining. The remaining piece was taken for the creation of the mammary tumour cell line. sgRNA abundance was determined from flash-frozen samples to eliminate effects of tissue culture variances. Sample randomization was not performed.

## Establishment of stable cell lines and viral production and infection

**Generation of cell lines expressing a hypomorphic *Mre11* mutation.** crRNA was designed using MIT CRISPR (crispr.mit.edu) to target codon 633 of *Mre11*, which is the site of a truncating mutation that gives rise to the radiosensitive ataxia-telangiectasia-like disorder. We performed Neon transfection (Invitrogen, MPK1025) with recombinant Cas9 purified by the University of North Carolina Center for Structural Biology Protein Expression and Purification Core), crRNA (Integrated DNA Technologies (IDT), customized) and tracrRNA (IDT, 1072532) using the manufacturer's protocol and electroporation settings. Forty-eight hours after transfection, cells were seeded for single-clone selection. Restriction enzyme screening, western blots, PCR screening and Sanger sequencing confirmed gene targeting and functional assays.

**Halo-conjugated mutant cell lines.** Retroviral pBABE-puromycin-Halo vectors (gift from E. Rothenberg, NYU) encoding the desired construct (15 µg) were co-transfected with pUMVC (10 µg) (Addgene, 8449) and pCMV-VSV-G (2 µg) (Addgene, 8454) into a 10 cm diameter dish of 70% confluent HEK 293T/17 cells to generate retrovirus. In brief, plasmids were mixed with 30 µl Lipofectamine 3000 (Thermo Fisher Scientific, L3000001) in 1.0 ml Opti MEM reduced serum medium (Thermo Fisher Scientific, 31985062) and incubated at room temperature for 20 min. The transfection mix was then added dropwise to a 10 cm dish. After 16 h of transfection, the transfection mix was removed, and 15 ml of fresh medium was added to the cells. Twenty-four hours later, the cell culture medium was collected, centrifuged for 5 min at 1,000*g* and the viral supernatant filtered using 0.45 µm sterile syringe filters. Next, 60% confluent target cells were seeded in a 6-well plate and were infected with 2 ml of the retroviral medium containing 8 µg ml⁻¹ hexadimethrine bromide–polybrene (Sigma-Aldrich, 107689) for 24 h. Medium containing 2 µg ml⁻¹ puromycin (Thermo Fisher Scientific, A1113802) was used for selection of cells that had integrated the constructs. A pool of transduced cells was utilized for subsequent experiments following complete death of non-transduced cells placed under selection in parallel.

**Cells lines for live-cell imaging.** The lentiviral vector pTRIP-CMV-tagRFP-Flag-cGAS (Addgene, 86676) or pTRIP-CMV-GFP-Flag-cGAS (Addgene, 86675) (12 µg) was co-transfected with 6 µg psPAX2 (Addgene, 12260) and 3 µg pMD2.G (Addgene, 12259) to 80% confluent HEK 293T/17 cells in a 10 cm culture dish. In brief, plasmids were mixed with 42 µl polyethylenimine (PEI; 1 mg ml⁻¹ linear PEI, Sigma Aldrich, 765090) and incubated at room temperature for 20 min. The transfection mix was then added dropwise to a 10 cm dish. After 16 h of transfection, the medium was removed and 15 ml of fresh medium was added to the cells. Twenty-four hours later, the cell culture medium was collected for 3 days, centrifuged for 5 min at 1,000*g* and filtered using 0.45 µm sterile syringe filters. Next, 60% confluent target cells were seeded in a 6-well plate and were infected with titred lentiviral medium containing 8 µg ml⁻¹ polybrene for 24 h. Medium was replaced every 24 h for 2 days. Cells were seeded for single-clone selection in a 96 well-plate. Single clones were selected for Sanger sequencing and functional assays for relevant genes.

To generate knockouts of indicated genes in pMMECs, HEK 293T/17 cells were transfected using PEI with the viral packaging plasmids psPax2 and pMD2.G and with LentiCRISPR-Cre-V2-sgRNA-Lumifluor harbouring sgControl, sg*MRE11*, sg*cGAS*, sg*STING*, sg*ZBP1* (1, 2 and 3),

sg*MLKL*, sg*RIPK3* or sg*ATM* plasmids. After 16 h of transfection, medium was removed, and 15 ml of fresh medium was added to the cells. Twenty-four hours after transfection, the cell culture medium was collected for 3 days, centrifuged for 5 min at 1,000*g* and filtered using 0.45 µm sterile syringe filters. Medium containing virus was spun down for 2 h at 16 °C at 21,000 r.p.m. The virus-containing pellet was resuspended in 1:100 the initial volume of PBS and incubated at 4 °C for 24 h then aliquoted and stored at −80 °C until use. Virus aliquots were thawed immediately before use to avoid loss of viral transduction efficiency with repeated freeze–thaw cycles. For lentiviral infections, cells were transduced with the appropriate virus combined with 8 µg ml⁻¹ polybrene overnight. Cells were refed with virus-containing medium and incubated for 2 days. Following the last infection, cells were washed three times with PBS and cultured with HuMec medium. For testing viral efficacy, a small sample of cells were fixed with 3% PFA and were assessed by flow cytometry (Attune NxT, UNC Flow Cytometry Core) for the presence of GFP (indicating Cas9 expression), mCherry (indicating PCNA expression) or were stained with anti-CD2-PE (indicating MYC expression).

To generate cGAS knockout MDA-MB-231 cell lines, synthetic gRNAs targeting human *cGAS* were purchased from Synthego (CRISPR gene knockout kit V2). gRNAs were diluted to 30 µM (pmol µl⁻¹) in 1× TE buffer (Synthego). Cas9 (Synthego) was diluted to 20 µM (pmol µl⁻¹) in 1× TE buffer. These were stored at −80 °C until use. A sgRNA to Cas9 ratio of 3:1 was used for RNP formation. Next, 3 µl (90 pmol) of sgRNA, 0.5 µl of 10 pmol Cas9 and 3.5 µl resuspension buffer were added to a 1.5 ml tube per sample. The RNP complexes were allowed to incubate for at least 10 min at room temperature. Cells were subcultured 2 days before electroporation and seed cells were cultured in an appropriately sized vessel so that they are 70–80% confluent on the day of transfection. Each electroporation reaction used approximately 1 × 10⁵ cells. Next, 1 ml of pre-warmed MEM medium was added into each well of a 12-well plate per reaction. RNP complex (7 µl) was added to 5 µl of MDA-MB-231 cell suspension to make 12 µl of cell–RNP solution per reaction and electroporated using a Neon Transfection system (Thermo-Fisher, MPK5000) at a pulse code (20 ms × 1 pulse) using 10 µl Neon tips (Thermo-Fisher, MPK1025) at various voltages (1,800–2,200 V). In knockout experiments, cell–RNP solutions were electroporated at 20 ms × 4 pulse and 1,400 V. Immediately following electroporation, cells were slowly pipetted into pre-warmed culture medium (MEM supplemented with 1% sodium pyruvate and 10% FBS) in a 12-well plate. Cells were then incubated at 37 °C. Cells were cultured for 3–6 days following electroporation before reading out gene-editing efficiency by flow cytometry or Sanger sequencing.

**Generation of stable doxycycline-inducible human cGAS MDA-MB-231 cell lines.** We used a piggyback system to generate two stable doxycycline-inducible human cGAS MDA-MB-231 cell lines. First, we obtained cGAS WT conjugated with the *attB* sequences by PCR using custom-designed primers (IDT, forward: 5′-GGGGACAAGTTTG TACAAAAAAGCAGGCTTCATGCAGCCTTGGCACGGAAAG-3′; reverse: 5′-GGGGACCACTTTGTACAAGAAAGCTGGGTTTCAAAATTCATCAAAA ACTGG-3′) from a pTRIP-CMV-tagRFP-Flag-cGAS vector. Then, we subcloned it into pDONR 221 (Thermo Fisher Scientific, 12536017) using Gateway BP Clonase II enzyme mix (Thermo Fisher Scientific, 11789100), followed by cloning into the PB-TA-ERP2 (Addgene, 80477) destination vector the LR Clonase II enzyme mix (Thermo Fisher Scientific, 11791100) reaction according to the manufacturer's protocol, resulting in the generation of PB-TA-ERP2-hcGAS-WT, and transformed into competent cells. We then performed site-directed mutagenesis of *cGAS* residues that engage the nucleosomal AP, in which R255 was replaced by alanine (R255A). Primers were designed and ordered (IDT, forward: 5′-ATTTAAAGCAAATCCGAAAGAA-AATCCTCTGA-3′; reverse: 5′-GGATTTGCTTTAAATTTCACAAAGTAATATGCA-3′) and PCRs performed. PCR products were confirmed by gel electrophoresis, and

purified PCR products were incubated with 20 U of Dpn1 (NEB, R0176S) to digest the template DNA. The amplified plasmid was used for transforming competent cells (Lucigen, 60241-2). *cGAS* genes on amplified plasmids (PB-TA-ERP2-hcGAS-R255A) were sequenced (Plasmidsaurus) to confirm that the mutation had been introduced at the correct place and that no additional mutation occurred during the PCR process. Cells were seeded in a 6-well plate at a density of 100,000 cells per well. piggyBac vectors (PB-TA-ERP2-hcGAS-WT or PB-TA-ERP2-hcGAS-R255A) were co-transfected with a piggyBac transposase vector (System Biosciences, PB210PA-1). DNA was transfected at a transposon to transposase ratio of 1 µg:200 ng *cGAS* knockout MDA-MB-231 cell lines using Lipofectamine 3000 following the manufacture's protocol. Three days after transfection, cells were selected with 2 µg ml$^{-1}$ puromycin. For expression assays, cells were induced with doxycycline hyclate (Sigma-Aldrich, D9891) at a final concentration of 1 µg ml$^{-1}$ for 24 h.

## Cloning

**LentiCRISPR-Cre-V2-sgRNA LumiFluor plasmids.** This plasmid was created using restriction enzymes (XbaI and BglII) to cut the Cre sequence from the pLV-Cre_LKO1 plasmid (Addgene, 25997) and swapping it for the Cas9 sequence in lentiCRISPR V2 (Addgene, 52961) through restriction digest and T4 ligation. To remove the BsmbI site within Cre, Gibson cloning was used (HiFi DNA Assembly master mix; NEB, E2611S) to change the sequence of a valine residue from GTC to GTA, thus removing the site while preserving the protein sequence. The Lumifluor construct was cloned using a Lentiviral_pRRL-EF1a-GpNLuc plasmid (gift from A. Amelio) as previously described[39]. Using the remaining BsmB1 sites, our custom DDR-CRISPR library sequences (containing 3,908 sgRNAs targeting 309 mouse DDR genes with an average of 10 sgRNAs per gene, as well as 834 non-targeting sgRNA controls) was inserted into the sgRNA scaffolding region. The same process was used to generate LentiCRISPR-Cre-V2-sgControl Lumifluor plasmid (*Trp53bp1* intronic sequence on chromosome 2), LentiCRISPR-Cre-V2-sgMRE11 Lumifluor plasmid (targeting codon 633 of *MRE11*) and additional plasmids targeting *cGAS*, *STING*, *ZBP1*, *RIPK3* and *MLKL*. All plasmids created were confirmed by Sanger sequencing (Eton Bioscience).

**pBABE-Puro-Halo-MRE11 mutant plasmids.** dsDNA molecules (gBlocks Gene Fragments) were ordered from IDT to design DNA constructs with the BstBI region added to the human MRE11(H129N) or DB1 and DB2 deletion fragment. The gBlocks and pBABE-Puro-Halo vector were then cut using the BstBI restriction enzyme (NEB, R0519S). Samples were resuspended in nuclease-free distilled water (IDT, 11-05-01-04) to a concentration of 10 ng µl$^{-1}$. Samples were heated at 50 °C for 20 min per the manufacturer's instructions and stored at −20 °C. Finally, the MRE11(H129N) and MRE11(DB1Δ/DB2Δ) sequences were inserted into the pBABE-Puro-Halo vector using Quick DNA ligase (NEB, M2200S).

**pBig1a-hRAD50-hMRE11N2×STR plasmids.** pFastBac1 plasmids containing genes for recombinant human MRE11−Flag (pTP813, Addgene, 113308) and RAD50−6×HIS (pTP2620, Addgene, 113311)[40,41] were provided as gifts from T. Paull. The pBig1a-hRAD50-hMRE11N2×STR plasmid (human RAD50 untagged, human MRE11 with a C-terminal TEV protease cleavage site (N) followed by a twin-strep tag (2×STR)) was cloned by Gibson assembly as previously described[42]. In brief, hMRE11N2×STR was cloned into the pLIB vector by Gibson assembly using pTP813 as a template. pLIB-hMRE11N2×STR and pTP2620 were then used as templates to prepare pBig1a-hRAD50-hMRE11N2×STR by Gibson assembly, which was confirmed by Sanger sequencing (Azenta Life Sciences). The nuclease-dead MRE11(H129N) mutation was incorporated by site-directed mutagenesis (Q5 Hot Start High-Fidelity 2× master mix, NEB), creating pBig1a-hRAD50-hMRE11(H129N) N2×STR, which was confirmed by whole plasmid sequencing (Plasmidsaurus).

## Transfections

**siRNA transfection.** Pre-designed siRNAs were purchased from Dharmacon: negative Control scramble siRNA (4390847); and siRNA directed against human targets *MRE11* (M-009271-01), *RAD50* (L-005232-00) and *NBN1* (L-009641-00). Cells were transfected with siRNAs using Lipofectamine RNAimax reagent (Thermo Fisher Scientific, 13778075) according to the manufacturer's instructions. In brief, MDA-MB-231 cell lines plated on 6-well plates at a density of 200,000 cells per well for siRNA treatment. Twenty-four hours after plating, cells were exposed to 30 nM per well siRNAs in Opti-MEM with Lipofectamine RNAimax reagent. Forty-eight hours after transfection, cells were transfected with 200 pmol of 90-mer dsDNA (ISD90), Oregon-green hNCP (OG-hNCP)-v139 or hNCP-v52 using Lipofectamine 2000 reagent (Thermo Fisher Scientific, 11668019) according to the manufacturer's instructions.

**DNA or nucleosome transfection.** A total of 500,000 cells were seeded on an 18 mm square cover glass in a 6-well plate. Twenty-four hours after plating, Lipofectamine 2000 reagent (10 µl) in 250 µl Opti-MEM per transfection were incubated with indicated concentrations of ISD90 or nucleosomes per reaction in 250 µl Opti-MEM for 15 min at room temperature before the addition of mix (500 µl) to the cells. Cells were then fixed or collected for imaging or western blotting at each time point.

## Quiescent assay (EdU incorporation)

pMMECs were infected with a lentivirus containing sgRNA and Cre. Between 5 and 9 days after viral infection, cells were reseeded on coverslips in 6-well plates. These cells were then incubated with EdU for 24 h, fixed in 4% formaldehyde for 20 min and permeabilized in 0.5% (v/v) Triton X-100 for another 20 min. EdU detection was performed using an EdU detection kit (Sigma-Aldrich, BCK-EDU594) according to the kit's protocol.

The cells were then blocked in a solution of 3% BSA in PBS for 30 min and subsequently incubated in DAPI for 10 min. The coverslips were mounted onto slides using a mounting solution (Thermo Fisher Scientific, P36934). EdU-negative cells were examined using an EVOS M7000 fluorescence microscope.

For the quiescence assay involving drug or chemical treatment, cells were incubated with or without 25 µl 2′3′-cGAMP (Invivogen, tlrl-nacga23-1), 0.5 µM C-176 (Selleck Chemicals, S6575), 1 µM NSA (Selleck Chemicals, S8251) or 10 µM HS-1371 (Selleck Chemicals, S8775) for 6 h before the EdU treatment.

## Determining the frequency of micronuclei

After fixation and staining with DAPI, the percentage of cells with micronuclei was determined using either a fluorescence microscope (Olympus BX61 or EVOS M7000) or confocal microscopes (LSM710) under blinded conditions. Micronuclei were defined as discrete DNA aggregates that were separate from the primary nucleus and were only identified in cells in which the interphase primary nuclear morphology was normal. Cells exhibiting an apoptotic appearance were excluded from this analysis.

## Immunofluorescence

**ICC analysis.** Cells were fixed in ice-cold methanol for 10 min at −20 °C. Fixed cells were pre-incubated in blocking solution (3% BSA in PBS) followed by incubation with primary antibodies at 4 °C overnight. After incubation with primary antibodies, cells were washed three times with shaking in PBS and probed with fluorescein-conjugated (Cy3, Cy5, Alexa 488, Alexa 549 and Alexa 674) anti-mouse or anti-rabbit secondary antibodies. After washing 3× with PBS, DAPI was used for DNA counterstaining followed by mounting on slides. To determine the localization of cGAS in the cytoplasm or nucleus, cells were fixed in

3.8% formaldehyde for 15 min at room temperature. The cells were then permeabilized in 0.2% Triton X-100 for 10 min at room temperature. Fluorescence images were taken using an EVOS M7000 or Olympus BX61 or a Zeiss LSM710 spectral confocal laser scanning microscope at the UNC Microscopy Service Laboratory. Image analysis was performed using ImageJ.

The following primary antibodies were used for immunofluorescence: anti-cGAS mouse-specific (Cell Signaling, 31659; 1:500); anti-cGAS (Cell Signaling, 15102; 1:500); anti-α-tubulin (Santa Cruz Biotechnology, sc-5286; 1:500); and anti-MLKL phosphor S345 (Abcam, ab196436; 1:500). The following secondary antibodies were used (at 1:500 dilution): anti-rabbit-Alexa 488 (Thermo Fisher Scientific, A11034); anti-rabbit-Alexa 594 (Thermo Fisher Scientific, A11037); anti-rabbit-Alexa 633 (Thermo Fisher Scientific, A21072); anti-mouse-Cy3 (Jackson ImmunoResearch, 715-165-151); anti-rabbit Cy5 (Jackson ImmunoResearch, 111-175-144); and streptavidin–Alexa 488 (Thermo Fisher Scientific, S32354).

All subsequent analyses and processing of images were performed using ZEN 2011 microscope software (Zeiss) or ImageJ software.

**cGAS/pH2A.X staining for tumour immunofluorescence.** Sequential dual immunofluorescence was performed on paraffin-embedded tissues that were sectioned at 5 μm. The immunofluorescence assays were performed using a Bond automated slide staining system (Leica Microsystems) with a Bond Research Detection System kit (Leica, DS9455). cGAS (31659S) and phospho-histone H2A.X (9718s) were both purchased from Cell Signaling Technology. Slides were deparaffinized in Bond dewax solution (Leica, AR9222), hydrated in Bond wash solution (Leica, AR9590) and sequentially stained for cGAS and then pH2A.X. Specifically, antigen retrieval for cGAS was performed for 20 min at 100 °C in Bond-epitope retrieval solution 2 pH 9.0 (Leica, AR9640). After pretreatment, slides were incubated for 1 h with cGAS antibody (1:4,000) followed by Novolink Polymer (Leica, RE7161) then TSA Cy5 (Akoya Biosciences, FP1117). Afterwards, a second round of antigen retrieval was performed for 20 min at 100 °C in Bond-epitope retrieval solution 1 pH 6.0 (Leica, AR9961). Slides were then incubated with the phospho-histone H2A.X antibody (1:3,000, 2 h) then Novolink Polymer and detected using TSA Cy3 (Akoya Biosciences, FP1046). Nuclei were stained using Hoechst 33258 (Invitrogen, H3569). The stained slides were mounted using ProLong Gold antifade reagent (Life Technologies, P36930). Positive and negative controls (no primary antibody) were included in this run. Single-stain controls were done for multiplex immunofluorescence stains for which one primary antibody was omitted to ensure that cross-reactivity between the antibodies did not occur. Slides were then digitalized using Aperio ScanScope FL (Aperio Technologies). The digital images were captured in each channel using a ×20 objective (0.468 μm per pixel resolution) using line-scan camera technology[43]. The adjacent 1 mm stripes captured across the entire slide were aligned into a contiguous digital image by an image composer. Slides were scanned on a Versa slide scanner (Leica Biosystems) using a ×40 objective (0.16276 μm per pixel resolution; MPP), with an image bit depth of 8 bits per channel. Images were analysed using ImageScope software and ImageJ.

**phospho-histone H3 staining for tumour immunofluorescence.** Chromogenic immunohistochemistry (IHC) was performed on paraffin-embedded tissues that were sectioned at 5 μm. IHC was carried out using a Bond III Autostainer system (Leica Microsystems). Slides were dewaxed in Bond dewax solution and hydrated in Bond wash solution. Heat-induced antigen retrieval was performed for 20 min at 100 °C in Bond-epitope retrieval solution 1 pH 6.0. After pretreatment, slides were incubated with phospho-histone H3 (Sigma-Aldrich, 06-570) at 1:1,000 for 1 h followed by Novolink Polymer (Leica, RE7260-K). Antibody detection with 3,3-diaminobenzidine (DAB) was performed using a Bond Intense R detection system (Leica, DS9263). Stained slides were dehydrated and coverslipped with Cytoseal 60 (Thermo Fisher Scientific, 8310-4). A positive control and a negative control (no primary antibody) were included for this run. IHC-stained slides were digitally imaged using an Aperio ScanScope AT2 (Leica Biosystems) with a ×20 objective.

## Live-cell imaging

**Time-lapse analysis of pMMECs.** For cell cycle time-lapse studies, pMMECs were collected from $R26^{cas9/cas9}$, $Myc^{OE}$, and $Trp53^{fl/fl}$ virgin female mice aged 8–12 weeks and plated on collagen-coated 6-well plates at a minimum density of 50,000 cells per well using the pMMEC collection protocol. pMMECs were isolated without performing randomization. At 4 days after collection, pMMECs were transduced with 1 μl of each lentivirus per well (PCNA–mCherry and Cre-sgControl-Lumifluor or PCNA–mCherry and Cre-sgMRE11-Lumifluor) at 1–5 × 10⁵ TU using 4 μg ml⁻¹ polybrene daily for 2 days[23]. On day 7, cells were washed using EpiCult basal medium before being transferred to 12-well glass bottom plates (Cellvis, P12-1.5H-N) coated with Cell Tak (Corning, 354240), and LA7s (ATCC, CRL-2283) were lethally irradiated at 70 Gy ionizing radiation using a Bio-Rad Source RS2000 irradiator (250,000 cells per well to create a confluent feeder monolayer). pMMECs were seeded at a density of 1 × 10⁵ per well in LA7 medium (DMEM with reduced sodium bicarbonate (1.5–2.0 g l⁻¹), 4.5 g l⁻¹ glucose (Thermo Fisher Scientific, 11965084), 0.005 mg ml⁻¹ insulin (VWR, 11061-68-0), 20 mM HEPES and 5% FBS) for the co-culture. On day 10, LA7 medium was replaced with imaging-optimized LA7 medium (made with Phenol Red-free DMEM:F12 medium, Gibco 21041-025). Images of cells were captured every 20 min for 72 h in the mCherry and GFP fluorescence channels at a constant temperature of 37 °C and atmosphere (5% CO₂). Fluorescence images were obtained using a Nikon Ti Eclipse inverted microscope with a ×40 objective and a Nikon Perfect Focus system to maintain acquisition focus. Nikon's NIS Elements AR software was utilized for image acquisition. Image analysis was performed using ImageJ. Cells positive for both GFP (indicating Cas9 activation) and mCherry were analysed in the imaging dataset.

**Imaging of stable cell lines.** cGAS recruitment over time after transfection with OG-hNCP was analysed in MDA-MB-231 WT and MRE11-deficient stable cell lines expressing RFP–cGAS. Cells were plated into 12-well glass-bottom plates (Cellvis, P12-1.5H-N) supplemented with 10% FBS and sodium pyruvate. Twenty-four hours after plating, cells were transfected with OG-hNCP, then images were captured. Microscopy images were obtained using a Nikon Ti2 widefield microscope equipped with a Plan Apo ×20 (0.75 NA) air objective. Throughout the imaging experiments, cells were maintained at 37 °C and 5% CO₂ using an Okolab H301-PI-736-160×110 live cell imaging chamber. A total of two fields of view per condition were imaged every 14.5 s. All images were obtained using a Lumencor SPECTRA Light Engine for illumination, a motorized emission filter turret and a Hamamatsu Orca-Flash4.0 camera. During each round of image acquisition, four successive images (Brightfield, Halo-JF646-MRE11, RFP–cGAS and OG-hNCP) were obtained. Nikon's NIS Elements AR software was utilized for image acquisition. Image analysis was performed using ImageJ.

**FRAP assays.** For FRAP assays, MDA-MB-231 cells expressing GFP–cGAS were grown in a 4-well glass-based dish (Thermo Scientific, 155382PK). Cells were transfected with siControl or si*MRE11* for 60 h, then FRAP assays were performed on a LSM710 confocal microscope. All experiments were carried out at 37 °C, and imaging was performed with a ×63 oil-objective lens using the bleaching mode of the Zeiss software. For each experiment, five pre-bleach images were taken, and a single ROI spot was bleached with the 405 nm line at 20% transmission. To gain the full stack of focal planes during recoveries, the pinhole of 488 nm was adjusted to 384 nm. Then, images were collected over a period

of 60 min every 1 min. Image analysis was performed using ZEN 2011 (Zeiss) and ImageJ software. Fluorescence images were taken using a Zeiss LSM710 spectral confocal laser scanning microscope at the UNC Microscopy Service Laboratory. Image analysis was performed using ImageJ.

## Growth assays

pMMECs were infected with sgControl, sg*Mre11*, sg*cGas* or sg*Sting* Cre-lenti-viruses. Five days after infection, the cells were seeded into 12-well plates at a density of $3 \times 10^4$ cells per well. Duplicate samples were then collected every 2–3 days for a period of 15 days. The total number of cells per well was counted, and the cells were fixed in 3% PFA. Following fixation, the cells were subjected to flow cytometry analysis using an Attune NxT flow cytometer to detect the presence of GFP.

## cGAMP ELISA

For 2′3′-cGAMP quantification, cells were collected by trypsinization using 0.05% trypsin EDTA (Gibco, 25300054) for 5 min. Cell pellets were then lysed in RIPA lysis buffer containing 50 mM Tris, 150 mM NaCl, 1% (w/v) sodium deoxycholate, 0.03% (v/v) SDS, 0.005% (v/v) Triton X-100, 5 mM EDTA, 2 mM sodium orthovanadate and complete protease inhibitor cocktail (Roche). The pellet from 1 well of a 6-well plate was resuspended in 130 μl of RIPA lysis buffer and incubated for 30 min on ice. The lysed cells were then centrifuged for 10 min at 18,200$g$ and 4 °C. The supernatant (100 μl) was subsequently used for the cGAMP ELISA assay (Cayman, 501700) as per the manufacturer's instructions. Protein concentration in the supernatant was measured using a Qubit 3.0 fluorometer system, which was then used to normalize the 2′3′-cGAMP levels.

## Immunoblotting

Cell pellets were lysed in 2× Laemmli sample buffer (Bio-Rad, 161-0737) containing 4% β-mercaptoethanol. The samples were then heated to 95 °C for 5 min, and protein concentrations were determined using a Qubit 3.0 fluorometer (Thermo Fisher Scientific, Q33216) according to the manufacturer's instructions. Approximately 10–30 μg of total protein was separated on SDS–PAGE gels (4–20% (Mini-PROTEAN TGX, Bio-Rad), 10% or 15% (v/v)). The gels were then transferred onto PVDF membranes (Bio-Rad, 1704272) using a Trans-Blot Turbo transfer system (Bio-Rad, 1704270).

The membranes were briefly washed in Tris-buffered saline-Tween 20 (TBST) (50 mM Tris-Cl, pH 7.5, 150 mM NaCl and 0.1% (v/v) Tween 20) and then blocked in 5% non-fat dry milk (NFDM) in TBST for 1 h at room temperature. Primary antibodies were incubated in TBST containing 1% NFDM overnight at 4 °C, whereas secondary antibodies were incubated in TBST containing 0.1% NFDM for 1 h at room temperature.

Proteins were visualized using enhanced chemiluminescence substrate ECL (Thermo Fisher Scientific, 32106) and imaged using a ChemiDoc XRS Bio-Rad Imager and Image Lab 6.0.0 software. Imaging was performed in two channels: chemiluminescence and colorimetry.

## Quantitative PCR with reverse transcription (RT–qPCR)

RNA was extracted using an RNeasy Plus Mini kit (Qiagen, 74136) following the manufacturer's instructions. The concentrations of RNA were determined using a NanoDrop spectrophotometer (Thermo Fisher Scientific, 701-058112). The RNA was then reverse-transcribed using a Maxima First Strand cDNA Synthesis kit (Thermo Fisher Scientific, FERK1672), with two reverse transcription reactions being performed for each sample, each using 100 ng of RNA. The RT–qPCR assays were carried out using Fast SYBR Green master mix (Thermo Fisher Scientific, 4385617), and run on QuantStudio 6 and 7 Flex Real-Time PCR systems (Thermo Fisher Scientific). The cycling conditions involved an initial step at 95 °C for 15 min, followed by 40 two-step cycles (95 °C for 15 s; 60 °C for 60 s).

## Luciferase ISG reporter assay

RAW-Lucia ISG cells are mouse macrophages containing an ISG-luciferase reporter. All cells were maintained following the manufacturer's recommendations and incubated at 37 °C in a humidified atmosphere containing 5% $CO_2$. RAW-Lucia ISG cells stably express an IFN-sensitive response element from the mouse ISG54 minimal promoter and five IFN-stimulated response elements (ISRE-ISG54) coupled to a synthetic coelenterazine-utilizing luciferase.

RAW-Lucia ISG cells were plated in 96-well plates (Corning, 3922) at $1 \times 10^5$ cells per well in 200 μl of the medium. Cells were allowed to adhere overnight before the medium was replaced with 180 μl of fresh medium. Target genes in pMMEC cells were knocked out by lentiviral infection. Medium from the pMMECs was centrifuged at 500 r.p.m. for 5 min before adding 20 μl to the RAW-Lucia ISG cells. After 24 h, 10 μl of medium from the RAW-Lucia ISG cells was added to a white-bottom 96-well plate, with 50 μl of QUANTI-Luc (Invivogen, rep-qlc1) detection reagent added immediately before luciferase detection. Luminescence was measured using a luminometer (SpectraMax i3x) with a 4 s start time and 0.1 s reading time.

## Expression and purification of MRN complexes

**The human MRN complex.** MRE11–Flag/RAD50–6×His/NBS1–Flag was prepared by co-expression in Sf9 cells. In brief, the pFastBac1 plasmids containing genes for recombinant human MRE11–Flag, RAD50–6×HIS and NBS1–Flag[40,41] were transformed into DH10Bac cells (Thermo Fisher Scientific, 10361012) to prepare individual bacmids. Individual bacmids were transfected into Sf9 cells to generate low titre P1 baculoviruses, which were subsequently used to prepare high titre P2 and P3 baculoviruses using the Bac to Bac system following the manufacturer's recommendations (Invitrogen). For co-expression of the MRN complex, Sf9 cells ($2.5 \times 10^6$ per ml) grown in Sf-900 III SFM (Thermo Fisher Scientific, 12659017) were co-infected with baculoviruses expressing MRE11–Flag, RAD50–6×HIS and NBS1–Flag, and the complex was expressed at 27 °C for 48 h. Cells were collected by centrifugation at 500$g$ for 15 min at 4 °C and the cell pellet was rinsed with PBS and stored at −80 °C until purification.

**Purification of the MRE11–Flag/RAD50–6×His/NBS1–Flag complex.** The Sf9 cell pellet was resuspended in lysis buffer (50 mM Tris-HCl, pH 8.0, 150 mM NaCl, 10% glycerol, 1% Triton X-100, 1 mM benzamidine and 0.2 mM PMSF) containing Roche complete EDTA-free protease inhibitor cocktail (Sigma-Aldrich, 11873580001) for 40 min with continuous stirring before centrifugation at 33,000$g$ for 45 min at 4 °C. The supernatant was loaded by gravity flow onto a column packed with 3 ml of anti-Flag M2 affinity gel (Sigma-Aldrich, A2220) pre-equilibrated in wash buffer (50 mM Tris-HCl, pH 8.0, 150 mM NaCl, 10% glycerol and 1 mM benzamidine). Bound MRN complex was washed with wash buffer and eluted with wash buffer supplemented with 0.1 mg ml$^{-1}$ purified 3×Flag peptide (Sigma-Aldrich, F4799). The eluted sample was diluted 2× with dilution buffer (25 mM Tris-HCl, pH 8.0, 10% glycerol and 10 mM β-mercaptoethanol) to decrease the NaCl concentration to 50 mM and immediately purified by anion-exchange chromatography with a Source Q resin (GE Healthcare, 17094705) using a gradient from 50 to 1,000 mM NaCl. Pooled fractions were further purified by gel filtration chromatography using a Superose 6 Increase 10/300 GL column (GE Healthcare, 29-0915-96) pre-equilibrated with 20 mM Tris-HCl, pH 8.0, 200 mM NaCl, 10% glycerol and 1 mM DTT. Purified MRN complex was concentrated using a Vivaspin 500 centrifugal concentrator (Vivascience) and stored at −80 °C. Sample purity was assessed by SDS–PAGE with Coomassie blue staining.

**Purification of untagged MR and nuclease-dead MR complex.** Sf9 cells were lysed as above. The supernatant was loaded onto a 4 ml strep-tactin XT (IBA Lifesciences, 2-4030-010) column, washed with

wash buffer and then bound MR complex was eluted with wash buffer supplemented with 25 mM biotin. The eluted complexes were digested with TEV protease in dialysis buffer containing 25 mM Tris-HCl, pH 8.0, 100 mM NaCl, 10% glycerol and 10 mM β-mercaptoethanol overnight in at 4 °C. The digested MR complexes were further purified by Source Q and Superdex 200 increase columns (GE Healthcare, 28990944) as described above. Purified MR complexes were concentrated as above and stored at −80 °C. The NBS1–Flag protein was purified as described above for reconstitution with the untagged MR complex.

**Mouse cGAS catalytic domain.** The mouse cGAS catalytic domain containing residues 141–507 (mcGAS) was prepared as previously described[4]. The human cGAS catalytic domain (residues 157–522) was prepared as previously described[44]. The 6×His-Sumo tagged mouse cGAS catalytic domain (6×His-Sumo-mcGAS) used in the TR-FRET assay was expressed in BL21(DE3)pLysS cells. Cells were grown in 2XTY medium at 37 °C, and protein expression was induced through the addition of 0.2 mM IPTG when the OD600 reached 1.2 for 18 h at 16 °C. The 6×His-Sumo-mcGAS protein was purified using metal-affinity chromatography with Talon resin (Clontech, 635504), ion-exchange chromatography with Source S resin (GE Healthcare, 17094405) and size-exclusion chromatography with a Superdex 75 increase 10/300 GL column (GE healthcare), concentrated, supplemented with 20% glycerol and stored at −80 °C.

**Fluorescent labelling of the mcGAS catalytic domain.** The purified mcGAS was applied to a Zeba Spin desalting column (Thermo Fisher Scientific, 89878) for buffer exchange into labelling buffer (20 mM HEPES, pH 7.5, 150 mM NaCl and 1 mM TCEP) following the manufacturer's recommendations. To facilitate in-gel detection, mcGAS was labelled with carboxyrhodamine. In brief, mcGAS was diluted to 0.5 mg ml$^{-1}$ in labelling buffer and combined with one molar equivalent of 5-(and-6)-carboxyrhodamine 6G, succinimidyl ester (Thermo Fisher Scientific, C6157) and labelling was allowed to proceed for 2 h at 4 °C. Labelling was repeated with the addition of one more molar equivalent of fluorophore, followed by 2 h of incubation at 4 °C. Unreacted fluorophore was quenched through the addition of Tris-HCl, pH 8.0 to a final concentration of 10 mM. Carboxyrhodamine-labelled mcGAS (mcGAS-CR) was separated from quenched, unreacted fluorophore using a Zeba Spin desalting column as described above, and protein was quantitated using previously reported methods[2].

## Preparation of nucleosomes

Nucleosomes were reconstituted using recombinant human histones (H2A, H2B, H3.2 and H4) and 147 bp (1 bp symmetric linker DNA) or 185 bp (20 bp symmetric linker DNA) 601 nucleosome positioning sequence and purified by anion-exchange chromatography as previously described[45]. AP mutant nucleosomes were prepared using the H2A (E61A, E64S, N68A, D72S, N89A, D90A, E91S) mutants as previously reported[45]. For preparation of OG-NCPs, a H2A-H2B(A21C) dimer (2 mg ml$^{-1}$) was incubated with Oregon Green 488 maleimide at a 1:1 ratio in labelling buffer (20 mM Tris-HCl, pH 7.5, 25 mM NaCl, 7 M guanidine HCl and 0.2 mM TCEP) for 2 h at 4 °C. Labelling was repeated with the addition of one more molar equivalent of fluorophore as above. The unfolded, labelled dimer was dialysed overnight at 4 °C in refolding buffer (10 mM HEPES, pH 7.5, 100 mM NaCl and 10 mM β-mercaptoethanol) and then purified by ion-exchange chromatography with a Source S resin (GE Healthcare). The labelled dimer was reconstituted into nucleosomes and purified as described above. The 147 bp nucleosomes were reconstituted with H2B(K125C)-biotin (biotin-NCP) or H2B(K125C)-Atto647N (647-NCP) for the nucleosome stacking assay. For this purpose, H2A-H2B(K125C) dimers were labelled with biotin-maleimide (Sigma-Aldrich, B1267) or Atto 647N maleimide (Sigma-Aldrich, 05316-1MG-F) as described above for Oregon Green 488 labelling.

## TR-FRET assay

cGAS–MRN competition assays were performed as previously described[25]. In brief, 15 nM 6×His-Sumo-mcGAS, 2 nM biotin NCP, 4 nM LANCE Eu-W1024 streptavidin (PerkinElmer), 10 nM Ulight LANCE Ultra ULight-anti-6×His (PerkinElmer, TRF0134) and 0–2,000 nM of untagged MRN complex (mixture of 2 unlabelled MR:1 NBS1–Flag) or 0–5,000 nM unlabelled nucleosome were mixed in a total volume of 10 μl of TR-FRET 50 buffer (10 mM Tris-HCl, pH 7.5, 50 mM NaCl, 5% glycerol, 0.01% NP-40, 0.01% CHAPS, 5 mM DTT and 100 μg ml$^{-1}$ BSA) in non-binding 384-well plates (Greiner Bio-One, 784904). Plates containing reaction mixture were shaken at 1,000 r.p.m. on a plate shaker for 1 min, centrifuged at 1,000 r.p.m. for 1 min and then incubated for 1 h at room temperature.

Nucleosome stacking by cGAS and MRN was tested by mixing 2 nM biotin-NCP, 50 nM 647-NCP, 2 nM LANCE Eu-W1024 streptavidin and 0–2,000 nM untagged MRN (mixture of 2 unlabelled MR:1 NBS1–Flag), 0–5,000 nM mcGAS or 0–5,000 nM hcGAS in TR-FRET 50 buffer.

For nucleosome stacking competition by MRN, the mixture containing 2 nM biotin-NCP, 50 nM 647-NCP, 2 nM LANCE Eu-W1024 streptavidin and 250 nM hcGAS was titrated with 0–2,000 nM MRN (mixture of 2 unlabelled MR:1 NBS1–Flag).

For all TR-FRET assays, FRET signals were measured after 1 h of incubation using 320 nm excitation and 615 nm/665 nm dual emission with a 100 μs delay between excitation and emission on an EnVision 2103 plate reader. All titrations were performed in triplicate. Competition data were fit to [inhibitor] versus response (three parameters) using GraphPad Prism v.7 software.

## MRN–cGAS interaction by size exclusion

mcGAS-CR (5 μM) was incubated with 1 μM MRN complex in T50 buffer (10 mM Tris-HCl, pH 7.5, 50 mM NaCl and 1 mM DTT) on ice for 20 min and then loaded onto a Superdex 200 Increase 10/300 column equilibrated with T50 buffer. The eluted fractions were separated by SDS–PAGE and fluorescence imaging was performed with a Typhoon FLA-9500 imager (GE Healthcare, excitation: 532 nm, emission: LPG low pass 575 nm), before Coomassie blue staining.

## Electrophoretic mobility shift assay

Electrophoretic mobility shift assays were performed by combining nucleosomes (50 or 100 nM final concentration) with serial dilutions of tagged MRN or MN complex and 200 nM mcGAS-CR in 10 μl of binding buffer (20 mM HEPES, pH 7.5, 50 mM NaCl, 5% sucrose and 1 mM DTT). Following 15–30 min equilibration on ice, samples were analysed by electrophoresis on 5% polyacrylamide gels run in 0.2× TBE at 150 V for 60 min at 4 °C. Gels were scanned to detect fluorescence signals of mcGAS-CR using a Typhoon FLA 9500 imager and then stained with ethidium bromide and imaged using a E-Gel Imager (Invitrogen).

## In vitro assay for cGAS activity

Assays were performed as previously described[4] with the following alterations. In vitro reactions (10 μl) reactions were assembled in PCR tubes with final concentrations of 0.5 μM or 1 μM purified mcGAS catalytic domain, 5 μM dsDNA (90 bp), 0.5 μM NCP, with varying (0–2 μM) MRN complex concentrations in reaction buffer (22 mM HEPES pH 7.5, 110 mM NaCl, 5.5 mM MgCl$_2$, 5 mM DTT, 10 μM ZnCl$_2$, 1.25 U of inorganic pyrophosphatase (Sigma-Aldrich, I1643) and 4% glycerol (final concentrations)). Reactions were equilibrated for 5 min at 37 °C and initiated with the addition of 2 mM (each) GTP/ATP mix containing about 12.5 nM [α-$^{32}$P]ATP (PerkinElmer, NEG003X250UC) and incubated at 37 °C for 120 min.

## Single-cell analysis

**Single-cell RNA-seq library preparation.** For 3′ single-cell RNA library construction, cells were processed using a 10x Genomics Chromium

Controller and a Chromium Single Cell 3′ GEM, Library & Gel Bead Kit v.3.1 (PN-1000121) following the manufacturer's user guide (tinyurl.com/3we33fb6). In brief, 1:10 volume of 10% BSA was added to freshly sorted cells before centrifuging for 6 min at 800 r.c.f. in a 4 °C cold room. The supernatant was removed, and the cell pellet was resuspended in 25 µl of 1× PBS + 0.04% BSA. Aliquots of the sorted, concentrated cells were then stained with acridine orange and propidium iodide and assessed for viability and concentration using the LUNA-FL dual fluorescence cell counter (Logos Biosystems). Approximately 16,000 viable cells per sample were loaded onto a Chromium Chip G with a target recovery of 10,000 cells per sample for library preparation. Single cells, reverse transcription reagents and gel beads coated with barcoded oligonucleotides were encapsulated together in an oil droplet to produce gel beads in emulsion (GEMs). Reverse transcription was performed using a C1000 thermal cycler (Bio-Rad) to generate cDNA libraries tagged with a cell barcode and unique molecular index (UMI). GEMs were then disrupted and the cDNA libraries were purified using Dynabeads MyOne SILANE (Invitrogen) before 11 amplification cycles. Amplified libraries were purified using SPRIselect magnetic beads (Beckman Coulter) and quantified using an Agilent Bioanalyzer High Sensitivity DNA chip (Agilent Technologies). Fragmentation, end repair, A-tailing and double-sided size selection using SPRIselect beads were then performed. Illumina-compatible adapters were ligated onto the size-selected cDNA fragments. Adapter-ligated cDNA was then purified using SPRIselect beads. Uniquely identifiable indexes were added during ten amplification cycles. The finalized sequencing libraries were purified using SPRIselect beads and visualized using an Agilent Bioanalyzer High Sensitivity DNA chip (Agilent Technologies).

**Processing of scRNA-seq data.** Data were imported into Seurat (v.3.1.2) using R (v.3.6.0). Cells with at least 5,000 UMIs and 1,000 genes detected and fewer than 10% mitochondrial contribution were retained for downstream analyses. Data were then normalized and scaled using scTransform, integrated using IntegrateData as previously described[45], then clustered using Louvain–Jaccard clustering with multilevel refinement (resolution of 1.0). Unique marker genes were detected for each cluster using Presto. Differential expression analysis was performed on CPM-normalized counts using Seurat FindMarkers, and significantly enriched pathways among upregulated and downregulated genes were detected using gProfiler. Cell cycle scoring was then performed in Seurat to identify the different phases. We performed differential expression between sgControl and sg*MRE11* in the G1 cells based on the nonparametric Wilcoxon rank sum test, following re-normalization as described above.

**Orthotopic transplantation-based tumorigenesis assay**
For mammary-fat-pad tumour assays, pMMECs were isolated from 8-week-old virgin female *R26^cas9/cas9*, *Myc^OE* and *Trp53^fl/fl* mice. After isolation, pMMECs were infected with 5 × 10^5 TU of indicated LentiCRISPR-Cre-V2-sgRNA lentiviruses. After a week, the pMMECs were collected by trypsinization, washed twice in PBS and counted. Cells were then resuspended (1 × 10^6 cells) in a 50:50 solution of PBS and Matrigel matrix (Corning, 356231). Each pMMEC preparation was injected into the left mammary fat pads of NOD.129S7(B6)-Rag1tm1Mom/J mice. Following the injections, the mice were monitored on a weekly basis for the development of tumours. After detection of tumour formation, the frequency of measurements was increased to three times a week.

The study adhered to humane end point criteria for tumour burden. Euthanasia was carried out when a single tumour reached a size of 2.0 cm in any direction. In cases when multiple tumours were present, the end point was established at 1.5 cm for any individual tumour. After reaching these end points, the mice were humanely euthanized in accordance with approved guidelines.

**Z-RNA and Z-DNA staining in pMMECs**
pMMECs were isolated from indicated 8-week-old virgin female *R26^cas9/cas9*, *Myc^OE* and *Trp53^fl/fl* mice, unless otherwise specified. Following isolation, pMMECs were infected with a multiplicity of infection of 3 using the indicated lentiviruses (sgControl, sg*Mre11*, sg*cGas* or sg*Zbp1*). Ten days after infection, cells were prepared for ICC to detect Z-RNA and Z-DNA. Cells were fixed with cold methanol for 10 min at 20 °C, then blocked with 3% BSA in PBS for 30 min. The cells were then incubated overnight at 4 °C with a Z-DNA antibody (Novous Biologicals, NB100-749) at a dilution of 1:500. The following day, cells were incubated with the secondary antibody (Alexa-594) for 1 h at room temperature. Cells that exhibited robust Z-RNA/Z-DNA signal aggregation were then counted. These signals manifested as either intense intranuclear clusters or as smaller aggregates dispersed around the perinuclear region.

**Statistics and reproducibility**
Statistical analyses are described in each figure legend. All data are plotted as averages, with error bars representing the s.e.m. unless stated otherwise. Data shown are representative of at least three independent biological replicates, unless otherwise indicated. Statistical analyses were performed using Prism (GraphPad Software). For all quantitative measurements, normal distribution was assumed, with *t*-tests performed unpaired and two-sided unless otherwise stated. For experiments combining several groups, an ordinary one-way ANOVA test was used. $P > 0.05$ was considered non-significant. Mouse tumorigenesis studies utilized publicly available sample size estimation calculators to attain at least 80% power to detect a 30% reduction in tumour latency using a two-tailed log-rank test. Sample sizes for in vitro experiments were determined empirically from previous experimental experience with similar assays and/or from sizes generally used in the field.

**Reporting summary**
Further information on research design is available in the Nature Portfolio Reporting Summary linked to this article.

## Data availability

All in vivo CRISPR screen sequencing data and single-cell RNA-seq data generated in this study have been deposited in the National Center for Biotechnology Information (NCBI) Gene Expression Omnibus. BioSample accession numbers SAMN36935033 and SAMN36935034, NCBI BioProject URL www.ncbi.nlm.nih.gov/bioproject/1004263. Raw, uncropped images of western blots are provided in the Supplementary Information. Mouse strains, CRISPR libraries, plasmids and all other reagents can be obtained from the corresponding author. Source data are provided with this paper.

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

**Acknowledgements** We would like to thank J.H. Petrini, Y. Pylayeva-Gupta and members of the Gupta, Purvis, McGinty and Zhang laboratories for insightful discussions and manuscript review; members of the UNC Pathology Services Core, Animal Surgery Core, Advanced Analytics Core, High Throughput Sequencing Facility, Hooker Imaging Core, Microscopy

Services Laboratory, and the Bioinformatics and Analytics Research Collaborative for excellent technical support. The schematic in Fig. 4h was created using BioRender (www.biorender.com). This work was supported by grants from the NIH/NCI (R37 CA227837, to G.P.G.), the ASTRO/BCRF (to G.P.G.) and the V Foundation (to G.P.G. and C.M.P.). G.P.G. is a recipient of the Burroughs Wellcome Career Award for Medical Scientists. Additional funding sources include the NIH/NIGMS (R01 GM138834 to J.E.P.), the NSF (CAREER Award 1845796 to J.E.P.), the NIH/NCI (P50 CA058223 to C.M.P.), the NIH/NIGMS (R35 GM133498 to R.K.M.), the University of North Carolina at Chapel Hill (start-up funds and a Jefferson Pilot Fellowship to Q.Z.) and a Gabrielle's Angel Foundation Medical Research Award (to P.L.). This material is based on work supported by the National Science Foundation Graduate Research Fellowship Program under grant no. DGE-1650116 (to J.A.S.). Any opinions, findings and conclusions or recommendations expressed in this material are those of the authors and do not necessarily reflect the views of the National Science Foundation. J.A.S. is supported by a fellowship from the Royster Society of Fellows at the University of North Carolina at Chapel Hill, and the aforementioned NSF Graduate Research Fellowship Program. Core Facility Services are supported by the Cancer Center Support Grant (P30 CA016086) to the UNC Lineberger Comprehensive Cancer Center, and the Advanced Analytics Core is supported by NIH Grant P30 DK034987.

**Author contributions** Conceptualization: G.P.G., M.-G.C. and R.J.K. Methodology: M.-G.C., R.J.K., C.-C.L., J.A.B., J.A.S., K.F.-S., D.A.S. and Y.W. Investigation: M.-G.C., R.J.K., C.-C.L., J.A.B., J.A.S., K.F.-S., D.A.S., Y.W., C.E.F., A.M.G., L.M.L, S.W.E. and Q.W. Funding acquisition: G.P.G., C.M.P., Q.Z., R.K.M. and J.E.P. Formal analysis: M.-G.C. (Figs. 1j, 2c–j, 3a–k,i,m,n and 4a–h and Extended Data Figs. 1–6 and 8–10), R.J.K. (Figs. 1c–j and 2b), C.-C.L., J.A.B., J.A.S., K.F.-S., D.A.S. and C.E.F. Data curation: M.-G.C. (Figs. 1j, 2c–j, 3a–k,i,m,n and 4a–h and Extended Data Figs. 1–6 and 8–10), R.J.K. (Figs. 1c–j and 2b), J.A.S. (Fig. 4a,b), D.A.S. (Fig. 1e), C.F. (Figs. 1a and 4i,j), C.-C.L. (Fig. 3h–i and Extended Data Fig. 7a–g), J.A.B. (Fig. 3j and Extended Data Fig. 7h–j), K.F.-S. (Figs. 1d,g and 2k), A.M.G. (Fig. 1j), L.M.L. (Fig. 4h), S.W.E. (Extended Data Fig. 4b), Q.W. (Fig. 3a,b) and Y.W. (Fig. 3c). Project administration: G.P.G. Resources: A.Y.H., P.L., C.M.P., Q.Z., R.K.M., J.E.P. and G.P.G.; Writing, original draft: M.-G.C., R.J.K. and G.P.G.; Writing, reviewing and editing: all authors. Supervision: G.P.G. (all figures), P.L. (Fig. 3c), C.M.P. (Figs. 1a and 4i,j), Q.Z. (Fig. 3j and Extended Data Fig. 7h–j), R.K.M. (Fig. 3j–i and Extended Data Fig. 7a–g) and J.E.P. (Fig. 2b).

**Competing interests** G.P.G. is a co-inventor on a patent held by the University of North Carolina at Chapel Hill on methods for detecting tumour-derived viral DNA in blood (US patent 11,168,373), receives patent licensing fees from and has equity in Naveris, and is the recipient of research funding from Breakpoint Therapeutics and Merck. C.M.P is an equity stockholder and consultant of BioClassifier. C.M.P. is also listed as an inventor on patent applications for the Breast PAM50 Subtyping assay. M.-G.C, R.J.K., C.-C.L., J.A.B., J.A.S., K.F.-S., D.A.S., C.F., C.E.F., A.M.G., L.M.L., S.W.E., Q.W., Y.W., A.Y.H., P.L., Q.Z., R.K.M. and J.E.P. have no competing interests.

## Additional information

**Correspondence and requests for materials** should be addressed to Gaorav P. Gupta.

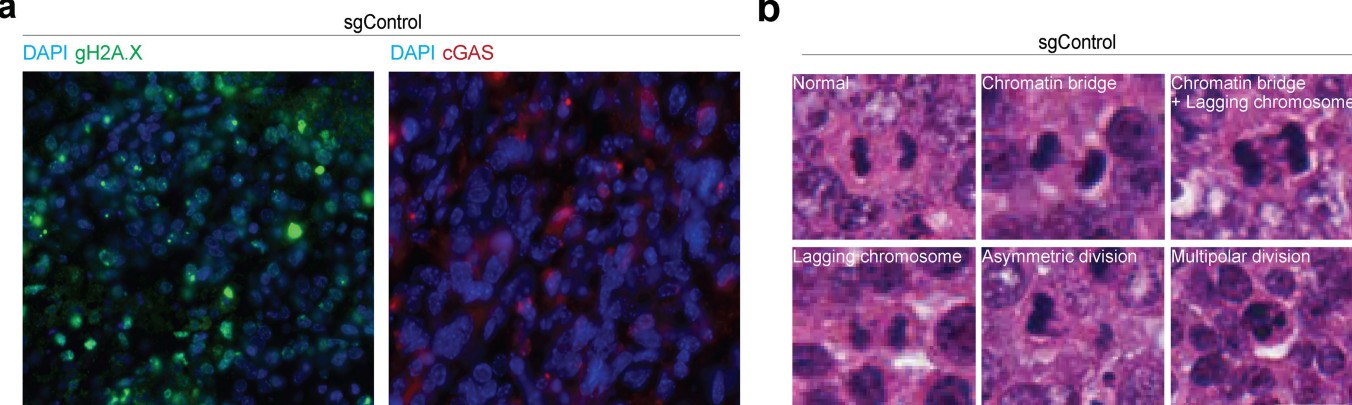

**a**

sgControl

DAPI gH2A.X

DAPI cGAS

**b**

sgControl

Normal

Chromatin bridge

Chromatin bridge + Lagging chromosome

Lagging chromosome

Asymmetric division

Multipolar division

**Extended Data Fig. 1 | DNA damage and genome instability in mammary tumors induced in *Rosa26*[LSL-Myc/LSL-Cas9]*;Trp53*[FL/FL] mice by in vivo transduction with Cre-sgControl or Cre-sgMre11. a**, Immunofluorescence imaging of mammary tumors induced by mammary intraductal injection of lentivirus expressing Cre recombinase and control sgRNA (targeting a non-coding region on Chromosome 2) for gH2A.X or cGAS showing abundant DNA damage (gH2A.X foci) and cytoplasmic cGAS localization, demonstrating consistent results across at least three independent repetitions. Scale bar, 50 µm. **b**, Mitotic aberration figures were morphologically analyzed in H&E-stained sections from mouse mammary tumors induced by Cre and sgControl (top panel) or *sgMre11* (bottom panel). sgControl-induced mouse tumor tissues have 56.5% normal, 17.4% chromatin bridges, 13% lagging chromosomes, 13% multipolar divisions, and 0% asymmetric divisions in mitotic cells. In contrast, *sgMre11*-induced tumors displayed 56% normal, 30% chromatin bridges, 1.2% lagging chromosomes, 1.2% multipolar division and 12% asymmetric division in mitotic cells, revealing consistent findings across at least three independent replicates. Scale bar, 20 µm.

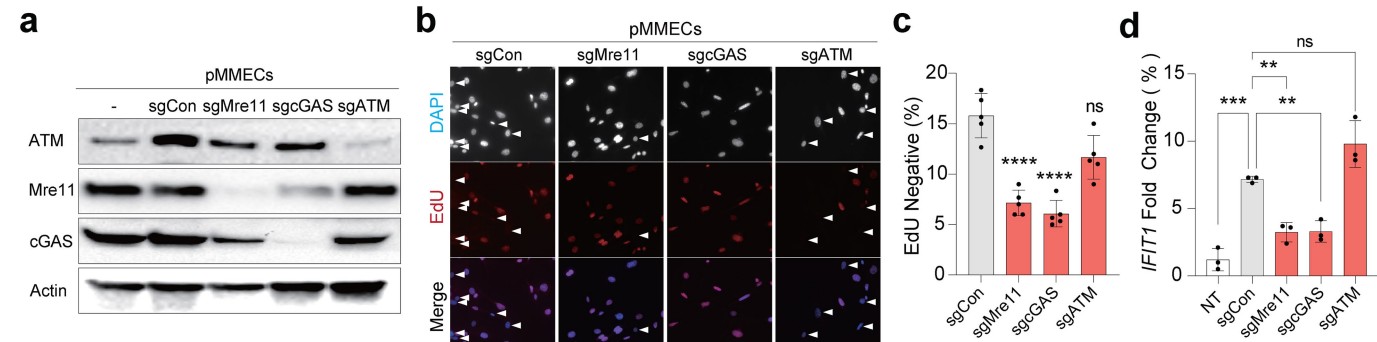

**Extended Data Fig. 2 | Mre11, but not ATM, is required for quiescence and ISG induction in $Myc^{OE}p53^{-/-}$ pMMECs. a**, Western blot confirmation of ATM, Mre11, cGAS, and Actin in $Myc^{OE}p53^{-/-}$ pMMECs expressing the indicated sgRNAs, demonstrating consistent findings across three independent replicates. **b**, Representative images for 24-hours EdU pulse-chase quiescence assay in $Myc^{OE}p53^{-/-}$ pMMECs. Scale bar, 20μm. **c**, Quantification of EdU negative percentage. n = 1,500 (*sgControl*, *sgMre11*, *sgcGas* and *sgAtm*) independent cells. ****, $p < 0.0001$. Representative of 2 independent biological experiments; **d**, 8 days after infection with each lentivirus, qRT-PCR was performed to normalize gene expression levels for interferon-stimulated genes *IFIT1*, with mRNA levels normalized to β-actin mRNA levels. N = 3 (*sgCon, sgMre11, sgcGas* and *sgAtm*) independent samples. NT vs sgControl; $p = 0.0002$. sgControl vs *sgMre11*; $p = 0.0047$, *sgcGas*; $p = 0.0052$. $n = 3$ independent biological experiments;. Data are mean ± SEM. Unless otherwise specified, statistical analyses were determined by one-way ANOVA followed by Sidak's multiple comparison post-test using a two-tailed test: *, $p < 0.05$; **, $p < 0.01$; ***, $p < 0.001$; ****, $p < 0.0001$.

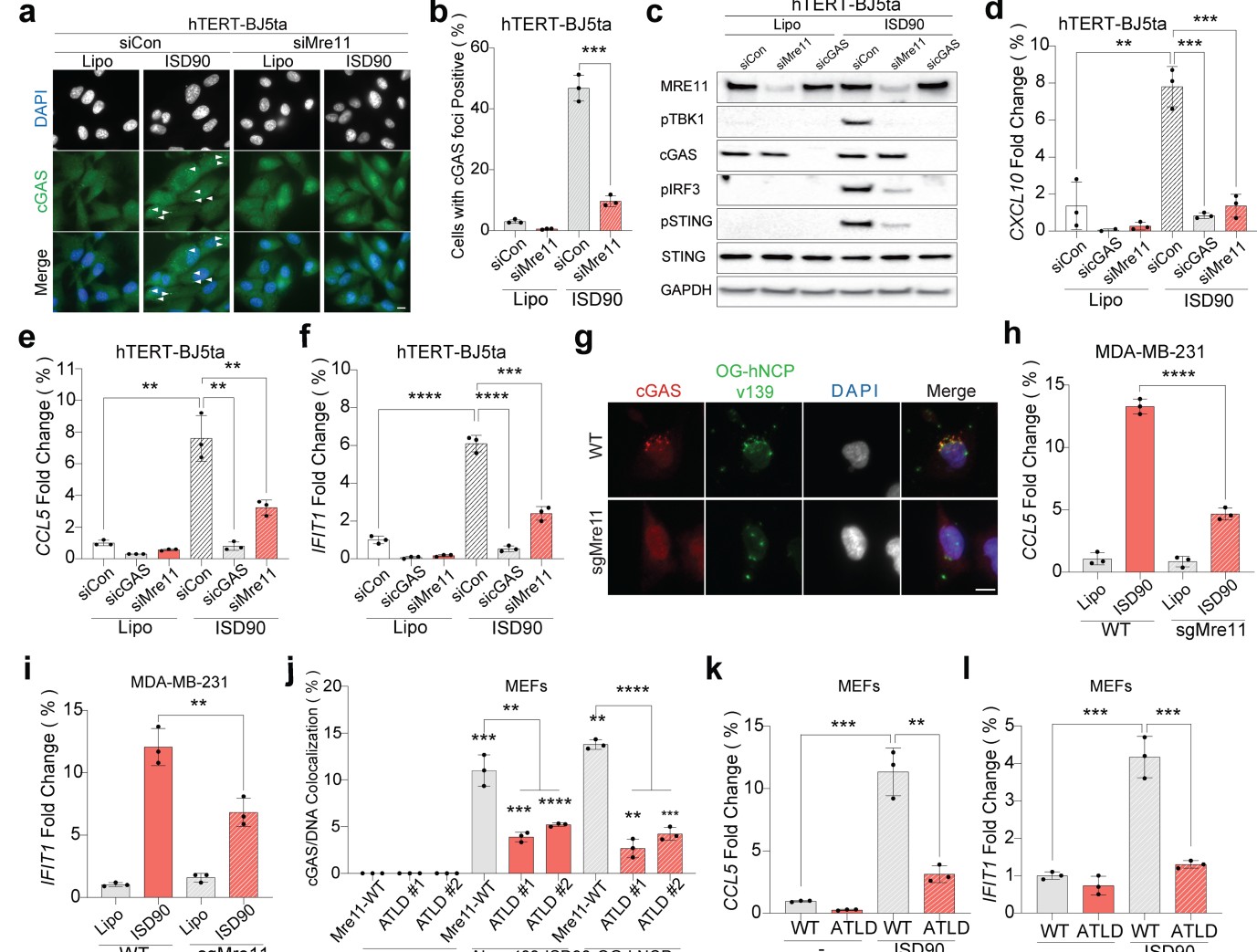

**Extended Data Fig. 3 | Mre11 is essential for activation of the cGAS-STING pathway by cytosolic DNA or nucleosomal core particles (NCPs). a**, **b**, ICC of cGAS localization to cytoplasmic interferon stimulating dsDNA 90 bp (ISD90) **a**, 1 h after transfection in siControl vs *siMRE11* BJ-5ta cells (72 h after siRNA transfection). **b**, Percentage of cells with cGAS foci. Scale bar, 10 μm. n = 900 cells for each cell lines. siControl+ISD90 vs *siMRE11*+ ISD90; *p* = 0.0001. *n* = 3 independent biological experiments; **c**, Western blot was performed on innate immune signaling pathways in siControl, *siMRE11*, and *sicGAS* BJ-5ta cells, 3 h post transfection with 4 μg/mL dsDNA 90 bp (ISD90), demonstrating consistent findings across three independent replicates. **d**, **e**, and **f**, Quantitative RT-PCR was used to analyze normalized gene expression for human interferon-stimulated genes (ISGs), namely *CXCL10*, *CCL5*, and *IFIT1* in BJ-5ta cells. *n* = 2 independent biological experiments; 3 samples for each cell lines. (**d**) siControl (Lipo vs ISD90); *p* = 0.0027. siControl+ISD90 vs *sicGAS*+ISD90; *p* = 0.0004 and *siMre11* + ISD90; *p* = 0.0009. (**e**) siControl (Lipo vs ISD90); *p* = 0.0014. siControl+ISD90 vs *sicGAS*+ISD90; *p* = 0.0013 and *siMRE11* + ISD90; *p* = 0.0076. (**f**) siControl (Lipo vs ISD90); *p* < 0.0001. siControl+ISD90 vs *sicGAS*+ ISD90; *p* < 0.0001 and *siMRE11* + ISD90; *p* = 0.0003. **g**, Representative immunocytochemistry analysis of cGAS (Red) localization to cytoplasmic Oregon Green conjugated human NCP (OG-hNCP) (Green), 60 min after transfection with 72 nM OG-hNCP in siControl or *siMRE11* human MDA-MB-231 cells. Cell nuclei stained by 4′,6-diamidino-2-phenylindole (DAPI; Blue). Scale bar, 10 μm. Image quantification is shown in Fig. 3b, right panel. **h**, **i**, qRT-PCR normalized gene expression for human interferon-stimulated genes (*ISG*) *IFIT1* and *CCL5*. Cells were transfected with Lipo only or 4 μg/mL ISD90 for 6 h. mRNA levels were normalized to β-actin mRNA levels. *n* = 3 independent biological experiments; 3 samples for each cell lines. (**h**) ISD90 (WT vs *sgMRE11*); *p* < 0.0001. (**i**) *p* = 0.0008. **j**, *WT* or two different *Mre11*[ATLD/ATLD] MEF cell lines were transfected with 4 μg/mL Alexa488-ISD90 or 72 nM OG-hNCP for 60 min. Quantification of cells with colocalization for cGAS foci and cytosol DNA (ISD90 or OG-hNCP) is shown. Alexa488-ISD90 (WT vs ATLD #1; *p* = 0.0021, ATLD#2; *p* = 0.004). OG-hNCP (WT vs ATLD #1; *p* < 0.0001, ATLD#2; *p* < 0.0001). *n* = 3 independent biological experiments; 900 cells for each cell lines. **k**, **l**, qRT-PCR normalized gene expression for mouse *ISG IFIT1* and *CCL5*. (**k**) WT vs WT + ISD90; *p* = 0.0007, WT + ISD90 vs ATLD + ISD90; *p* = 0.0022. (**l**) WT vs WT + ISD90; *p* = 0. 0006, WT + ISD90 vs ATLD + ISD90; *p* = 0.0009. *n* = 3 independent biological experiments; 3 samples for each cell lines. Data are mean ± SEM. Unless otherwise specified, P values estimated using a two-tailed t-test. *, p < 0.05; **, p < 0.01; ***, p < 0.001; ****, p < 0.0001.

**a**

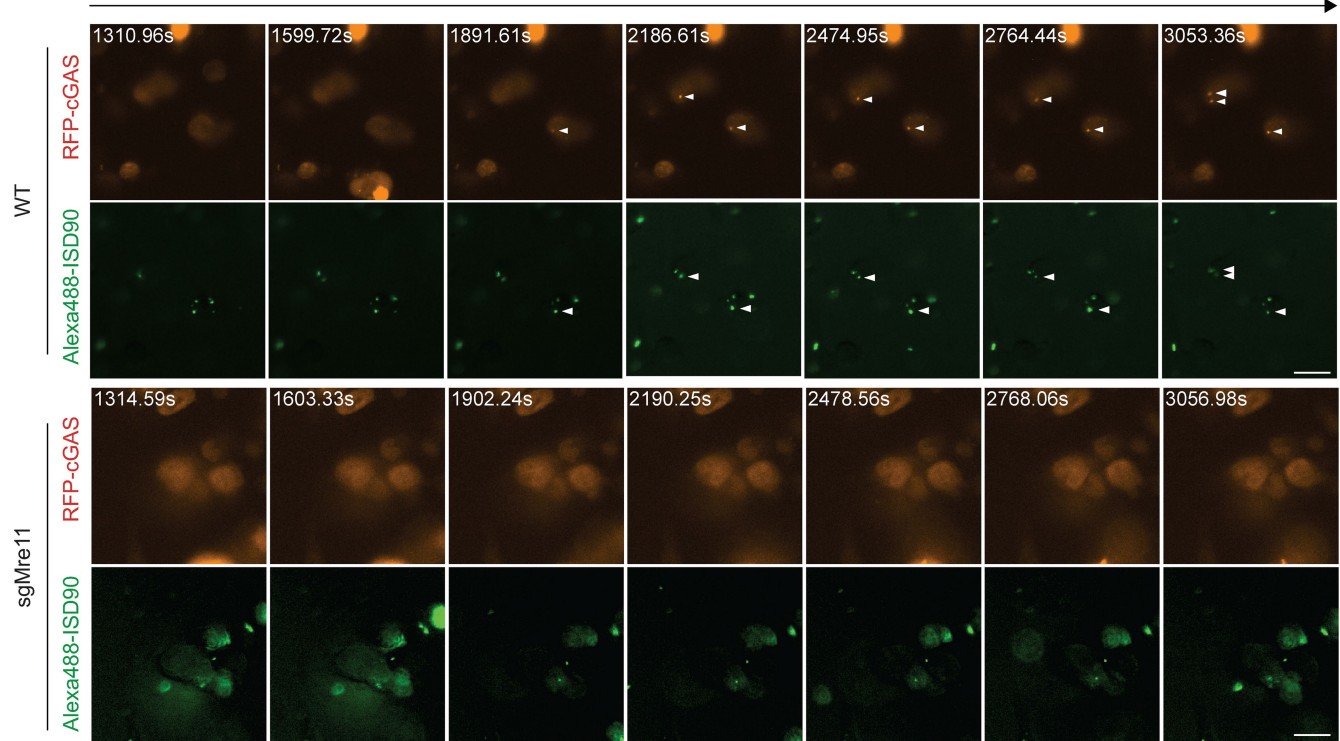

**Extended Data Fig. 4 | Time-lapse microscopy of cGAS recruitment to cytoplasmic dsDNA in WT and Mre11-mutant cells. a**, Time-lapse microscopy of WT and *sgMRE11* MDA-MB-231 cells expressing RFP-cGAS after transfection with 4 μg/mL Alexa 488-ISD90. Images were captured with a Nikon fluorescence microscope, and the time stamp is relative to transfection. White arrows indicate co-localization of RFP-cGAS foci and Alexa 488-ISD90 foci. Scale bar, 20 μm.

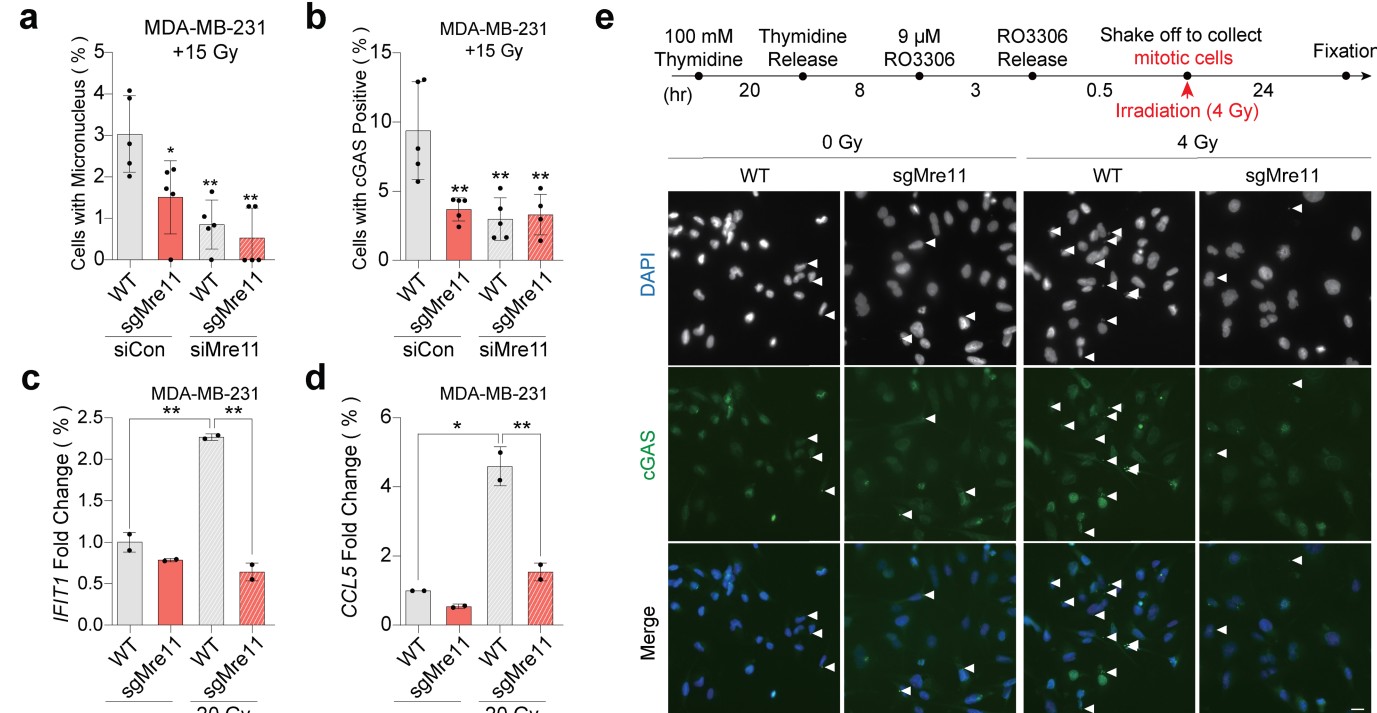

**Extended Data Fig. 5 | The translocation of cGAS to the micronuclei in mitotic MDA-MB-231 cells in response to ionizing radiation (IR) is dependent on Mre11. a, b** WT or *sgMRE11* MDA-MB-231 cells transfected with control siRNA (siControl) or Mre11-targeting siRNA (*siMRE11*) and analyzed 24 h after 15 Gy IR. Cells with micronuclei (a) or cGAS foci (b) were counted by ICC analysis. *n* = 2 independent experiments; 615 (WT + siControl), 387 (*sgMre11*+ siControl), 467 (WT + *siMre11*), 620 (*sgMre11* + *siMre11*) samples for each cell lines. (**a**) WT+siControl vs *sgMre11*+siControl; *p* = 0.029, WT+*siMRE11*; *p* = 0.0022, *sgMRE11*+*siMRE11*; *p* = 0.0014. (**b**) WT+siControl vs *sgMRE11*+siControl; *p* = 0.008, WT+*siMRE11*; *p* = 0.0058, *sgMRE11*+*siMRE11*; *p* = 0.0074. **c, d**, human *ISG* expression *IFIT1* (c) and *CCL5* (d) were observed in WT and *sgMRE11* MDA-

MB-231 cells 48 h after IR (20 Gy) by RT-qPCR. mRNA levels were normalized to β-actin mRNA levels. (**c**) WT vs WT + 20 Gy; *p* = 0.0121, WT + 20 Gy vs *sgMRE11* + 20 Gy; *p* = 0.0099. (**d**) WT vs WT + 20 Gy; *p* = 0.0048, WT + 20 Gy vs *sgMRE11*+ 20 Gy; *p* = 0.0025. *n* = 2 independent experiments; 3 samples for each cell lines. **e**, Mitotic cells of the indicated MDA-MB-231 cells were irradiated and then fixed 24 h after irradiation. Scale bar, 20 μm. Image quantification is shown in Fig. 3e. Data are mean ± SEM. Unless otherwise specified, statistical analyses were determined by one-way ANOVA followed by Sidak's multiple comparison post-test using a two-tailed test: *, p < 0.05; **, p < 0.01; ***, p < 0.001; ****, p < 0.0001.

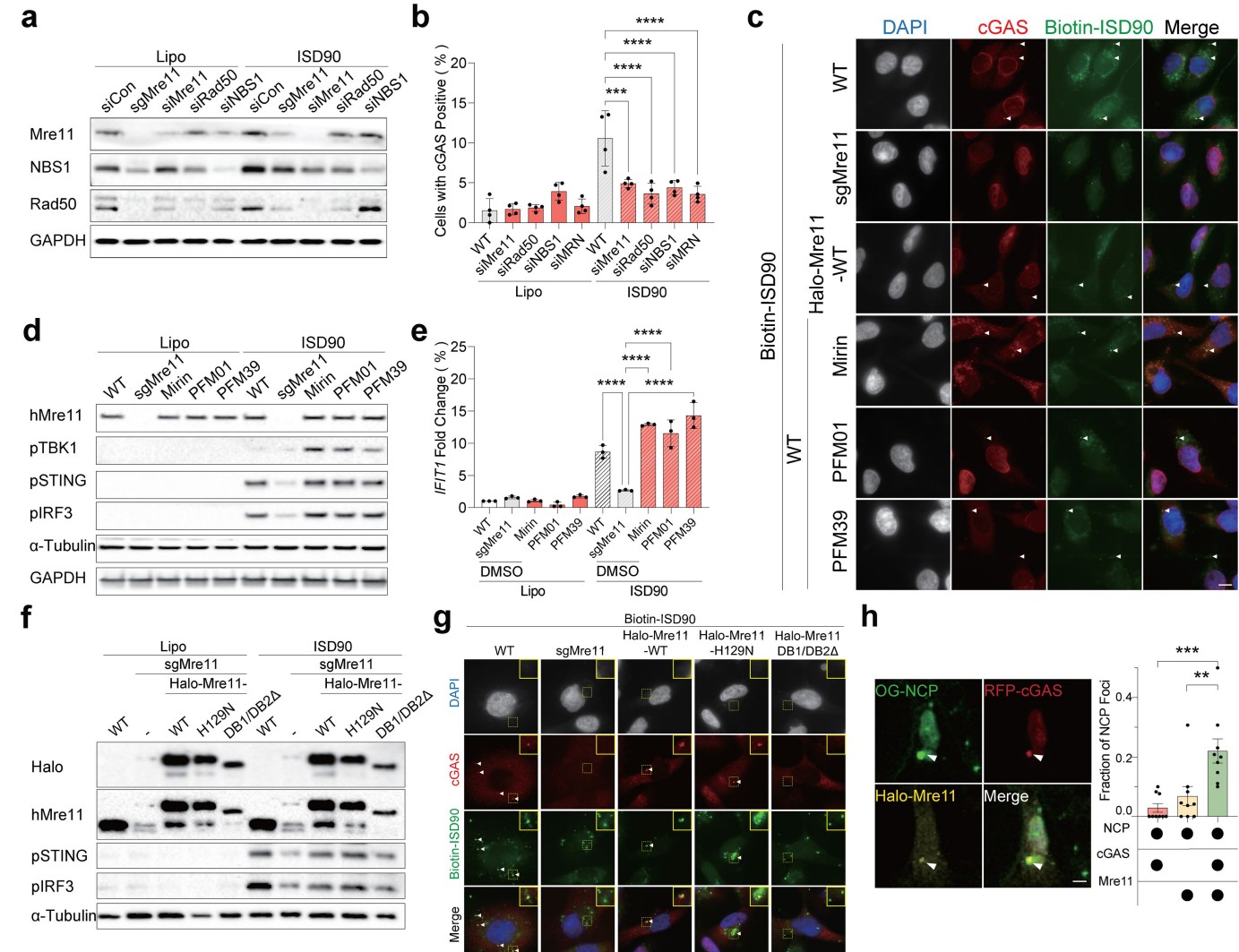

**Extended Data Fig. 6 | The DNA binding domain, but not the nuclease activity, of Mre11 is required to facilitate cGAS-dsDNA interaction. a, b**, MDA-MB-231 cells were transfected with the indicated siRNA and incubated for 48 h. Subsequently, cells were harvested 6 h (**a**) or 1 h (**b**) after transfection with 4 μg/mL ISD90. **a**, Western blot analysis of each cell line with the indicated antibodies. Mre11, Nbs1, Rad50 and GAPDH (a loading control). **b**, Quantification of cells positive for cGAS foci. (a) and (b) were double-checked using a blinded experiment. *n* = 2 independent biological experiments; 400 cells for each cell lines. WT + ISD90 vs *siMRE11* + ISD90; *p* < 0.0001, *siRad50, siNBS1* and *siMRN*; *p* = 0.0002. **c, d, e**, MDA-MB-231 cells were transfected with dsDNA, following treatment with indicated nuclease inhibitors (Mirin, PFM01 and PFM39) for 30 min. **c**, Representative images showing the localization of cGAS and Biotin-ISD90 were obtained 2 h after transfection with Biotin-ISD90. Scale bar, 10 μm. Image quantification is shown in Fig. 3f. **d**, Western blot analysis for cGAS-STING pathways. Cells were obtained 3 h after transfection of ISD90. **e**, Cells were obtained 6 h after transfection of ISD90. qRT-PCR normalized gene expression for human interferon-stimulated genes *IFIT1*. mRNA levels were normalized to human β-actin RNA levels. *n* = 3 independent biological

experiments; 3 samples for each cell lines. *p* < 0.0001. **f, g**, *sgMRE11* MDA-MB-231 stable cell lines were generated via retroviral infection, with each line expressing human Mre11 WT, H129N (nuclease dead mutant), and DB1/DB2 deletion (DNA binding mutant) in *sgMRE11* MDA-MB-231 cells. These cell lines were then transfected with dsDNA. **f**, Western blot analysis of Halo, human Mre11, pSTING, pIRF3 and α-tubulin in each MDA-MB-231 cell lines, demonstrating consistent findings across three independent replicates. **g**, Representative images showing the localization of cGAS and Biotin-ISD90 were obtained 2 h after transfection with Biotin-ISD90. Scale bar, 10 μm. Image quantification is shown in Fig. 3g. **h**, ICC 30 min after transfection with Oregon Green (OG)-NCP in MDA-MB-231 cells expressing RFP-cGAS and HaloTag-Mre11. Right panel, quantification of Mre11 and cGAS colocalization at cytoplasmic NCP foci. Scale bar, 10 μm. *n* = 3 independent biological experiments; 474 cells were analyzed for Fraction of NCP foci. NCP+cGAS vs NCP + cGAS+Mre11; *p* = 0.0006, NCP+Mre11 vs NCP+cGAS+Mre11; *p* = 0.0055. Data are mean ± SEM. Unless otherwise specified, statistical analyses were determined by one-way ANOVA followed by Sidak's multiple comparison post-test using a two-tailed test: *, p < 0.05; **, p < 0.01; ***, p < 0.001; ****, p < 0.0001.

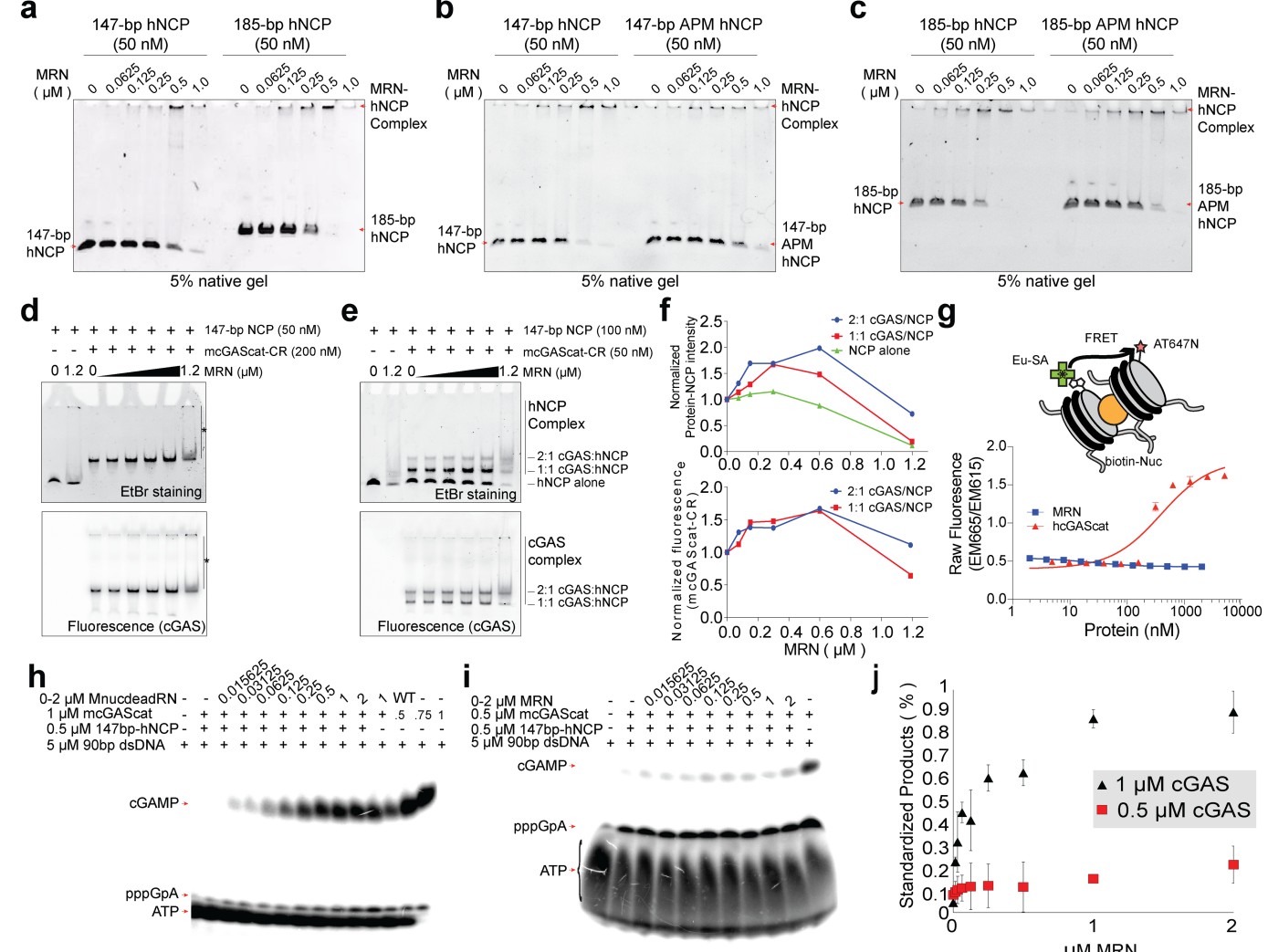

**Extended Data Fig. 7 | Biochemical analysis of purified MRN, Nucleosome Core Particles (NCP), and cGAS. a, b, c, d**, Electrophoretic mobility shift assays (EMSA) **a**, Native gel of nucleosomes containing 147 or 185 bp DNA in presence of increasing concentrations of MRN complex, stained with ethidium bromide. **b**, Native gel of 147 bp nucleosomes containing either wild type or acidic patch mutant (APM) histones in the presence of increasing concentrations of MRN complex, stained with ethidium bromide. **c**, Native gel as in panel (**b**) using 185 bp nucleosomes with 20 bp symmetric linker on each side of the nucleosome core. **d** and **e**, Native gel of 50 nM (**d**) or 100 nM (**e**) 147-bp hNCPs without or with 200 nM (**d**) or 50 nM (**e**) carboxyrhodamine-labelled mouse cGAS catalytic domain (mcGAScat-CR) and 0-1.2 μM MRN. The upper gel shows ethidium bromide staining of DNA, and the lower gel shows fluorescence signal from mcGAScat-CR. hNCP alone as well as 1:1 and 2:1 cGAS:hNCP complexes are indicated. (The results from the tests (**a**, **b**, **c**, **d**, **e**) demonstrate consistent findings across three independent replicates. **f**, Quantification of bands from

the corresponding gels (**e**). **g**, TR-FRET assay for detection of nucleosome stacking mediated by human cGAS catalytic domain (hcGAScat) or MRN complex. EU-SA, LANCE Eu-W1024 Streptavidin; AT647N, Atto647N; biotin-NUC, H2BK125C-biotin 147-bp nucleosomes. Data are mean ± s.d. **h, i**, Substrate, intermediate, and product are ATP, pppGpA, and cGAMP, respectively. **h**, 1 μM cGAS: 5 μM dsDNA, 0.5 μM hNCP, 0–2 μM MnucdeadRN (containing Mre11[H129N] nuclease deficient protein), demonstrating consistent findings across three independent replicates. **i**, 0.5 μM cGAS: 5 μM dsDNA, 0.5 μM hNCP, 0–2 μM MRN demonstrating consistent findings across two independent replicates. **j**, Quantification of standardized percentage of cGAMP product across three biological replicates demonstrates greater cGAS activity in the presence of increasing concentrations of MRN, but only when the cGAS:hNCP ratio is 2:1 versus 1:1. Shown are the mean value with error bars representing the standard deviation.

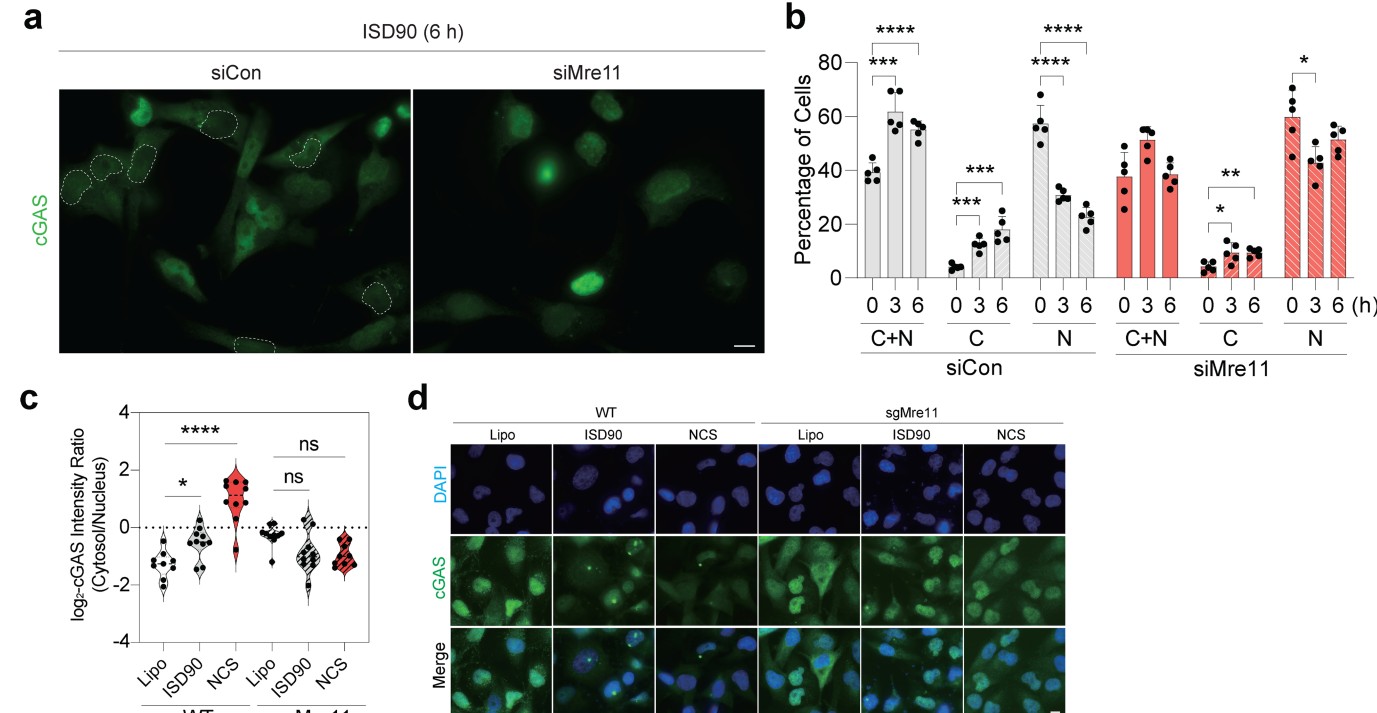

**Extended Data Fig. 8 | Mre11 facilitates cGAS cytoplasmic translocation in response to cytosolic DNA and DNA damage induction. a, b,** ICC of cGAS re-localization to cytoplasmic ISD90 in a subset of cells, **a,** 6 h after transfection in siControl versus *siMRE11* BJ-5ta cells (72 h after siRNA transfection). **b,** Percentage of cells with cGAS localization in both cytosol and nucleus (C + N; Cytosol + Nucleus [0.4 </= C/N ratio </= 1.25]), predominantly cytosolic (C: cytosol [C/N ratio > 1.25]), and predominantly nuclear (N: Nucleus [C/N ratio < 0.4]) 0, 3, and 6 h after 4 μg/mL ISD90 transfection in a series of images containing at least 10 evaluable cells each. *n* = 2 independent experiments; n = 500 cells for each cell condition. siControl (C + N) 0 h vs 3 h: *p* = 0.0002, 0 h vs 6 h: *p* < 0.0001; (C) 0 h vs 3 h: *p* = 0.0001, 0 h vs 6 h: *p* = 0.0002; (N) 0 h vs 3 h:

*p* < 0.0001, 0 h vs 6 h: *p* < 0.0001. *siMRE11* (C) 0 h vs 3 h: *p* = 0.0234, 0 h vs 6 h: *p* = 0.0012; (N) 0 h vs 3 h: *p* = 0.0116. **c, d,** Indicated MDA-MB-231 cells were either transfected with dsDNA or treated with NCS for 6 h. **c,** The Log$_2$-transformed cGAS localization ratio (Cytosol/Nucleus). *N* = 2 independent experiments; 10 (WT and *sgMRE11*) cells. WT + Lipo vs WT + ISD90; *p* = 0.034, WT + Lipo vs WT + NCS; *p* < 0.0001. **d,** Representative images of cGAS staining, demonstrating consistent findings across three independent replicates. Scale bar, 10μm. Data are mean ± SEM. Unless otherwise specified, statistical analyses were determined by one-way ANOVA followed by Sidak's multiple comparison post-test using a two-tailed test: *, p < 0.05; **, p < 0.01; ***, p < 0.001; ****, p < 0.0001.

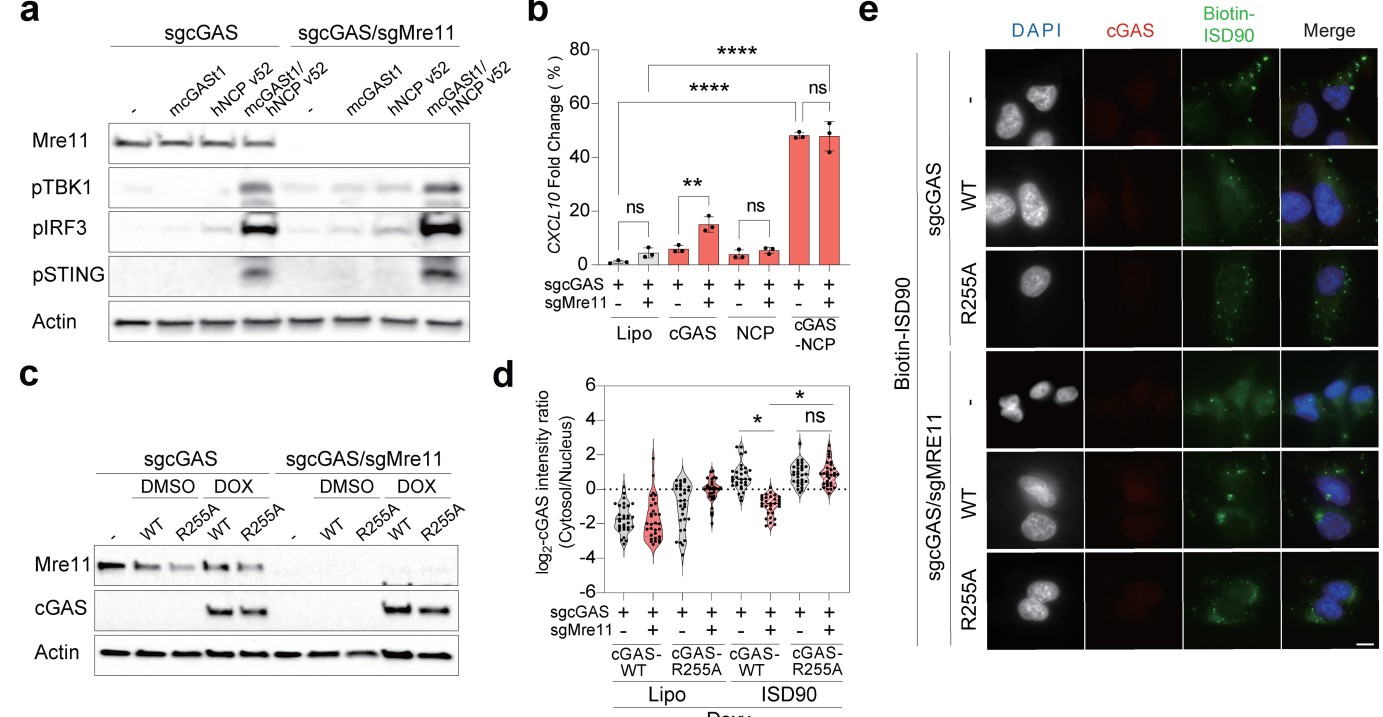

**Extended Data Fig. 9 | Mre11 is required to release cGAS from nucleosome tethering. a, b,** Human cGAS knock-out MDA-MB231 cell lines were made with CRISPR technology using multiguided sgRNA (Synthego). Then, MDA-MB-231 *sgcGAS* stable cell lines were transfected with mcGASt1, hNCP v52, or mcGASt1/ hNCP v52. **a,** After 2 h, cells were harvested for Western blot analysis to confirm human Mre11, pTBK, pSTING, and Actin, demonstrating consistent findings across two independent replicates. **b,** Six hours after transfection, qRT-PCR was performed to normalize gene expression levels for interferon-stimulated genes *CXCL10*, with mRNA levels normalized to β-actin mRNA levels. *n* = 3 independent experiments; 3 samples for each cell lines. *sgcGAS* + Lipo vs *sgcGAS* + cGAS-NCP; *p* < 0.0001, *sgcGAS/sgMRE11* + Lipo vs *sgcGAS/sgMRE11* + cGAS-NCP; *p* < 0.0001, *sgcGAS* + cGAS vs *sgcGAS/sgMRE11* + cGAS; *p* = 0.0067. **c, d,** PiggyBac Transposon system was used to deliver PB-cGAS-WT or PB-cGAS-R255A (an AP site binding mutant) to each cell line, followed by selection with puromycin for two days. After selection, cells were incubated

with doxycycline for 24 h, and then transfected with ISD90. **c,** Western blot analysis of human Mre11, cGAS and Actin 24 h after doxycycline treatment, demonstrating consistent findings across two independent replicates. **d,** The cGAS localization ratio (Nucleus/cytosol) 2 h after Biotin-ISD90 transfection. *n* = 2 independent experiments; 31 cells for each cell lines. *sgcGAS* + cGAS-WT + ISD90 vs *sgcGAS/sgMRE11* + cGAS-WT + ISD90; *p* = 0.0394, *sgcGAS/sgMRE11* + cGAS-WT + ISD90 vs s*sgcGAS/sgMRE11* + cGAS-R255A + ISD90; *p* = 0.022. **e,** Representative images showing the localization of cGAS and Biotin-ISD90 were obtained 2 h after transfection with Biotin-ISD90, demonstrating consistent findings across two independent replicates. Scale bar, 10 μm. Data are mean ± SEM. Unless otherwise specified, statistical analyses were determined by one-way ANOVA followed by Sidak's multiple comparison post-test using a two-tailed test: *, p < 0.05; **, p < 0.01; ***, p < 0.001; ****, p < 0.0001. n.s., not significant.

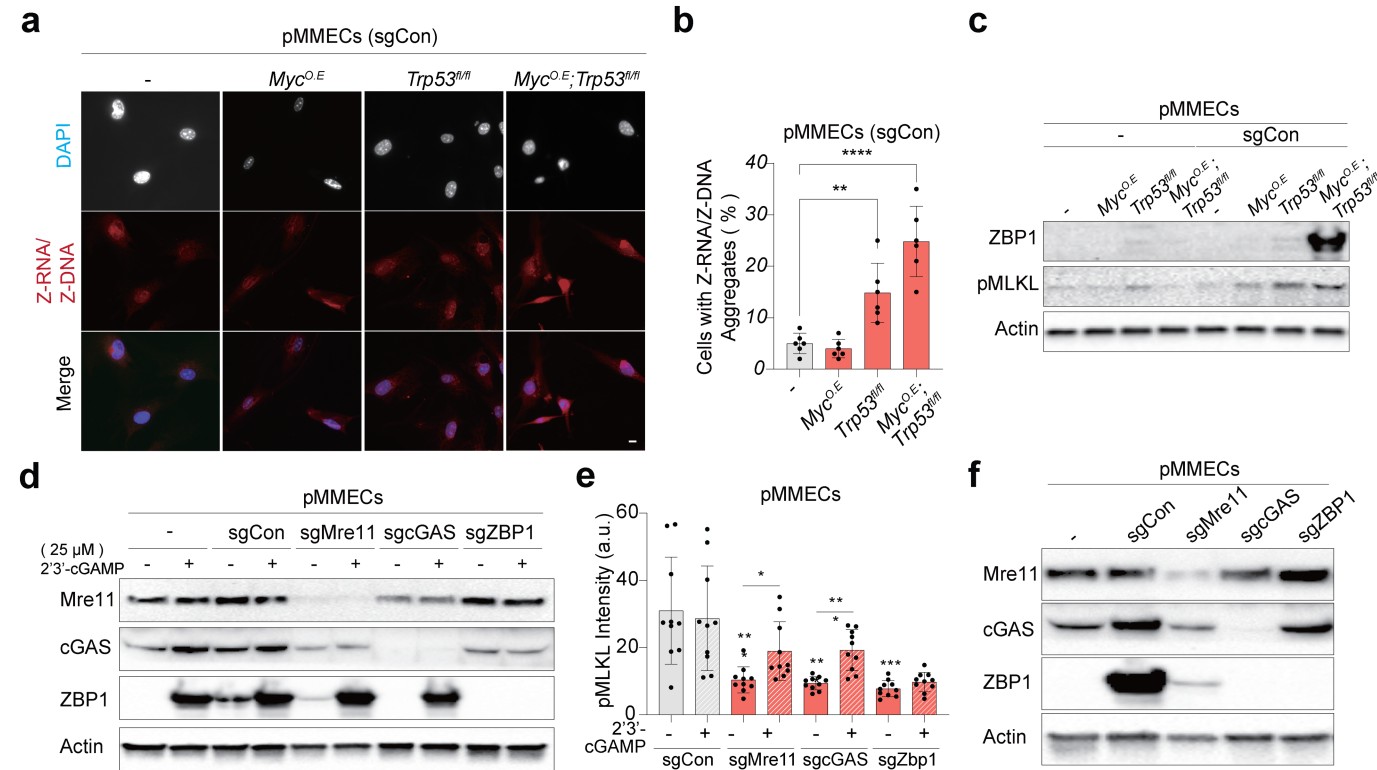

**Extended Data Fig. 10 | Myc overexpression and p53 depletion stimulate Z-RNA/Z-DNA accumulation and ZBP1-dependent necroptosis in pMMECs. a**, Representative images of Z-RNA/Z-DNA immunocytochemistry in sgControl pMMECs. Scale bar, 10 μm. **b**, Quantification of cells with Z-RNA/Z-DNA aggregation percentage. $n = 2$ independent experiments; 600 cells for each cell lines. NT vs $Trp53^{fl/fl}$; $p = 0.0043$, NT vs $Myc^{OE};Trp53^{fl/fl}$; $p < 0.0001$. **c**, Western blot confirmation of ZBP, pMLKL and Actin in pMMECs 8 days after sgControl lentiviral infection, demonstrating consistent findings across three independent replicates. **d**, Western blot analysis of untransduced or $Myc^{OE}p53^{-/-}$ pMMECs expressing the indicated sgRNA, 6 h after treatment with DMSO control or 25 μM 2'3'-cGAMP. **e**, ICC analysis for pMLKL intensity 6 h after treatment with DMSO control or 25 μM 2'3'-cGAMP in $Myc^{OE}p53^{-/-}$ pMMECs expressing the indicated sgRNA. $n = 2$ independent experiments; 10 cells for each cell lines. $sgMre11$ vs $sgMre11$+cGAMP; $p < 0.011$, $sgcGas$ vs $sgcGas$+cGAMP; $p = 0.0001$. **f**, Western blot confirming inhibition of Mre11, cGAS, and ZBP1 proteins in $Myc^{OE}p53^{-/-}$ pMMECs expressing the indicated sgRNA, in support of the tumorigenesis experiment shown in Fig. 4g. Data are mean ± SEM. Unless otherwise specified, statistical analyses were determined by one-way ANOVA followed by Sidak's multiple comparison post-test using a two-tailed test: *, p < 0.05; **, p < 0.01; ***, p < 0.001; ****, p < 0.0001.

# Reporting Summary

## Statistics

For all statistical analyses, confirm that the following items are present in the figure legend, table legend, main text, or Methods section.

| n/a | Confirmed | |
|---|---|---|
| ☐ | ☒ | The exact sample size (*n*) for each experimental group/condition, given as a discrete number and unit of measurement |
| ☐ | ☒ | A statement on whether measurements were taken from distinct samples or whether the same sample was measured repeatedly |
| ☐ | ☒ | The statistical test(s) used AND whether they are one- or two-sided *Only common tests should be described solely by name; describe more complex techniques in the Methods section.* |
| ☒ | ☐ | A description of all covariates tested |
| ☐ | ☒ | A description of any assumptions or corrections, such as tests of normality and adjustment for multiple comparisons |
| ☐ | ☒ | A full description of the statistical parameters including central tendency (e.g. means) or other basic estimates (e.g. regression coefficient) AND variation (e.g. standard deviation) or associated estimates of uncertainty (e.g. confidence intervals) |
| ☐ | ☒ | For null hypothesis testing, the test statistic (e.g. *F*, *t*, *r*) with confidence intervals, effect sizes, degrees of freedom and *P* value noted *Give P values as exact values whenever suitable.* |
| ☒ | ☐ | For Bayesian analysis, information on the choice of priors and Markov chain Monte Carlo settings |
| ☒ | ☐ | For hierarchical and complex designs, identification of the appropriate level for tests and full reporting of outcomes |
| ☒ | ☐ | Estimates of effect sizes (e.g. Cohen's *d*, Pearson's *r*), indicating how they were calculated |

*Our web collection on statistics for biologists contains articles on many of the points above.*

## Software and code

Policy information about availability of computer code

| | |
|---|---|
| Data collection | All microscopy images were collected on a NIS Elements AR software (Nikon), Aperio ScanScope AT2 (Leica Biosystem) or LSM software (Zeiss). Western blotting data was collected by ImageLab v6.0 (Bio-Rad). Realtime PCR was performed by QuantStudio 6 and 7 Flex Real-Time PCR System. Graph analysis : Graphpad Prism v7, v8 and v9. Cloning : SnapGene software v4.3.4. The processing of scRNA-seq data was primarily carried out using the Seurat package (version v3.1.2) in the R environment (version v3.6.0). Initial data filtering, normalization, scaling, and integration were performed using Seurat and scTransform. Clustering was executed using the Louvain-Jaccard method, and unique marker genes for each cluster were identified using Presto. Differential expression analyses were conducted using Seurat's FindMarkers function, and pathway enrichment analyses were done with gProfiler. |
| Data analysis | Statistical analyses for cell biological experiments were performed with GraphPad Prism (version 7, 8 and 9) as described in the Method. Software versions: BowTie2, version 2.3.4.1; Samtools, version 1.6.0; BedTools, version 2.26.0; Python, ≥v3.5; Ginkgo; Fiji ImageJ, version 1.53c; SnapGene Software, version 4.3.4; ZEN 2011 microscope software; NIS Elements AR software, version 4.50. |

For manuscripts utilizing custom algorithms or software that are central to the research but not yet described in published literature, software must be made available to editors and reviewers. We strongly encourage code deposition in a community repository (e.g. GitHub). See the Nature Portfolio guidelines for submitting code & software for further information.

## Data

Policy information about availability of data

All manuscripts must include a data availability statement. This statement should provide the following information, where applicable:
- Accession codes, unique identifiers, or web links for publicly available datasets
- A description of any restrictions on data availability
- For clinical datasets or third party data, please ensure that the statement adheres to our policy

In vivo CRISPR screen sequencing data and Single cell RNA sequencing data are deposited in the NCBI Gene Expression Omnibus. BioSample accession Number : SAMN36935033, SAMN36935034, NCBI BioProject URL : http://www.ncbi.nlm.nih.gov/bioproject/1004263.

## Research involving human participants, their data, or biological material

Policy information about studies with human participants or human data. See also policy information about sex, gender (identity/presentation), and sexual orientation and race, ethnicity and racism.

| | |
|---|---|
| Reporting on sex and gender | N/A |
| Reporting on race, ethnicity, or other socially relevant groupings | N/A |
| Population characteristics | N/A |
| Recruitment | N/A |
| Ethics oversight | N/A |

Note that full information on the approval of the study protocol must also be provided in the manuscript.

# Field-specific reporting

Please select the one below that is the best fit for your research. If you are not sure, read the appropriate sections before making your selection.

☒ Life sciences ☐ Behavioural & social sciences ☐ Ecological, evolutionary & environmental sciences

For a reference copy of the document with all sections, see nature.com/documents/nr-reporting-summary-flat.pdf

# Life sciences study design

All studies must disclose on these points even when the disclosure is negative.

| | |
|---|---|
| Sample size | Murine tumorigenesis studies utilized publicly available sample size estimation calculators to attain at least 80% power to detect a 30% reduction in tumor latency using a 2-tailed log rank test.  Sample sizes for in vitro experiments were determined empirically from previous experimental experience with similar assays, and/or from sizes generally employed in the field.  Specific sample sizes and the count of independent experiments conducted for each study can be found in the figures, their accompanying legends, or within the methods section. |
| Data exclusions | No data were excluded from analysis or reporting |
| Replication | All experiments were conducted independently as described in the figure legends or were biologically replicated at least 2-3 times. All p-values were derived from measurements obtained in experiments performed independently at least three times. Representative images or numerically similar results were used, which were replicated and obtained from at least more than two independent biological experiments. |
| Randomization | The quantification of Immunocytochemistry (ICC) data was obtained from random regions on the slide. For experiments not mentioned here, allocation was not random because data were compared or collected under identical conditions where randomness was not required. |
| Blinding | For practically feasible (ex_Extended Figure 6b), samples were blinded during analysis. However, the majority of experiments were not conducted under blinded conditions.  Most of the data were obtained from at least 2 to 3 biologically independent experiments, were replicated by different investigators, or were validated to produce the same results through different experimental methods. |

# Reporting for specific materials, systems and methods

We require information from authors about some types of materials, experimental systems and methods used in many studies. Here, indicate whether each material, system or method listed is relevant to your study. If you are not sure if a list item applies to your research, read the appropriate section before selecting a response.

## Materials & experimental systems

| n/a | Involved in the study |
|-----|----------------------|
| ☐ | ☒ Antibodies |
| ☐ | ☒ Eukaryotic cell lines |
| ☒ | ☐ Palaeontology and archaeology |
| ☐ | ☒ Animals and other organisms |
| ☒ | ☐ Clinical data |
| ☒ | ☐ Dual use research of concern |
| ☒ | ☐ Plants |

## Methods

| n/a | Involved in the study |
|-----|----------------------|
| ☒ | ☐ ChIP-seq |
| ☒ | ☐ Flow cytometry |
| ☒ | ☐ MRI-based neuroimaging |

## Antibodies

| | |
|---|---|
| Antibodies used | Antibody# Species# Company Cat# Application Dilution<br>Mre11, Rabbit monoclonal, Novus Biologicals, BN100-142, W.B 1:1,000<br>β-Actin (clone AC-15), Mouse, Sigma-Aldrich, A1978-100UL, WB 1:10,000<br>cGAS (D3O8O), Rabbit monoclonal, Cell Signaling Technology, 31659S, Tumor IF 1:4,000, W.B 1,000, ICC 1:500<br>cGAS (D1D3G), Rabbit monoclonal, Cell Signaling Technology, 15102S, W.B 1:,000, ICC 1:500<br>α Tubulin, Mouse monoclonal (B-7), Santa Cruz Biotechnology, SC-5286, W.B 1:1000<br>Phospho-Histone H2A.X (Ser139)(20E3), Rabbit monoclonal, Cell Signaling Technology, 9718S, Tumor IF 1:3,000<br>STING/TMEM173, Rabbit polyclonal, Novus Biologicals, NBP2-24683, W.B 1:1,000<br>IRF3 (Ser386)[EPR2346], Rabbit monoclonal, Abcam, ab76493, W.B 1,000<br>STING (Ser366)(D7C3S), Rabbit monoclonal, Cell Signaling Technology, 19781, W.B 1:1,000<br>TBK1/NAK (Ser172)(D52C2) , Rabbit monoclonal, Cell Signaling technology, 5483, W.B 1:1,000<br>GAPDH (G-9), Mouse monoclonal, Santa Cruz Biotechnology, sc-365062, W.B 1:1,000<br>Rad50, Rabbit polyclonal, Novus Biologicals, NBP2-20054, W.B 1:1,000<br>NBS1, Mouse monoclonal, Novus Biologicals, NBP2-20554, W.B 1:1,000<br>Z-DNA antibody, Novous Biologicals NB100-749 ICC 1:500<br>Horse polyclonal anti-Mouse IgG, HRP-linked antibody, Cell signaling Technology, 7076S, W.B 1:5000<br>Goat polyclonal anti-Rabbit IgG, HRP linked antibody, Cell signaling Technology, 7074S, W.B 1:5000<br>Goat polyclonal anti-Hamster IgG, HRP-linked antibody, Thermo Fisher Scientific, PA1-29626, W.B 1:5000<br>HaloTag® protein antibody, Mouse monoclonal, Promega, G9211, W.B 1,000, ICC 1:500<br>MLKL (S345)[EPR9515(2)], Rabbit monoclonal, Abcam, ab196436, W.B 1:1,000, ICC 1:500<br>MLKL antibody, Mouse monoclonal, Proteintech, 66675-1-Ig, W.B 1:1000<br>ZBP1(Zippy-1), Mouse monoclonal, AdipoGen, AG-20B-0010, W.B 1:1,000<br>ATM(D2E2), Rabbit monoclonal, Cell Signaling Technology, 2873S, W.B 1:1000<br>Goat anti-Rabbit IgG (H+L) Secondary Antibody, Alexa Fluor 488, Thermo Fisher Scientific, A11034, ICC : 1:500<br>Goat anti Rabbit IgG (H+L) Secondary Antibody, Alexa Fluor 594, Thermo Fisher Scientific, A11037, ICC : 1:500<br>F(ab')2-Goat anti-Rabbit IgG (H+L) Cross-Adsorbed Secondary Antibody, Alexa Fluor 633, Thermo Fisher Scientific, A11072, ICC : 1:500<br>Cy™3 AffiniPure Donkey Anti-Mouse IgG (H+L), Jackson ImmunoResearch, 715-165-151, ICC : 1:500<br>Cy™5 AffiniPure Goat Anti-Rabbit IgG (H+L), Jackson ImmunoResearch, 111-175-144, ICC : 1:500<br>Streptavidin, Alexa Fluor™ 488 Conjugate, Thermo Fisher Scientific, S32354, ICC : 1:500<br>Z-DNA, Sheep Polyclonal, Novous Biologicals, NB100-749, ICC 1:500<br>Donkey anti-Sheep IgG (H+L) Cross-Adsorbed Secondary Antibody, Alexa Fluor™ 594 Invitrogen™, Invitrogen, A-11016, ICC 1:500 |
| Validation | Appropriate positive and negative controls were included in the experimental design to confirm the antibodies were specific. Critical antibodies were validated by depletion or knockout of target genes using RNA interference or CRISPR/Cas9 editing (lentivirus), respectively. |

## Eukaryotic cell lines

Policy information about cell lines and Sex and Gender in Research

| | |
|---|---|
| Cell line source(s) | SF9 (Thermo Fisher Scientific, 11496015), WT MEFs (Gift from John Petrini, Ph.D), ATLD/ATLD MEFs (Gift from John Petrini, Ph.D), BJ-5ta (ATCC, CRL-4001), MDA-MB-231 cell lines (ATCC, CRM-HTB-26), Lucia ISG Cells (Invivogen, rawl-isg), HEK293T/17 (ATCC® CRL-11268), primary murine mammary epithelial cell lines (pMMECs) from female mice. As cited in method. |
| Authentication | We used DNA fingerprint analysis and PCR genotyping for authentication (MEFs, BJ-5ta, MDA-MB-231 cell lines). For cell lines not mentioned in this field such as the sf9, Lucia ISG Cells and pMMEC cell lines were not authenticated; pMMEC was directly isolated from from female mice and used without additional authentication, and sf9 and Lucia ISG Cells were delivered from the company and used immediately. |
| Mycoplasma contamination | All cells are routinely tested for and found to be mycoplasma free as described in the Methods. |
| Commonly misidentified lines<br>(See ICLAC register) | We never use misidentified cell lines. |

# Animals and other research organisms

Policy information about studies involving animals; ARRIVE guidelines recommended for reporting animal research, and Sex and Gender in Research

| Laboratory animals | Mus musculus; FVB.129P2-Trp53tm1Brn/Nci, C57BL/6N-Gt(ROSA)26Sortm13(CAG-MYC,-CD2*)Rsky/J, B6;129-Gt(ROSA)26Sortm1(CAG-cas9*,-EGFP)Fezh/J, NOD.129S7(B6)-Rag1tm1Mom/J. We have detailed the information on animals and housing conditions for mice in the Materials and Methods section. |
| --- | --- |
| Wild animals | Wild animals were not used in this study. |
| Reporting on sex | We exclusively utilized female mice for our study/experiments and outlined their respective ages for each experiment in the Methods section. |
| Field-collected samples | Samples collected in the field were not used in this study. |
| Ethics oversight | The UNC Institutional Animal Care and Use Committee (IACUC) approved guidance on our animal study protocol. IACUC ID : 22-163.0. The mice were euthanized in a humane manner in accordance with the guidelines set by IACUC when they reached a predetermined experimental endpoint. |

Note that full information on the approval of the study protocol must also be provided in the manuscript.

