## [Peer Review File · Nature]

Manuscript Title: Mre11 liberates cGAS from nucleosome sequestration during tumorigenesis

Reviewer Comments & Author Rebuttals

Reviewer Reports on the Initial Version:

Referees' comments:

Referee #1 (Remarks to the Author):

The manuscript by Cho et al describes a novel function for the MRN complex to directly activate cGAS. cGAS is a key sensor of foreign DNA, which is kept inactive through its interaction with nucleosomal acidic patch, which prevents its activation by self-DNA. Mechanisms leading to cGAS activation are not very clear. The authors propose that MRN, independently of its nuclease function, displaces cGAS from nucleosomal DNA, which would explain its activation. Although potentially important, I feel that the data in their current form do not support the key conclusions that MRN employs a novel, nuclease-independent mode to activate cGAS.

Specific comments

(1) MRE11 has been implicated in cGAS activation previously by multiple groups in response to both mitochondrial and nuclear DNA (e.g., PMID: 29670289, PMID: 34910513). In these contexts, cGAS activation was described to be dependent on the MRE11 nuclease activity (i.e., through the release of DNA fragments). Here, the authors conclude that MRE11 nuclease does not play a role, which is initially supported by a single panel (ED6d), which shows that mirin, a Mre11 nuclease inhibitor, does not change cGAS colocalization with NCP. At the same time, a large fraction of the data in Fig. 1-2 could be in principle explained by the nuclease-dependent mechanism. The authors should discuss the previous data/papers, and present multiple lines of evidence to demonstrate that the described mechanism is indeed different (i.e., additional cellular assays with mirin, MRE11 knockdown-rescue experiments with WT and nuclease-dead variants etc.). Mirin also inhibits ATM activation, so it may have broader effects. Therefore, other MRE11 nuclease inhibitors (PFM01/39) could also be used.

(2) The authors then present biochemical data in support of the nuclease-independent activation. The electrophoretic mobility shift assay in Fig 3e shows that MRN has a similar affinity to NCP without or with cGAS (i.e. no preference for cGAS-bound NCP). Also, the observed DNA affinity is very low: 1.2 micromolar MRN shows only very minimal binding activity. It seems unlikely that such a low affinity could efficiently displace cGAS.

The data in Fig 3f show that MN has a higher affinity to NCP. Is there a difference in binding of MN to DNA without or with NCP? RAD50 is almost an obligate partner of MRE11 (the homologs form a heterodimer already in bacteria), so it is not clear whether the DNA binding data without RAD50 are relevant.

(3) The data in ED7 are concerning, because there is a clear decrease of DNA (NCP) being detected after incubation with higher MRN concentrations (1 micromolar). Are these results reproducible? A potential nucleolytic activity (either intrinsic to MRE11, although this should not be present without manganese, or a contamination) could explain such a loss of signal. This could affect the interpretation of other data (such as the critical experiment in panel 3g). The authors should show the purified MRN (and MN) proteins used, and analyze the protein preparation for intrinsic and potential contaminating activities. The nuclease-dead version should also be used, in

particular for the experiment in 3g.

(4) Is there a direct interaction between MRN and cGAS?

Referee #2 (Remarks to the Author):

General Comment

The manuscript by Cho et al. entitled "Mre11 liberates cGAS from nucleosome sequestration during tumorigenesis" identified Mre11 as an essential component of cGAS activation that suppresses Myc-induced proliferation and inhibits tumorigenesis using an in vivo CRISPR screen. The authors mechanistically describe that the MRN complex (Mre11-Rad50-Nbn) displaces cGAS from AP sequestration for cGAS activation in response to oncogene-induced DNA damage. cGAS activation suppresses Myc-induced proliferation via ZBP1/RIPK3/MLKL-mediated necroptosis, ultimately suppressing tumorigenesis in breast cancer. While the authors have provided mechanistic links to several grey areas in the field, the depth of the mechanistic findings do not fully substantiate the conclusion, and thus, compromising the novelty and quality of the paper. Overall, this manuscript provides significant molecular insights, but the authors should provide more in-depth mechanistic studies using relevant assays to bring out their novelty and overall quality of the paper.

Major Comments:

1. Novelty: While the paper describes a mechanistic finding that links the previously unanswered questions, the details of the results do not add sufficient novelty to previous papers, such as:
a. Sci Adv. 2021;7(51):eabf9441 ("Mre11-dependent instability in mitochondrial DNA fork protection activates a cGAS immune signaling pathway")
b. Mol Cell 2013;52(3):353-65 ("The Mre11 complex suppresses oncogene-driven breast tumorigenesis and metastasis"), and
c. Cell Rep. 2020;30(5):1385-1399.e7 ("A p53-independent DNA damage response suppresses oncogenic proliferation and genome instability").

The authors should refer to these papers and discuss their manuscript's novelty in regards to the previous papers.

2. While this study shows Mre11 antagonizing Myc-induced proliferation, other studies have shown that Mre11 overexpression leads to breast cancer malignancy (Breast Cancer 2018;25:350-355 and JNCI 2012;104:1485-1502). The authors need to address the discrepancy between the tumor suppressive role of Mre11 in their model in comparison to the oncogenic role of Mre11 found in breast cancer patient tissue. To strengthen their findings, the authors may need to perform experiments using patient samples or better mimic the upregulation of Mre11 in breast cancer in their in vitro experimental models.

3. Figure 3a-c: cGAS, stimulated by ISD, is predominantly expressed in the cytosol (Science 2013;339(6121):786-91, Figure 6a). Based on Figure 3a sgMre11 data that shows no ISD90-cGAS foci in the cytosol, what may be the role of Mre11 in regulating cytosolic cGAS? How is this distinct from its role in the nucleus (Figure 2e)?

4. Figure 3f: It is unclear whether Mre11 directly displaces cGAS from the histone AP surface. The formation of a super-shifted ternary complex was apparent with Mre11-Nbn, which may also be interpreted as Mre11-Nbn enhancing cGAS binding to NCP. The direct effect of Mre11 on modulating cGAS interactions should be demonstrated.

5. Figure 3h: As implied by the title, the key point of the manuscript is that Mre11 liberates cGAS from nucleosome sequestration. Although the authors hypothesize that MRN binding may displace one of the cGAS molecule from the histone AP surface based on FRAP analysis, it does not fully demonstrate the direct displacement of cGAS from AP sequestration. The author should further demonstrate with additional experiment, such as a structural analysis, that reflects the results

before and after the displacement of cGAS.

6. Figure 4: The mechanism of how Mre11/cGAS/STING stimulates ZBP1/RIPK3/MLKL-dependent necroptosis is completely unclear. ZBP1 acts as a key mediator of interferon-induced necroptotic cell death. Additionally, a recent report described that Z-DNA binding of ZBP1 is critical for mediating tumor necroptosis (Nat. Comm. 2021;12(10):2666). It is critical for the authors to address the previously reported stimulators of ZBP1-mediated necroptosis and link how Mre11-mediated cGAS activation stimulates necroptosis, which is a major weakness in this manuscript.

7. Figure 4: Based on a recent study (Nat. Comm. 2021;12(10):2666), ZBP1 expression was highly elevated in human breast cancer tumors and was essential for tumor necroptosis and metastasis. How does the discrepancy between the previous report and Mre11-dependent cGAS activation in suppressing tumorigenesis via ZBP1-dependent necroptosis be explained? It seems highly essential to address this question as well as to detail the functional consequence of Mre11-dependent cGAS activation in inhibiting tumorigenesis.

Minor Comments:

1. Scale bars and labels are missing in many figures (ex. Figure 1c, 1j, 2c, 3a, 3d, 3h, and 4e).
2. Figure 1: The group has already published similar findings of Mre11 in suppressing breast cancer tumorigenesis and its role in cell cycle arrest. Thus, Figure 1 is not a novel finding and should be included in supplementary data.
3. Consistent use of abbreviations should be used throughout the manuscript.

Referee #3 (Remarks to the Author):

Cho, Gupta et al describe an interesting connection between Mre11 and c-Myc overexpression induced mammary tumorigenesis in mice. An in vivo CRISPR screen revealed that guide RNA (sgRNA) to Mre11 was the top hit for increased tumor growth in mice. Despite its known function in the DNA damage response, Mre11 deficiency resulted in less micronuclei in these cancer cells and reduced cGAS association. The authors provide more data that is consistent with a model that Mre11 alleviates nucleosome core particle inhibition of cGAS by displacing it from its interaction with the NCP acidic patch. Well controlled, compelling biochemical and imaging experiments nicely support this model with data that shows increased mobility of cGAS when Mre11 is present. The results are also commensurate with reduced cGAS dependent interferon stimulated gene expression. Additional data reveals reduced ZBP1 expression, which the authors show is cGAS-STING dependent and is involved in necroptosis. They hypothesize that Mre11 is necessary for cGAS dependent signaling through ZBP1 and other ISGs to slow mammary tumor growth.

cGAS-STING provides an important link to communication between chromosome instability and inflammatory signaling. The response is complicated by dichotomous interactions between cGAS and chromatin. While DNA binding is necessary for it to produce cGAMP, cGAS activity is inhibited by interaction with NCPs. Cho, Gupta and colleagues provide an interesting model to understand how this switch from NCP to DNA binding occurs. The work may also illuminate important concepts for how Mre11 serves as a tumor suppressor. These strengths are somewhat offset by models that require more supportive data as described below. While this may eventually be a substantial conceptual advance, additional experiments are required prior to publication.

1. Fig 1g-i. The authors show that sgMre11 modestly reduces tumor free survival and show faster growth upon initial implantation. Subsequent experiments lead them to conclude this is through reduced cGAS activation. It would be important to determine if sgGAS, sgSTING, or sgZBP1 produce similar findings to sgMre11. Although this is examined in pMEC proliferation assays, it is not in xenografts.

2. Fig 2 shows reduced micronuclei formation and increased proliferation in sgMre11 cells compared to controls. This suggests that Mre11 is responsible for mitotic errors, which could account for much of the increase in cGAS dependent inflammatory signaling. It is potentially confounding to their conclusion that Mre11 displacement of cGAS from nucleosomes is responsible for the reduced inflammatory signaling. This issue needs to be addressed.

3. Fig 2i – how do sgCgAS cells respond to cGAMP as shown by the elevated mIFIT1 mRNA? Wouldn't cGAS deficiency eliminate this response?

4. The authors perform biochemical and FRAP experiments in Fig 3 to support a model that Mre11 directly displaces cGAS interaction with the acidic patch in nucleosome core particles. This argues for a stoichiometric displacement to free cGAS for DNA binding in the cytoplasm. It is hard to understand this argument given that nucleosomes are in vast excess to either Mre11 or cGAS. How could Mre11 be required to displace cGAS from a vast pool of nucleosomes? Perhaps more plausible is displacement from NCPs in micronuclei, where there is far less chromatin. Some discussion of this point is warranted as alternative models are also possible. For example, is Mre11 activity in resection required to generate nucleosome free regions that could activate cGAS?

5. More defined Mre11 mutants that no longer displace cGAS would be valuable to test in cells. It would be interesting to determine if such mutants recapitulate the null phenotypes observed.

Author Rebuttals to Initial Comments:

Response to Reviewers

We are grateful to the reviewers for their constructive feedback, which has resulted in significant improvement of our manuscript. We sincerely apologize for the delay in resubmission, which was necessary to develop improved methods for the larger-scale production of the Mre11-Rad50-Nbn complex to clarify the biochemical mechanisms underlying cGAS activation. We now present definitive biochemical and cell biological evidence demonstrating the essential role for Mre11 in displacing cGAS from nucleosome sequestration. This displacement is required for cGAS mobilization from chromatin and its activation by double-stranded DNA (dsDNA). Furthermore, we show this function of Mre11 is tumor suppressive due to activation of ZBP1-dependent necroptosis. Suppression of this DNA damage-induced necroptosis pathway in patient cohorts of TNBC is associated with increased genome instability, immune suppression, and worse clinical outcomes. Thus, our study fills a major gap in our understanding of how chronically inhibited cGAS becomes activated upon DNA damage, with implications for DNA damage tolerance and immune modulation in cancer through stimulation of ZBP1-dependent necroptosis.

Referee #1 (Remarks to the Author):

The manuscript by Cho et al describes a novel function for the MRN complex to directly activate cGAS. cGAS is a key sensor of foreign DNA, which is kept inactive through its interaction with nucleosomal acidic patch, which prevents its activation by self-DNA. Mechanisms leading to cGAS activation are not very clear. The authors propose that MRN, independently of its nuclease function, displaces cGAS from nucleosomal DNA, which would explain its activation. Although potentially important, I feel that the data in their current form do not support the key conclusions that MRN employs a novel, nuclease-independent mode to activate cGAS.

Thank you for the constructive comments. Our revised manuscript provides additional data to clarify a novel nuclease-independent mechanism for Mre11 activation of cGAS through release from nucleosome inhibition. We have also provided more robust biochemical evidence to provide molecular insight into how MRN displaces cGAS from high-affinity interactions with the NCP acidic patch.

Specific comments

(1) MRE11 has been implicated in cGAS activation previously by multiple groups in response to both mitochondrial and nuclear DNA (e.g., PMID: 29670289, PMID: 34910513). In these contexts, cGAS activation was described to be dependent on the MRE11 nuclease activity (i.e., through the release of DNA fragments). Here, the authors conclude that MRE11 nuclease does not play a role, which is initially supported by a single panel (ED6d), which shows that mirin, a Mre11 nuclease inhibitor, does not change cGAS colocalization with NCP. At the same time, a large fraction of the data in Fig. 1-2 could be in principle explained by the nuclease-dependent mechanism. The authors should discuss the previous data/papers, and present multiple lines of evidence to demonstrate that the described mechanism is indeed different (i.e., additional cellular assays with mirin, MRE11 knockdown-rescue experiments with WT and nuclease-dead variants etc.). Mirin also inhibits ATM activation, so it may have broader effects. Therefore,

other MRE11 nuclease inhibitors (PFM01/39) could also be used.

Thank you for the insightful comments. We agree that previous studies have implicated Mre11 nuclease activity in cGAS activation when canonical DNA repair pathways are disrupted (e.g., Fanconi Anemia pathway or SAMHD1). Under these settings, Mre11 nuclease activity can degrade stalled replication forks into byproducts that are detected by cGAS. Our study adds a new mechanism of direct modulation of cGAS by Mre11 that is independent of its nuclease activity and relevant in cells without any other DNA repair deficiency. Our claim is now supported by three independent lines of evidence in the revised manuscript. First, cGAS activation by cytosolic DNA is insensitive to three different Mre11 nuclease inhibitors (**Figure 3f, Extended Data Figure 6c-e**), yet sensitive to MRN knockdown or protein destabilizing hypomorphic Mre11 mutations (**Figure 3a-d, Extended Data Figures 3, 4, 5, and 6a-b**). Second, cGAS responsiveness to cytosolic DNA can be restored by reconstituting Mre11 hypomorphic cells with nuclease-deficient, but not DNA binding domain-deficient, Mre11 constructs (**Figure 3g, Extended Data Figure f-g**). Third, we demonstrate that biochemically purified MRN comprised of nuclease-deficient Mre11 is also capable of releasing cGAS from nucleosome core particle-mediated inhibition (**Extended Data Figure 7h,j**). Collectively, we assert that both direct and indirect effects of Mre11 on cGAS activation are consistent with the overarching conclusion of our paper that Mre11 is critical for activating cGAS in response to diverse forms of DNA damage.

(2) The authors then present biochemical data in support of the nuclease-independent activation. The electrophoretic mobility shift assay in Fig 3e shows that MRN has a similar affinity to NCP without or with cGAS (i.e. no preference for cGAS-bound NCP). Also, the observed DNA affinity is very low: 1.2 micromolar MRN shows only very minimal binding activity. It seems unlikely that such a low affinity could efficiently displace cGAS. The data in Fig 3f show that MN has a higher affinity to NCP. Is there a difference in binding of MN to DNA without or with NCP? RAD50 is almost an obligate partner of MRE11 (the homologs form a heterodimer already in bacteria), so it is not clear whether the DNA binding data without RAD50 are relevant.

To address limitations of the EMSA data, the revised manuscript utilizes time-resolved fluorescence resonance energy transfer to measure cGAS interactions with the nucleosome core particle (NCP) acidic patch. We show MRN displaces ~50% of the cGAS from NCP interaction (**Figure 3h**). These effects are observed with MRN concentrations as low as ~20-100nM that correspond well with MRN dose response effects in the cGAS enzymatic assay (see **Figure 3j**). MRN also disrupts cGAS-mediated nucleosome “stacking”, suggesting that MRN interactions with NCP fragments alters higher-order interactions between cGAS and NCPs (**Figure 3i**). The finding that MRN displaces 50% of NCP-bound cGAS is consistent with our EMSA observations, where a tripartite complex of MRN-NCP-cGAS only allows for only one rather than two cGAS molecules. We believe these effects are not due to specific interactions between MRN and the histone AP, but rather through dsDNA binding by MRN through the Mre11 DNA binding domains, which are required for cGAS activation (**Figure 3j**). As noted by the reviewer, the interactions between the 147bp NCP and MRN seem to be relatively labile and not well resolved by EMSA until higher MRN concentrations. However, the 147bp NCP was used in the TR-FRET assays, demonstrating that long dsDNA tails are not required for MRN-dependent cGAS displacement. Of note, the 187bp NCP, which has a longer dsDNA tail, exhibits higher

affinity for MRN by EMSA gel shift (**Extended Data Figure 7c**). We also agree with the reviewer regarding the questionable evolutionary/physiological relevance of MN experiments and have removed these data from the revised manuscript.

(3) The data in ED7 are concerning, because there is a clear decrease of DNA (NCP) being detected after incubation with higher MRN concentrations (1 micromolar). Are these results reproducible? A potential nucleolytic activity (either intrinsic to MRE11, although this should not be present without manganese, or a contamination) could explain such a loss of signal. This could affect the interpretation of other data (such as the critical experiment in panel 3g). The authors should show the purified MRN (and MN) proteins used, and analyze the protein preparation for intrinsic and potential contaminating activities. The nuclease-dead version should also be used, in particular for the experiment in 3g.

We have not observed any nuclease activity in our MRN or NCP preparations (**Response Fig. 1**). We attribute the reduction in DNA signal at higher MRN concentrations in the EMSA experiments to formation of higher order complexes that were not entering the gel. We do not believe Mre11 nuclease activity is contributing to the rescue of cGAS enzymatic activity despite presence of NCP, as reconstitution occurs even when using MnuceadRN complex (**Extended Data Figure 7h**)

(4) Is there a direct interaction between MRN and cGAS?

We analyzed gel filtration of cGAS with or without MRN and observe no detectable shift of cGAS into the MRN fractions, and thus interpret no measurable interaction as purified proteins, in the absence of DNA (**Response Fig. 2**). Due to space limitations, we have not included this data in the revised manuscript.

Referee #2 (Remarks to the Author):

General Comment

The manuscript by Cho et al. entitled “Mre11 liberates cGAS from nucleosome sequestration during tumorigenesis” identified Mre11 as an essential component of cGAS activation that suppresses Myc-induced proliferation and inhibits tumorigenesis using an in vivo CRISPR screen. The authors mechanistically describe that the MRN complex (Mre11-Rad50-Nbn) displaces cGAS from AP sequestration for cGAS activation in response to oncogene-induced DNA damage. cGAS activation suppresses Myc-induced proliferation via ZBP1/RIPK3/MLKL-mediated necroptosis, ultimately suppressing tumorigenesis in breast cancer. While the authors have provided mechanistic links to several grey areas in the field, the depth of the mechanistic findings do not fully substantiate the conclusion, and thus, compromising the novelty and quality of the paper. Overall, this manuscript provides significant molecular insights, but the authors should provide more in-depth mechanistic studies using relevant assays to bring out their novelty and overall quality of the paper.

Major Comments:

1. Novelty: While the paper describes a mechanistic finding that links the previously unanswered questions, the details of the results do not add sufficient novelty to previous papers, such as:

a. Sci Adv. 2021;7(51):eabf9441 (“Mre11-dependent instability in mitochondrial DNA fork protection activates a cGAS immune signaling pathway”)

b. Mol Cell 2013;52(3):353-65 (“The Mre11 complex suppresses oncogene-driven breast tumorigenesis and metastasis”), and

c. Cell Rep. 2020;30(5):1385-1399.e7 (“A p53-independent DNA damage response suppresses oncogenic proliferation and genome instability”).
The authors should refer to these papers and discuss their manuscript’s novelty in regards to the previous papers.

We have updated the introduction and discussion to provide more context regarding how the present study provides substantial new insight into how Mre11 suppresses tumorigenesis through direct regulation of cGAS-dependent necroptosis induction.

2. While this study shows Mre11 antagonizing Myc-induced proliferation, other studies have shown that Mre11 overexpression leads to breast cancer malignancy (Breast Cancer 2018;25:350-355 and JNCI 2012;104:1485-1502). The authors need to address the discrepancy between the tumor suppressive role of Mre11 in their model in comparison to the oncogenic role of Mre11 found in breast cancer patient tissue. To strengthen their findings, the authors may need to perform experiments using patient samples or better mimic the upregulation of Mre11 in breast cancer in their in vitro experimental models.

There are multiple pathways to evade tumor suppression, so we would not expect Mre11 to be directly inactivated in all breast cancers. Mre11 is known to have essential functions in S phase¹⁻⁴ and its transcript levels are highest in S phase (Cyclebase 3.0, see **Response Fig. 3**). Thus, Mre11 overexpression in some breast cancers may be a byproduct of higher proliferative activity. We and others have shown that 10-30% of breast cancers demonstrate abnormally reduced levels of Mre11/Nbn/Rad50⁵⁻⁷. Mre11 complex downregulation is most prominent in triple negative breast cancers that exhibit the highest chromosomal instability and proliferation

rates. Our findings using a clinically representative immune competent model of TNBC driven by Myc overexpression and p53 deficiency indicate that Mre11 is a bona fide tumor suppressor. Possible roles for Mre11 overexpression in tumorigenesis are not addressed in our study, but could be of future interest in tumors that inactivate cGAS or ZBP1-dependent necroptosis while retaining intact Mre11.

3. Figure 3a-c: cGAS, stimulated by ISD, is predominantly expressed in the cytosol (Science 2013;339(6121):786-91, Figure 6a). Based on Figure 3a sgMre11 data that shows no ISD90-cGAS foci in the cytosol, what may be the role of Mre11 in regulating cytosolic cGAS? How is this distinct from its role in the nucleus (Figure 2e)?

When inactive, cGAS is predominantly in the nucleus and bound to chromatin through a high affinity interaction with the nucleosome acidic patch that occludes its DNA binding domain⁸⁻¹². How cGAS transitions from this inactive state in the nucleus to an activated state in the cytosol was previously unknown. We now present additional evidence to support Mre11 as a critical regulator of this transition:

- 1) Mre11 is required for DNA damage-induced cGAS mobilization from chromatin (**Fig. 3k-l**)
- 2) Mre11 is required for cytosolic relocation of cGAS after DNA damage (**Extended Data Fig. 8a-d**)
- 3) cGAS transfection into the cytosol of Mre11 deficient cells restores dsDNA-induced cGAS activation (**Extended Data Fig. 9a-b**)
- 4) Expression of a cGAS mutant that is unable to bind nucleosomes no longer requires Mre11 for activation (**Fig. 3m-n, Extended Data Fig. 9c-e**).

Collectively, the evidence reveals Mre11 as a critical mediator of cGAS release from nucleosomes that links DNA damage to innate immune activation. We also note that the original Fig. 2e had a non-representative image that appeared to show nuclear cGAS foci in Myc^{OE}p53^{-/-} pMECs, which has now been updated with a more representative image of cGAS+ micronuclei.

4. Figure 3f: It is unclear whether Mre11 directly displaces cGAS from the histone AP surface. The formation of a super-shifted ternary complex was apparent with Mre11-Nbn, which may also be interpreted as Mre11-Nbn enhancing cGAS binding to NCP. The direct effect of Mre11 on modulating cGAS interactions should be demonstrated.

We agree that the EMSA data does not demonstrate a role for Mre11 in disrupting cGAS-nucleosome interactions. See below for new data that addresses this key issue.

5. Figure 3h: As implied by the title, the key point of the manuscript is that Mre11 liberates cGAS from nucleosome sequestration. Although the authors hypothesize that MRN binding may displace one of the cGAS molecule from the histone AP surface based on FRAP analysis, it does not fully demonstrate the direct displacement of cGAS from AP sequestration. The author should further demonstrate with additional experiment, such as a structural analysis, that reflects the results before and after the displacement of cGAS.

The revised manuscript includes Time-Resolved Fluorescence Resonance Energy Transfer (TR-FRET) data to directly measure cGAS-Nucleosome AP interactions. We now show that MRN stimulates 50% release of cGAS from nucleosome binding, and also disrupts cGAS-mediated nucleosome stacking (**Fig. 3h-i**). The MRN concentration at which this effect is seen (~20-50nM) corresponds to the concentrations at which MRN is able to release cGAS from NCP-mediated inhibition of its enzymatic activity (**Fig. 3j**). Resolving the structure of the MRN-NCP complex is not trivial because it may involve higher-order oligomerization, and indeed efforts to examine the complexes by cryogenic electron microscopy have not been successful to date.

6. Figure 4: The mechanism of how Mre11/cGAS/STING stimulates ZBP1/RIPK3/MLKL-dependent necroptosis is completely unclear. ZBP1 acts as a key mediator of interferon-induced necroptotic cell death. Additionally, a recent report described that Z-DNA binding of ZBP1 is critical for mediating tumor necroptosis (Nat. Comm. 2021;12(10):2666). It is critical for the authors to address the previously reported stimulators of ZBP1-mediated necroptosis and link how Mre11-mediated cGAS activation stimulates necroptosis, which is a major weakness in this manuscript.

The revised manuscript demonstrates that cGAMP, the second messenger produced by cGAS, is sufficient to stimulate ZBP1 expression (**Extended Data Fig. 10d-e**), which is consistent with its known status as an interferon-stimulated gene. ZBP1 requires Z-DNA/RNA binding for activation of RIPK3/p-MLKL-dependent necroptosis in both tumor and infection contexts¹³. In the revised manuscript, we show that Myc overexpression and p53 deficiency induce Z-DNA/RNA (**Extended Data Fig. 10a-c**). We thus propose that the combined effect of Mre11/cGAS on ZBP1 induction in conjunction with elevated Z-RNA/Z-DNA during oncogenic stress results in potent engagement of necroptosis. Our revised manuscript includes discussion of how our findings complement two recently published studies that also demonstrate ZBP1 activation as an important effector of cGAS-dependent cell death in response to telomere crisis and adriamycin-associated cardiotoxicity^{14,15}.

7. Figure 4: Based on a recent study (Nat. Comm. 2021;12(10):2666), ZBP1 expression was highly elevated in human breast cancer tumors and was essential for tumor necroptosis and metastasis. How does the discrepancy between the previous report and Mre11-dependent cGAS activation in suppressing tumorigenesis via ZBP1-dependent necroptosis be explained? It seems highly essential to address this question as well as to detail the functional consequence of Mre11-dependent cGAS activation in inhibiting tumorigenesis.

Baik et al use an established breast cancer cell line from the MMTV-Myc-VEGF bitransgenic mouse model to show that ZBP1 contributes to tumor necrosis and in this cell line model

promotes metastatic progression. The finding that ZBP1 is a driver of metastatic progression was not validated in any additional cancer models. Transplantable cell line models may have secondary alterations that impact observed phenotypes. In the Baik et al study, the immune reaction to the tumor is not characterized, and a possible lack of an immune response in their model may partly explain why necroptosis activation does not suppress both tumor and metastasis growth. The model that they used may also be hypervascular due to VEGF overexpression, and the generalizability of their findings are certainly not yet known. In contrast, we now show in vivo data using an tumorigenesis model that ZBP1/MLKL suppresses Myc/p53-induced breast tumorigenesis (**Fig. 4g**). Our new data that Myc overexpression/p53 deficiency induces Z-DNA/Z-RNA and ZBP1/MLKL activation (**Extended Data Fig. 10a-c**) also suggests that its tumor suppressive function may be most relevant in a background of p53 deficiency, whereas the model used in Baik et al does not incorporate a p53 mutation. Clinically, we also observe greater ZBP1 expression in p53-mutant breast cancers (**Response Fig. 4**). Thus, the data in Baik et al showing increased ZBP1 expression in patient breast tumors presenting at advanced stages (that are more likely to be p53 deficient) may be confounded by this genetic association. Our data in **Fig. 4j** suggests that within the TNBC molecular subtype (most of which are p53-deficient), increased expression of necroptosis pathway genes is associated with improved survival and increased immune infiltration. Additional work is certainly needed to clarify the context-dependent effects of ZBP1 and MLKL during tumorigenesis, which are likely more complex than we currently appreciate. However, we believe that our study provides robust evidence that Mre11/cGAS/ZBP1 constitutes an important tumor suppressor pathway that restrains p53-deficient breast tumorigenesis.

Minor Comments:

1. Scale bars and labels are missing in many figures (ex. Figure 1c, 1j, 2c, 3a, 3d, 3h, and 4e).

Thank you for noting this – we have corrected this in the revised manuscript.

2. Figure 1: The group has already published similar findings of Mre11 in suppressing breast cancer tumorigenesis and its role in cell cycle arrest. Thus, Figure 1 is not a novel finding and should be included in supplementary data.

This study is the first to conduct an in vivo CRISPR breast tumorigenesis screen that simultaneously evaluates the tumor suppressive activity of ~300 DDR genes in a background of Myc overexpression and p53 deficiency. The observation that Mre11 was the top hit was highly unexpected, and a novel discovery of this study. These data are not redundant with our prior findings of single-gene analysis of Mre11 in Her2/Neu-driven and Rb/p53-deficient models. Our study provides a mechanistic explanation for the potent tumor suppressive functions of Mre11 –

through activation of innate immune signaling in response to oncogene-induced DNA damage. Since the mechanistic studies follow from the novel *in vivo* DDR CRISPR screen results in Figure 1, we respectfully prefer to keep this in the main portion of the manuscript.

3. Consistent use of abbreviations should be used throughout the manuscript.

Thank you for this comment. We have reviewed our use of abbreviations for consistency in the revised manuscript.

Referee #3 (Remarks to the Author):

Cho, Gupta et al describe an interesting connection between Mre11 and c-Myc overexpression induced mammary tumorigenesis in mice. An *in vivo* CRISPR screen revealed that guide RNA (sgRNA) to Mre11 was the top hit for increased tumor growth in mice. Despite its known function in the DNA damage response, Mre11 deficiency resulted in less micronuclei in these cancer cells and reduced cGAS association. The authors provide more data that is consistent with a model that Mre11 alleviates nucleosome core particle inhibition of cGAS by displacing it from its interaction with the NCP acidic patch. Well controlled, compelling biochemical and imaging experiments nicely support this model with data that shows increased mobility of cGAS when Mre11 is present. The results are also commensurate with reduced cGAS dependent interferon stimulated gene expression. Additional data reveals reduced ZBP1 expression, which the authors show is cGAS-STING dependent and is involved in necroptosis. They hypothesize that Mre11 is necessary for cGAS dependent signaling through ZBP1 and other ISGs to slow mammary tumor growth.

cGAS-STING provides an important link to communication between chromosome instability and inflammatory signaling. The response is complicated by dichotomous interactions between cGAS and chromatin. While DNA binding is necessary for it to produce cGAMP, cGAS activity is inhibited by interaction with NCPs. Cho, Gupta and colleagues provide an interesting model to understand how this switch from NCP to DNA binding occurs. The work may also illuminate important concepts for how Mre11 serves as a tumor suppressor. These strengths are somewhat offset by models that require more supportive data as described below. While this may eventually be a substantial conceptual advance, additional experiments are required prior to publication.

1. Fig 1g-i. The authors show that sgMre11 modestly reduces tumor free survival and show faster growth upon initial implantation. Subsequent experiments lead them to conclude this is through reduced cGAS activation. It would be important to determine if sgcGAS, sgSTING, or sgZBP1 produce similar findings to sgMre11. Although this is examined in pMEC proliferation assays, it is not in xenografts.

The revised manuscript includes *in vivo* evidence that sgcGAS, sgZBP1, and sgMLKL accelerate *in vivo* breast tumorigenesis, similarly to sgMre11 (**Fig. 4g, Extended Data Fig. 10f**). We are also conducting an in-depth analysis of the immunological effects of disrupting these pathways at baseline and in response to DNA damage. However, these studies are beyond the scope of the present manuscript.

2. Fig 2 shows reduced micronuclei formation and increased proliferation in sgMre11 cells compared to controls. This suggests that Mre11 is responsible for mitotic errors, which could account for much of the increase in cGAS dependent inflammatory signaling. It is potentially confounding to their conclusion that Mre11 displacement of cGAS from nucleosomes is responsible for the reduced inflammatory signaling. This issue needs to be addressed.

To address this potential concern, we irradiated Mre11 WT and mutant cells while arrested in mitosis, to ensure a comparable burden of mitotic DNA fragments were produced, and to evaluate their potential to generate cGAS positive micronuclei. In revised **Fig. 3e**, we show that Mre11 remains essential for the formation of cGAS positive micronuclei in this setting as well. Additionally, we have observed higher levels of unrepaired DSBs in Mre11 mutant cells, consistent with the well-known roles for Mre11 in DSB repair. Thus, we favor the interpretation that the reduced abundance of classical-appearing micronuclei in sgMre11 cells is not due to a reduction in DNA fragments but rather due to a defect in Mre11-dependent damage clustering, which may have a role in micronuclei maturation. In data not shown, we observe a higher frequency of nuclear “blebs” in Mre11 mutant cells, which may reflect a partial defect in MN biogenesis. Nonetheless, the fact that cells expressing a mutant cGAS that does not bind nucleosomes is able to become fully activated in the absence of Mre11 (revised **Fig. 3m-n**) indicates that this partial abnormality in MN morphology does not interfere with cGAS activation.

3. Fig 2i – how do sgcGAS cells respond to cGAMP as shown by the elevated mIFIT1 mRNA? Wouldn't cGAS deficiency eliminate this response?

cGAS knockout cells still express STING and all downstream factors required for ISG expression. So, while they are impaired in endogenous cGAMP production, they remain responsive to exogenous cGAMP treatments that directly bind to and activate STING.

4. The authors perform biochemical and FRAP experiments in Fig 3 to support a model that Mre11 directly displaces cGAS interaction with the acidic patch in nucleosome core particles. This argues for a stoichiometric displacement to free cGAS for DNA binding in the cytoplasm. It is hard to understand this argument given that nucleosomes are in vast excess to either Mre11 or cGAS. How could Mre11 be required to displace cGAS from a vast pool of nucleosomes? Perhaps more plausible is displacement from NCPs in micronuclei, where there is far less chromatin. Some discussion of this point is warranted as alternative models are also possible. For example, is Mre11 activity in resection required to generate nucleosome free regions that could activate cGAS?

We fully agree with the reviewer's suggestion that spatial confinement of micronuclei, in conjunction with known DNA fragmentation mechanisms that occur within micronuclei, may be the context where MRN is most critical for releasing cGAS from nucleosome inhibition, which is mentioned in the revised discussion section. While we do not observe a role for Mre11 nuclease activity in activating cGAS in the presence of nucleosome fragments in biochemically reconstituted experiments (see **Extended Data Fig. 7h**), we cannot exclude indirect effects of

Mre11 nuclease activity that may give rise to DNA repair byproducts that can stimulate cGAS activity, which is also discussed in the revised discussion section.

5. More defined Mre11 mutants that no longer displace cGAS would be valuable to test in cells. It would be interesting to determine if such mutants recapitulate the null phenotypes observed.

We agree that a separation of function mutant of Mre11 that only impacts cGAS activation would be highly desirable. In the revised manuscript, we show more definitively that Mre11 nuclease activity is dispensable for cGAS activation by cytosolic DNA. However, the DNA binding domains of Mre11 are required (see **Fig. 3g and Extended Data Fig. 6f-g**). Because DNA binding is required for Mre11 DNA repair functions, it may not be possible to develop a separation of function Mre11 mutant that only disrupts its role in cGAS activation.

References

1. Mirzoeva, O. K. & Petrini, J. H. J. DNA replication-dependent nuclear dynamics of the Mre11 complex. *Mol. cancer Res. : MCR* 1, 207–18 (2003).
2. Foster, S. S., Balestrini, A. & Petrini, J. H. J. Functional Interplay of the Mre11 Nuclease and Ku in the Response to Replication-Associated DNA Damage. *Mol. Cell. Biol.* 31, 4379–4389 (2011).
3. Robison, J. G., Elliott, J., Dixon, K. & Oakley, G. G. Replication Protein A and the Mre11·Rad50·Nbs1 Complex Co-localize and Interact at Sites of Stalled Replication Forks*. *J. Biol. Chem.* 279, 34802–34810 (2004).
4. Olson, E. *et al.* The Mre11 Complex Mediates the S-Phase Checkpoint through an Interaction with Replication Protein A. *Mol. Cell. Biol.* 27, 6053–6067 (2007).
5. Fagan-Solis, K. D. *et al.* A P53-Independent DNA Damage Response Suppresses Oncogenic Proliferation and Genome Instability. *Cell Rep.* 30, 1385-1399.e7 (2020).
6. Bartkova, J. *et al.* Aberrations of the MRE11–RAD50–NBS1 DNA damage sensor complex in human breast cancer: MRE11 as a candidate familial cancer-predisposing gene. *Mol. Oncol.* 2, 296–316 (2008).
7. Alblihy, A. *et al.* Untangling the clinicopathological significance of MRE11-RAD50-NBS1 complex in sporadic breast cancers. *npj Breast Cancer* 7, 143 (2021).
8. Volkman, H. E., Cambier, S., Gray, E. E. & Stetson, D. B. Tight nuclear tethering of cGAS is essential for preventing autoreactivity. *eLife* 8, e47491 (2019).
9. Zhao, B. *et al.* The Molecular Basis of Tight Nuclear Tethering and Inactivation of cGAS. *Nature* 587, 673–677 (2020).
10. Boyer, J. A. *et al.* Structural basis of nucleosome-dependent cGAS inhibition. *Science* 370, 450–454 (2020).
11. Cao, D., Han, X., Fan, X., Xu, R.-M. & Zhang, X. Structural basis for nucleosome-mediated inhibition of cGAS activity. *Cell Res.* 30, 1088–1097 (2020).
12. Kujirai, T. *et al.* Structural basis for the inhibition of cGAS by nucleosomes. *Science* 370, 455–458 (2020).
13. Jiao, H. *et al.* Z-nucleic-acid sensing triggers ZBP1-dependent necroptosis and inflammation. *Nature* 580, 391–395 (2020).

14. Nassour, J. *et al.* Telomere-to-mitochondria signalling by ZBP1 mediates replicative crisis. *Nature* 614, 767–773 (2023).
15. Lei, Y. *et al.* Cooperative sensing of mitochondrial DNA by ZBP1 and cGAS promotes cardiotoxicity. *Cell* (2022) doi:10.1016/j.cell.2023.05.039.

Reviewer Reports on the First Revision:

Referees' comments:

Referee #1 (Remarks to the Author):

The authors made a tremendous effort to revise the manuscript, which includes a lot of new experiments, and it has improved considerably. The majority of my comments have been addressed. The manuscript uncovers an unexpected nuclease-independent role of MRN in the activation of cGAS, and will be of interest to a broad audience.

Referee #2 (Remarks to the Author):

The manuscript by Cho et al. entitled "Mre11 liberates cGAS from nucleosome sequestration during tumorigenesis" identified Mre11 as an essential component of cGAS activation that suppresses Myc-induced proliferation and inhibits tumorigenesis using an in vivo CRISPR screen. The authors mechanistically describe that the MRN complex (Mre11-Rad50-Nbn) activate cGAS by displacing cGAS from AP sequestration in response to oncogene-induced DNA damage. The authors propose that cGAS activation suppresses Myc-induced proliferation via ZBP1/RIPK3/MLKL mediated necroptosis, ultimately suppressing tumorigenesis in breast cancer. The authors have addressed most comments in their revision, while there are still some minor issues can be further addressed.

Major comments

Figure 3i, a negative control is needed.

Minor comments

1. Figure 2 d-e, standard errors of the sgCon and sgMre11 have great variation. Make sure t-test is applicable to your data.
2. Figure 3m, are these images taken at the same magnification?
3. Figure 4c, how does Mre11 KO affect the expression of unphosphorylated MLKL?
4. Figure 3h, abbreviations are not consistent.

Referee #3 (Remarks to the Author):

The authors have returned a revised study that comprehensively addresses all reviewer concerns. This includes better defined biochemistry studies and the judicious use of cGAS and Mre11 mutants that support a model that the MRN partially antagonizes cGAS nucleosome sequestration to stimulate its activities. Additional advances include the more thorough genetic analyses of ZBP1 and cGAS in the c-Myc, p53 mutant murine tumor model. These advances complement the already important use of in vivo screening to identify Mre11 as a tumor suppressor in this model. The work is important and will be of broad interest to the Nature readership. I am supportive of publication without further revision.

Author Rebuttals to First Revision:

Response to Reviewers

Referee #1 (Remarks to the Author):

The authors made a tremendous effort to revise the manuscript, which includes a lot of new experiments, and it has improved considerably. The majority of my comments have been addressed. The manuscript uncovers an unexpected nuclease-independent role of MRN in the activation of cGAS, and will be of interested to a broad audience.

Thank you for the positive feedback.

Referee #2 (Remarks to the Author):

The manuscript by Cho et al. entitled “Mre11 liberates cGAS from nucleosome sequestration during tumorigenesis” identified Mre11 as an essential component of cGAS activation that suppresses Myc-induced proliferation and inhibits tumorigenesis using an in vivo CRISPR screen. The authors mechanistically describe that the MRN complex (Mre11-Rad50-Nbn) activate cGAS by displacing cGAS from AP sequestration in response to oncogene-induced DNA damage. The authors propose that cGAS activation suppresses Myc-induced proliferation via ZBP1/RIPK3/MLKL mediated necroptosis, ultimately suppressing tumorigenesis in breast cancer. The authors have addressed most comments in their revision, while there are still some minor issues can be further addressed.

Major comments

Figure 3i, a negative control is needed.

The appropriate negative control has been added.

Minor comments

1. Figure 2 d-e, standard errors of the sgCon and sgMre11 have great variation. Make sure t-test is applicable to your data.

We have consulted with our statistics core, and they considered use of a t-test appropriate for our data. While the standard deviation may be larger, there is no reason to reject a normality assumption.

2. Figure 3m, are these images taken at the same magnification?

Yes they were taken at the same magnification. However, the prior example included a cell that likely had undergone endoreduplication, resulting in it being a larger size. For ease of interpretation, we have updated the representative images to avoid this issue. The quantification of the data remains unchanged.

3. Figure 4c, how does Mre11 KO affect the expression of unphosphorylated MLKL?

Total MLKL levels are unchanged in the Mre11 KO. This can now be seen in Supplementary Figure 1, which includes the raw Western Blot image files corresponding to Figure 4C.

4. Figure 3h, abbreviations are not consistent.

We have updated the abbreviations to conform with the rest of the manuscript.

Referee #3 (Remarks to the Author):

The authors have returned a revised study that comprehensively addresses all reviewer concerns. This includes better defined biochemistry studies and the judicious use of cGAS and Mre11 mutants that support a model that the MRN partially antagonizes cGAS nucleosome sequestration to stimulate its activities. Additional advances include the more thorough genetic analyses of ZBP1 and cGAS in the c-Myc, p53 mutant murine tumor model. These advances complement the already important use of in vivo screening to identify Mre11 as a tumor suppressor in this model. The work is important and will be of broad interest to the Nature readership. I am supportive of publication without further revision.

Thank you for the positive feedback.